# Fractal decompositions and tensor network representations of Bethe wavefunctions

Subhayan Sahu[1] and Guifre Vidal[2]

[1] *Perimeter Institute for Theoretical Physics, Waterloo, Ontario N2L 2Y5, Canada*
[2] *Google Quantum AI, Goleta, CA 93111, USA*

We investigate the entanglement structure of a generic $M$-particle Bethe wavefunction (not necessarily an eigenstate of an integrable model) on a 1d lattice by dividing the lattice into $L$ parts and decomposing the wavefunction into a sum of products of $L$ local wavefunctions. We show that a Bethe wavefunction accepts a *fractal* multipartite decomposition: it can always be written as a linear combination of $L^M$ products of $L$ local wavefunctions, where each local wavefunction is in turn also a Bethe wavefunction. Building upon this result, we then build *exact, analytical* tensor network representations with finite bond dimension $\chi = 2^M$, for a generic planar tree tensor network (TTN), which includes a matrix product states (MPS) and a regular binary TTN as prominent particular cases. For a regular binary tree, the network has depth $\log_2(N/M)$ and can be transformed into an adaptive quantum circuit of the same depth, composed of unitary gates acting on $2^M$-dimensional qudits and mid-circuit measurements, that deterministically prepares the Bethe wavefunction. Finally, we put forward a much larger class of *generalized* Bethe wavefunctions, for which the above decompositions, tensor network and quantum circuit representations are also possible.

## I. INTRODUCTION

**Motivation:**

Integrable many-body Hamiltonians in one spatial dimension are a special class of models characterized by an extensive number of locally conserved quantities. Using the Bethe Ansatz, introduced by Hans Bethe in 1931 for the spin half Heisenberg model [1], we can solve a variety of integrable models with particle number conservation, including the XXZ model [2], one-dimensional Bose gas with delta function interactions [3], the Hubbard model [4] etc. The coordinate Bethe Ansatz represents eigenstates of a quantum-integrable Hamiltonian as linear superpositions of plane waves and provides explicit expressions for the corresponding eigenenergies, scalar products, correlation functions, etc. An alternative characterization is provided by the Algebraic Bethe Ansatz [5, 6] (see [7–9] for recent reviews). In spite of the solvable character of integrable models, the exact computation of their long-range and high order correlation functions and of dynamical properties remains a challenging task. Consequently, it makes sense to attempt to compute such quantities using numerical and experimental approaches.

On the other hand, progress in our understanding of the structure of quantum entanglement has led to remarkable insights into the universal properties of quantum matter [10–13], as well as informed the development of numerical and conceptual tools such as tensor networks. Gapped ground states in one dimension can be often well-approximated by one-dimensional tensor networks, namely, matrix product states (MPS) [14] of modest bond dimension [15, 16]. Generalizations of tensor networks for systems in higher dimensions such as projected entangled-pair states (PEPS) [17], and with holographic geometries such as tree tensor networks (TTN) [18] and multiscale entanglement renormalization ansatz (MERA) [19, 20] have been used successfully to represent quantum many-body states of interest. Tensor network representations of many-body wavefunctions are not only relevant for their classical simulation, but they sometimes also provide a blueprint for preparing such states on a quantum computer [21–25], some of which have also been explored in experiments [26–28].

In this work, we investigate the structure of Bethe wavefunctions from the perspective of quantum entanglement, by investigating how to decompose them as a linear combination of local wavefunctions when we partition the 1d lattice into two or more parts. In doing so, we uncover a surprising property of Bethe wavefunctions, namely that we can write their bipartite and multipartite decompositions in terms of local wavefunctions that are Bethe wavefuctions themselves. These *fractal* decompositions of an $M-$particle Bethe wavefunction, which are also seen to only require $2^M$ local Bethe wavefunction on each part of the lattice, further lead to simple analytical expressions for their (exact) tensor network representation as an MPS and as a TTN. From the latter, we then

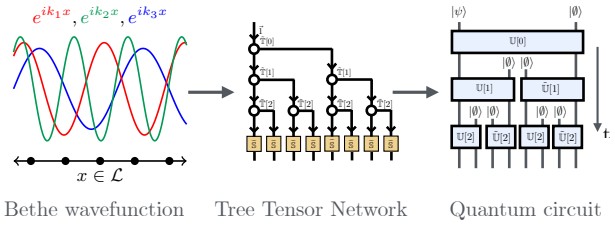

Bethe wavefunction     Tree Tensor Network     Quantum circuit

FIG. 1. The kinematic constraints in a Bethe wavefunction lead to an exact tree tensor network representation with bond dimension $\chi = 2^M$ and a quantum circuit with depth $\log_2(N/M)$ (made of unitary gates on $2^M$-dimensional qudits and mid-circuit measurements) for deterministically preparing such states using a quantum computer.

build a short depth quantum circuit that could be used to prepare the Bethe wavefunction on a quantum computer.

We highlight that our results are not specific to Bethe wavefunctions that are eigenstates of integrable Hamiltonians, namely Bethe wavefunctions satisfying the relevant integrability constraints. Rather, our results arise from the built-in *kinematic* constraints of the Bethe wavefunction itself, without the need to further require that the integrability conditions be satisfied. This scenario has also been recently considered by several authors in Refs. [29–31]. Finally, we also put forward a generalized Bethe wavefunction that retains all the key properties exploited in this work, including the fractal character of its bipartite and multipartite decompositions, which again leads to explicit MPS, TTN and quantum circuit representations, see Fig. 2.

It is highly remarkable that, almost a century after its proposal by Hans Bethe, we are still gaining new insights into the structure of Bethe wavefunctions, mostly thanks to the relatively new lens of quantum information and quantum computation. Indeed, closely related questions to the ones addressed in this paper have been previously studied by a number of authors over the last 20 years [29–38]. Below we first provide a more exhaustive summary of our results, followed by a comparison between our work and previous literature.

**Results:**

Let the integers $M$ and $N$ denote the number of particles in a Bethe wavefunction and the number of sites in the 1d lattice $\mathcal{L}$ on which the wavefunction is defined. Each Bethe wavefunction is specified by a set of parameters, the Bethe data, namely $M$ *quasi-momenta* and $M(M-1)/2$ *scattering angles*. To investigate its entanglement structure, we decompose a Bethe wavefunction as a linear combination of products of local wavefunctions according to a partition of the lattice $\mathcal{L}$ into $L$ parts. We first find that Bethe wavefunctions have a very particular bipartite structure ($L = 2$ parts). We then see that their multipartite structure (for $L > 2$ parts) is also very special in a related manner.

To explain the bipartite and multipartite structure of Bethe wavefunctions, we must introduce the concept of *local* Bethe wavefunctions, which are defined analogously to a *global* Bethe wavefunction but supported only on a *part* of the 1d lattice $\mathcal{L}$, namely on a subset of adjacent sites. Given an $M$-particle Bethe wavefunction, we can build $2^M$ local Bethe wavefunctions compatible with the global Bethe data. Each such local Bethe wavefunction has a fixed number $m$ of particles, ranging from 0 to $M$, with $m$ corresponding quasi-momenta and $m(m-1)/2$ scattering angles. We find two key results. First, in any (contiguous) part of the lattice, a Bethe wavefunction is locally supported on a subspace of dimension at most $2^M$. Second, this subspace is spanned by the $2^M$ local Bethe wavefunctions compatible with the global Bethe data. These two results then have a number of consequences in how the Bethe wavefunction can be expanded as a linear combination of product states.

In particular, when we partition the lattice into two parts ($L = 2$), we obtain a bipartite decomposition of the Bethe wavefunction as a linear combination of (at most) $2^M$ products of two local Bethe wavefunctions. This fractal decomposition automatically implies that the Schmidt rank, or minimal number of product terms in a bipartite decomposition, is upper bounded by $2^M$. This is remarkable, as this number is independent of the number of sites $N$ in the lattice or in any of its two parts, in sharp contrast with what we would encounter when considering generic wavefunctions with the same particle number, also when considering the subset of such wavefunctions that are the ground state of a particle-preserving local Hamiltonian. More generally, for $L$ parts, our fractal decomposition of a Bethe wavefunction consists of a linear combination of (at most) $L^M$ products of $L$ local Bethe wavefunctions. Again, this implies that the multipartite Schmidt rank (minimal number of product terms in an $L$-partite decomposition) is upper-bounded by $L^M$, independently of the size $N$ of the lattice (or of the size of any of its $L$ parts).

Furthermore, a global Bethe wavefunction may be hierarchically decomposed, via sequential bipartite decompositions, according to a pattern of bipartitions organized as a planar tree graph, which leads to a planar TTN representation of the Bethe wavefunction. In each such bipartite decomposition, up to $2^M$ Bethe wavefunctions are simultaneously decomposed into (at most) $2^M$ products of two local Bethe wavefunctions. By capturing the corresponding bipartite decomposition coefficients in a three-index tensor, we arrive at an analytical expression for the tensors in the exact TTN representation of the Bethe wavefunction. This includes tensor networks such as an MPS and a regular binary TTN representations as prominent particular examples, see Fig. 2b. The bond dimension $\chi$ in these exact tensor network representations is upper bounded by $\chi \leq 2^M$. Moreover, for an MPS and regular binary TTN, one can choose the tensor network to be *homogeneous* in space (i.e. the same tensors are repeated throughout all, or parts of, the network), even when the underlying Bethe wavefunction is not translationally invariant.

In particular, a regular binary TTN of a Bethe wavefunctions of $M$ particles on $N$ sites consists of $\log_2(N/M)$ layers of tensors, each one with bond dimension $2^M$. In the homogeneous case, this provides a compact representation in terms of just $\log_2(N/M)$ distinct tensors and a fast method for computing the norm of a Bethe wavefunction as well as the overlap between two Bethe wavefunctions with computational time that scales as $O(\log(N))$ in the system size $N$.

Moreover, this regular binary TTN representation also provides, after we bring it into a canonical form, an efficient quantum circuit that can prepare the Bethe wavefunctions on a quantum computer. Specifically, to a local adaptive quantum circuit with depth $O(\log(N))$, that uses local measurements and local unitary gates conditioned on the measurement outcomes. See Fig. 1 for a

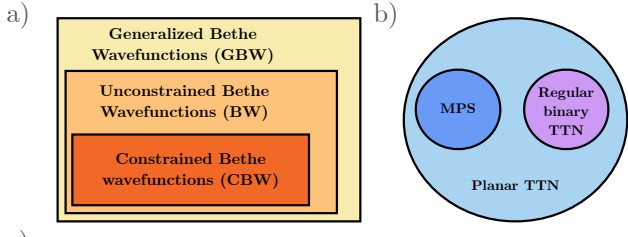

FIG. 2. (a) Hierarchy of Bethe wavefunctions: *constrained* (fulfilling integrability constraints), *unconstrained* (or simply Bethe wavefunctions; see definition II.2; not fulfilling integrability constraints) and *generalized* (see definition VII.1). (b) Our explicit tensor network representations for Bethe wavefunctions cover any planar tree tensor network (planar TTN), which includes the matrix product states (MPS) and the regular binary tree tensor network (regular binaty TTN) as particular cases. (c) In this work we propose fractal decompositions, as well as planar TTN representations (including MPS and regular binary TTN representations) for Bethe wavefunctions, then also for generalized Bethe wavefunctions, thus covering the 12 entries of this table. Several important particular cases had been previously covered in the literature, most notably the MPS representation of constrained Bethe wavefunctions in Refs. [32–34] and more recently of unconstrained Bethe wavefunctions in Refs. [29, 31] and a version of the generalized Bethe wavefunction in Ref. [31]. As explained in a note added at the end of the paper, the fractal bipartite decomposition for constrained Bethe wavefunctions (in the algebraic Bethe wavefunction formalism) had been previously proposed in Ref. [6].

schematic summary of this application.

Finally, we introduce a larger class of wavefunctions that, obeying similar kinematic constraints as a Bethe wavefunction, have analogous properties in terms of accepting $L$-partite decompositions as a linear combination of $L^M$ terms, each being the product of $L$ *local* wavefunctions in the same class, and exact MPS and regular binary TTN representation with bond dimension $2^M$. This generalized Bethe wavefunctions are specified by $M$ *completely arbitrary* single-particle wavefunctions and $M(M-1)/2$ *complex* scattering angles (instead of the $M$ *plain wave* single-particle wavefunctions and $M(M-1)/2$ *real* scattering angles of regular Bethe wavefunctions).

**Comparison to previous works:**

Our results build on top of, and overlap with, several lines of recent progress on this topic. From the perspective of our work, previous contributions can be organized in three phases. First, MPS representations of constrained Bethe wavefunctions were discovered in [32–34],

where the authors used algebraic Bethe ansatz techniques for specific models satisfying the integrability conditions. They reached the conclusion that the MPS has a bond dimension $\chi = 2^M$. Second, recent works have considered the prospect of using quantum simulators to prepare Bethe wavefunctions [30, 35–38], building on the earlier results on the MPS representations. Such state preparation can be useful, in the context of variational quantum eigensolver algorithms, to solve Bethe equations and extract correlation functions and other static and dynamic properties of Bethe wavefunctions using a quantum computer. These works provide a deterministic (unitary) quantum circuit of linear depth $O(N)$. Third, a set of recent works [29, 31] have provided MPS representation and quantum circuits for unconstrained Bethe wavefunctions, without requiring the integrability conditions to be satisfied.

In our work we produce an explicit tensor network representation for an unconstrained Bethe wavefunction for *any* planar tree tensor network. For the particular case of an MPS (that is, when the tree graph is a 1d chain), our tensor network representation becomes equivalent to the MPS representation previously proposed by Ruiz, Sopena, Gordon, Sierra and López in Ref. [29]. Furthermore, the notion of a local Bethe wavefunction can be seen to also emerge (on the right side of a bipartition) in the MPS construction of Ref. [29] as well as in Ref. [31] by Ruiz, Sopena, López, Sierra and Pozsgay, see Fig. 2c. Finally, our generalized Bethe wavefunction, which depends on a number of parameters that scales with $N$ as $O(MN)$, includes as a particular case the inhomogeneous Bethe wavefunctions (which depend on $M + N$ parameters) recently and independently proposed in Ref. [31], and can be seen to actually equate that proposal if we omit the integrability constraints imposed in Ref. [31].

We also highlight that our TTN representation, with bond dimension $\chi = 2^M$, is possible thanks to a special property of Bethe wavefunctions and as such it could not be anticipated from the MPS representation by applying the usual conversion procedure that maps an MPS into a TTN. Indeed, such conversion procedure generically transforms an MPS with bond dimension $\chi_{\mathrm{MPS}}$ into a TTN with bond dimension $\chi_{\mathrm{TTN}} = (\chi_{\mathrm{MPS}})^2$, which for $\chi_{\mathrm{MPS}} = 2^M$ would have been expected to lead to $\chi_{\mathrm{TTN}} = 4^M$, much larger than the bond dimension $\chi_{\mathrm{TTN}} = 2^M$ in our TTN representation. One can ultimately trace the significant reduction in the bond dimension of the TTN representation to the two-particle factorization of scattering amplitudes in Bethe wavefunctions. The quantum circuit derived from our exact tree tensor network improves upon the depth of the existing deterministic quantum circuit to prepare such states exponentially for constant number of particles, from $O(N)$ to $O(\log N)$. This, however, comes at a cost of requiring mid-circuit measurements and adaptive unitary gate control.

**Outline:**

We provide an outline for the rest of the paper. In Sec. II we review the definition of Bethe wavefunctions.

In Sec. III we describe and derive a central result of our work: the fractal decomposition of a Bethe wavefunction as a sum of products of local Bethe wavefunctions, which we generalize to the multipartite case in Sec. IV. In Sec. V we use the fractal decomposition to construct exact MPS and TTN representations of Bethe wavefunctions with bond dimension $\chi = 2^M$. We use the TTN representation to construct a compact quantum circuit to prepare such a state in Sec. VI. Finally, in Sec. VII we introduce the generalized Bethe wavefunctions, and we summarise the main results and conclude in Sec. VIII.

## CONTENTS

## II. BETHE WAVEFUNCTIONS

In this work we study Bethe wavefunctions, which are trial states for Bethe ansatz integrability in one dimensional (1d) quantum systems with a $U(1)$ symmetry. This symmetry often corresponds to the conservation of the number of particles (in fermionic/bosonic models) or the number of magnetic excitations or magnons (in spin models). From now on we will simply use the term *particle* to refer to all these cases.

Below we introduce a Bethe wavefunction with $M$ particles, characterized by a set of *quasi-momenta* and 2-particle *scattering angles*, which we call *Bethe data*. For specific values of these parameters, a Bethe wavefunction can be an eigenstate of some integrable model Hamiltonian [1, 9, 39]. However, all the results presented in this work, including the bipartite and multipartite decompositions of Bethe wavefunctions in Secs. III and IV, the exact tensor network representations in Sec. V and the quantum circuit representation in Sec. VI, apply to generic values of the Bethe data - that is, they do not assume that the Bethe wavefunction relates to an integrable model.

We start by considering generic wavefunctions of $M$ indistinguishable particles.

### A. Generic M-particle wavefunctions

Consider a 1d lattice $\mathcal{L}$ made of $N$ sites, where the sites are labelled by an integer $x$, with $1 \leq x \leq N$. Each lattice site hosts a 2-dimensional vector space $\mathbb{C}_2$ (or qubit) spanned by vectors $|\sigma\rangle \in \{|0\rangle, |1\rangle\}$ corresponding to the absence or presence of a particle. We refer to this basis as the occupation basis throughout the paper.

The Hilbert space $\mathcal{H}^{\mathcal{L}} \cong (\mathbb{C}_2)^{\otimes N}$ decomposes into a direct sum of subspaces $\mathcal{H}^{\mathcal{L}}_M$ with well-defined particle number $M$,

$$\mathcal{H}^{\mathcal{L}} = \bigoplus_M \mathcal{H}^{\mathcal{L}}_M, \quad d^{\mathcal{L}}_M = \binom{N}{M} = \frac{N!}{M!(N-M)!}, \quad (1)$$

where $d^{\mathcal{L}}_M$ denotes the dimension of $\mathcal{H}^{\mathcal{L}}_M$. For instance, in the $M = 0$ particle sector, with $d^{\mathcal{L}}_{M=0} = 1$, we find the unique vacuum state $|\emptyset\rangle \equiv |\sigma_1 \sigma_2 \cdots \sigma_N\rangle$ with $\sigma_a = 0$ for $a = 1, 2, \cdots, N$. In the $M = 1$ particle sector, with $d^{\mathcal{L}}_{M=0} = N$, a basis $\{|x\rangle\}$ of single particle states is labeled by the location $x$ of the particle, namely $|x\rangle \equiv |\sigma_1 \sigma_2 \cdots \sigma_N\rangle$, where $\sigma_a = 1$ only if $a = x$ and the rest of $\sigma_a$ vanish, that is $\sigma_a = \delta_{x,a}$. More generally, for $M$ (indistinguishable) particles, we can specify their locations by a vector $\vec{x} = (x_1, x_2, \cdots, x_M)$ with $M$ ordered integers $\{x_j\}$, where $1 \leq x_1 < x_2 < \cdots < x_M \leq N$. We call the set of such vectors the domain $\mathcal{D}^{\mathcal{L}}_M$. Then a basis $\{|\vec{x}\rangle\}$ of $M$ particle states is labeled by the $M$ ordered positions $\vec{x} \in \mathcal{D}^{\mathcal{L}}_M$ of the $M$ particles, namely $|\vec{x}\rangle \equiv |\sigma_1 \sigma_2 \cdots \sigma_N\rangle$ where only $\sigma_a = 1$ if $a \in \vec{x}$, that is $\sigma_a = \sum_j \delta_{x_j,a}$. Notice that there are $\binom{N}{M}$ inequivalent ways of placing $M$ indistinguishable particles in $N$ sites, as reflected in the dimension $d^{\mathcal{L}}_M = |\mathcal{D}^{\mathcal{L}}_M|$ in Eq. (1). For fixed particle number $M$ and large number of sites $N$ we have, to leading order in $N$,

$$d^{\mathcal{L}}_M = \frac{N^M}{M!} - \frac{(M+1)N^{M-1}}{2(M-1)!} + \cdots \sim \frac{N^M}{M!}. \quad (2)$$

**Definition II.1.** *We denote a generic wavefunction of $M$ particles on lattice made of $N$ sites by $|\Psi_M\rangle$, and refer to it as a **generic $(M, N)$ wavefunction**. It can be expressed in the basis $|\vec{x}\rangle$ as*

$$|\Psi_M\rangle = \sum_{\vec{x} \in \mathcal{D}^{\mathcal{L}}_M} \Psi_M(\vec{x}) |\vec{x}\rangle, \quad (3)$$

*where the number of complex amplitudes $\Psi_M(\vec{x})$ is $|\mathcal{D}^{\mathcal{L}}_M| = d^{\mathcal{L}}_M = \binom{N}{M}$. Therefore, for fixed particle number $M$, the wavefunction is specified by a number of complex amplitudes that scales with the number of sites as $d^{\mathcal{L}}_M \sim N^M/M!$, see Eq. (2).*

### B. M-particle Bethe wavefunction

A Bethe wavefunction $|\Phi_M\rangle$ is a restricted class of $M$-particle wavefunction $|\Psi_M\rangle \in \mathcal{H}^{\mathcal{L}}_M$ that can be intuitively

thought of as some form of a multi-particle 1d plane-wave wavefunction with generalized scattering angles that describe interacting particles.

A note on notation: Throughout this section and also Sec. III we use $\Phi$ to denote a Bethe wavefunction, and thus differentiate it from a generic wavefunction, denoted by $\Psi$. This may include a number of additional subscripts and superscripts, such as in $\left|\Phi_M^{\vec{k},\boldsymbol{\theta}}\right\rangle$ or $\left|\Phi_m^{\vec{c}}\right\rangle$, which we use to specify the particle number and the Bethe data. [Starting in Sec. IV, we will use a more compact notation, such as $\left|\vec{1}\right\rangle$, to denote a Bethe wavefunction.]

Although a non-trivial Bethe wavefunction requires $M \geq 2$ particles, it is useful to include the cases $M = 0, 1$ in the same formalism. They correspond to the vacuum and to a single-particle 1d plane wave with quasi-momentum $k$,

$$\left|\Phi_{M=0}\right\rangle = \left|\emptyset\right\rangle, \qquad \left|\Phi_{M=1}^{(k)}\right\rangle \equiv \sum_{x\in\mathcal{L}} e^{ikx}\left|x\right\rangle. \qquad (4)$$

Note that $\left\langle\Phi_{M=1}^{(k)}\middle|\Phi_{M=1}^{(k)}\right\rangle = N$, that is $\left|\Phi_{M=1}^{(k)}\right\rangle$ is not normalized to 1. In most of this work we will consider unnormalized wavefunctions such as $\left|\Phi_{M=1}^{(k)}\right\rangle$. [In Sec. V we will explain how to efficiently compute the norm of a Bethe wavefunction (more generally, the overlap between two Bethe wavefunctions) using an exact TTN representation.]

For $M = 2$, the first non-trivial case, a two-particle Bethe wavefunction is given in terms of two quasi-momenta $k_1$ and $k_2$ and a scattering angle $\theta_{21}$ as

$$\sum_{(x_1,x_2)\in\mathcal{D}_2^{\mathcal{L}}} \left(e^{ik_1x_1}e^{ik_2x_2} - e^{i\theta_{21}}e^{ik_2x_1}e^{ik_1x_2}\right)\left|x_1x_2\right\rangle. (5)$$

We denote such a 2−particle Bethe wavefunction as $\left|\Phi_{M=2}^{\vec{k},\boldsymbol{\theta}}\right\rangle$, where $\vec{k} = (k_1, k_2)$ and $\boldsymbol{\theta} \equiv \{\theta_{21}\}$. It is instructive to note that for the choices of scattering angle $\theta_{21} = 0$ and $\theta_{21} = \pi$, namely for

$$\sum_{(x_1,x_2)\in\mathcal{D}_2^{\mathcal{L}}} \left(e^{ik_1x_1}e^{ik_2x_2} \mp e^{ik_2x_1}e^{ik_1x_2}\right)\left|x_1x_2\right\rangle, \qquad (6)$$

we recover a plane wave state of two fermions (minus sign) or two bosons (plus sign), respectively.

More generally, a Bethe wavefunction $\left|\Phi_M^{\vec{k},\boldsymbol{\theta}}\right\rangle$ for an arbitrary number $M$ of particles is characterized by the Bethe data $(\vec{k}, \boldsymbol{\theta})$, consisting of $M$ quasi-momenta $\vec{k} = (k_1, k_2, \cdots, k_M)$ and $M(M-1)/2$ scattering angles $\boldsymbol{\theta} = \{\theta_{j_2j_1}\}$ for $1 \leq j_1 < j_2 \leq M$. In order to build the wavefunction from its data $(\vec{k}, \boldsymbol{\theta})$, first we introduce the set $\mathcal{S}_M$ of permutations of $M$ symbols, namely the *symmetric group*. The symmetric group $\mathcal{S}_M$ contains $|\mathcal{S}_M| = M!$ permutations, each represented by a vector $P = (P_1, P_2, \cdots, P_M)$ whose components $P_j$ are $M$ distinct integers, with $1 \leq P_j \leq M$. For $M = 3$, the $3! = 6$ permutations are described in Fig. 3.

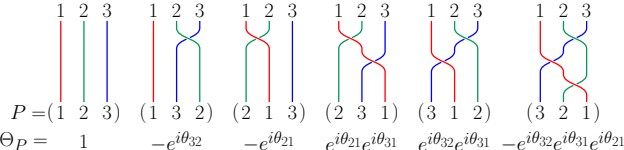

FIG. 3. Example of permutation diagrams. All the permutations for $M = 3$ symbols are described using strands and their crossings, and their corresponding scattering amplitudes are noted below.

Given the vector $\vec{k} = (k_1, k_2, \cdots, k_M)$ of quasi-momenta and a permutation $P$, we denote by $\vec{k}^P \equiv (k_{P_1}, k_{P_2}, \cdots, k_{P_M})$ the vector of quasi-momenta obtained from $\vec{k}$ by permuting its $M$ components according to permutation $P$. In particular, if $e^{i\vec{k}\cdot\vec{x}} = \prod_{j=1}^M e^{ik_jx_j}$ is the amplitude of an $M$-particle plane wave where the particle in position $x_j$ has quasi-momentum $k_j$, then $e^{i\vec{k}^P\cdot\vec{x}} = \prod_{j=1}^M e^{ik_{P_j}x_j}$ corresponds to the amplitude where the particle in position $x_j$ now has quasi-momentum $k_{P_j}$ instead.

**Definition II.2.** *A **Bethe wavefunction** of $M$ particles on a 1d lattice $\mathcal{L}$ made of $N$ sites and Bethe data $(\vec{k}, \boldsymbol{\theta})$ is an $(M, N)$ wavefunction of the form*

$$\left|\Phi_M^{\vec{k},\boldsymbol{\theta}}\right\rangle \equiv \sum_{\vec{x}\in\mathcal{D}_M^{\mathcal{L}}} \sum_{P\in S_M} \Theta[P]e^{i\vec{k}^P\cdot\vec{x}}\left|\vec{x}\right\rangle, \qquad (7)$$

*where the scattering amplitude $\Theta[P]$ factorises into a product of a subset of the two-particle scattering phases $-e^{i\theta_{j_2j_1}}$, namely*

$$\Theta[P] = \prod_{j_1=1}^{M-1}\prod_{j_2=j_1+1}^{M} f_P(j_1, j_2), \text{ where}$$

$$f_P(j_1, j_2) = \begin{cases} 1, & \text{if } P_{j_1} < P_{j_2} \\ -e^{i\theta_{P_{j_1}P_{j_2}}}, & \text{if } P_{j_1} > P_{j_2} \end{cases}. \qquad (8)$$

The above amplitude $\Theta[P]$ for permutation $P$ can be understood in terms of the crossings of two quasi-momenta during the permutation: starting with the originally ordered quasi-momenta $k_1, k_2, \cdots, k_M$, a relative phase $-e^{i\theta_{j_2j_1}}$ for $j_2 > j_1$ is included as a factor in $\Theta[P]$ if the permutation $P$ reverses the order of $k_{j_2}$ and $k_{j_1}$ (that is, if $k_{j_1}$, which is to the left of $k_{j_2}$ in $\vec{k}$, appears to the right of $k_{j_2}$ in the permuted vector $\vec{k}^P$).

For instance, for $M = 2$ particles, the trivial permutation $P = (1, 2)$ produces a trivial amplitude $\Theta[P = (1, 2)] = 1$, whereas the permutation $P = (2, 1)$ leads to the amplitude $\Theta[P = (2, 1)] = f_{P=(2,1)}(1, 2) = -e^{i\theta_{21}}$, and we recover Eq. (5).

As an additional example, for $M = 3$ particles, the amplitudes $\Theta[P]$ for the $3! = 6$ permutations $P \in S_3$ are depicted in Fig. 3. Accordingly, for $M = 3$ particles, the

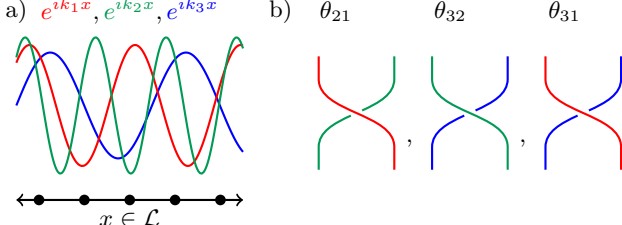

c)  Set of choice vectors $\{\vec{c}\}$ :

$$\{\emptyset, (1), (2), (3), (1,2), (1,3), (2,3), (123)\}$$

FIG. 4. A Bethe wavefunction for $M$ particles on lattice $\mathcal{L}$ (here for $M = 3$) is characterized by (a) M quasi-momenta $\vec{k} = (k_1, k_2, \cdots, k_M)$ and (b) $M(M-1)/2$ scattering angles $\theta_{j_2 j_1}$. From each quasi-momentum $k_j$ we generate a single-particle plain wave with amplitude $e^{ik_j x}$, and to each pair of plain waves we associate a scattering angle $\theta_{j_2 j_1}$. (c) Set of choice vectors for an $M = 3$ particle bethe wavefunction, as described in Sec. II C.

Bethe wavefunction in Eq. (7) reads

$$\left| \Phi_{M=3}^{(k1,k2,k3),\{\theta_{21},\theta_{32},\theta_{31}\}} \right\rangle = \sum_{(x_1,x_2,x_3)\in\mathcal{D}_3^{\mathcal{L}}} \Bigg($$
$$e^{i(k_1 x_1 + k_2 x_2 + k_3 x_3)}$$
$$-e^{i\theta_{32}} e^{i(k_1 x_1 + k_3 x_2 + k_2 x_3)}$$
$$-e^{i\theta_{21}} e^{i(k_2 x_1 + k_1 x_2 + k_3 x_3)}$$
$$+e^{i\theta_{21}} e^{i\theta_{31}} e^{i(k_2 x_1 + k_3 x_2 + k_1 x_3)}$$
$$+e^{i\theta_{32}} e^{i\theta_{31}} e^{i(k_3 x_1 + k_1 x_2 + k_2 x_3)}$$
$$-e^{i\theta_{32}} e^{i\theta_{31}} e^{i\theta_{21}} e^{i(k_3 x_1 + k_2 x_2 + k_1 x_3)} \Bigg) |x_1 x_2 x_3\rangle \quad (9)$$

Note that the data $(\vec{k}, \boldsymbol{\theta})$ for an $M$ particle Bethe wavefunction $|\Phi_M\rangle$ consists of just $M + M(M-1)/2$ real parameters (a number that is independent of $N$) compared to $\binom{N}{M} \sim N^M/M!$ parameters for a generic $(M, N)$ wavefunction $|\Psi_M\rangle$. Thus, for large $N$, Bethe wavefunctions $|\Phi_M\rangle$ are a relatively small subset, depending on $O(N^0)$ parameters, within the set of $(M, N)$ wavefunctions $|\Psi_M\rangle$, which depends on $O(N^M)$ parameters.

### C.  Bethe data and choice vectors

In preparation for the next section, we explain how the data $(\vec{k}, \boldsymbol{\theta})$ for a Bethe wavefunction $\left| \Phi_M^{\vec{k},\boldsymbol{\theta}} \right\rangle$ of $M$ particles can also be used to define other Bethe wavefunctions with $m$ particles, for $m < M$. Given a vector $\vec{c} = (c_1, c_2, \cdots, c_m)$ with $m$ *distinct* positive integers $c_j$ such that $1 \leq c_j \leq M$, where we also enforce that the components are increasingly ordered, namely $1 \leq c_1 < c_2 < \cdots < c_m \leq M$, we define *reduced* Bethe

data $(\vec{k}^{\vec{c}}, \boldsymbol{\theta}^{\vec{c}})$ for an $m$ particle wavefunction by

$$\vec{k}^{\vec{c}} \equiv (k_{c_1}, k_{c_2}, \cdots, k_{c_m}), \quad (10)$$
$$\boldsymbol{\theta}^{\vec{c}} \equiv \{\theta_{c_{j_2} c_{j_1}}\} \text{ for } 1 \leq j_1 < j_2 \leq m. \quad (11)$$

That is, we collect the $m$ quasi-momenta $\{k_{c_j}\}$ that are found in positions $\{c_j\}$ of $\vec{k}$, as well as the $m(m-1)/2$ scattering angles $\{\theta_{c_{j_2} c_{j_1}}\}$ in positions $\{c_{j_2}, c_{j_1}\}$ of $\boldsymbol{\theta}$. We call the vector $\vec{c}$ with a choice of $m$ symbols (out of the $M$ symbols $1, 2, \cdots, M$) a *choice* vector, or simply a *choice*.

For instance, for $M = 4$ particles and $m = 2$, the choice $\vec{c} = (1, 3)$ leads to the reduced data $\vec{k}^{\vec{c}} = (k_1, k_3)$ and $\boldsymbol{\theta}^{\vec{c}} = \{\theta_{31}\}$, corresponding to the $m = 2$ particle Bethe wavefunction $\left| \Phi_{m=2}^{(k_1,k_3),\{\theta_{31}\}} \right\rangle$.

Given $M$ particles, there are $\binom{M}{m}$ possible choice vectors $\vec{c}$ with $m$ particles. We denote the set of choices with $m$ particles $\mathcal{C}_m^M$, and write $\vec{c} \in \mathcal{C}_m^M$.

Thus, by considering all possible choices $\vec{c}$ or, equivalently, all possible reduced Bethe data $(\vec{k}^{\vec{c}}, \boldsymbol{\theta}^{\vec{c}})$ for different number $m = 0, 1, \cdots, M$ of particles, we can generate $\sum_{m=0}^{M} |\mathcal{C}_m^M| = \sum_{m=0}^{M} \binom{M}{m} = 2^M$ different Bethe wavefunctions from the original Bethe data $(\vec{k}, \boldsymbol{\theta})$. Notice that for $m = 0$ particles we always have the trivial, zero-component choice vector $\vec{c} = \emptyset$, corresponding to the vacuum $|\emptyset\rangle$, whereas for $m = M$ particles we have the unique $M$-component choice vector $\vec{c} = \vec{1}$, where we define $\vec{1} \equiv (1, 2, \cdots, M)$, corresponding to the original Bethe wavefunction with full data $(\vec{k}, \boldsymbol{\theta})$.

Let us exemplify the above for the cases of $M = 0, 1, 2, 3$ particles. First, for $M = 0$ particles, the only choice is the zero-component vector $\vec{c} = \emptyset$, corresponding to $|\emptyset\rangle$. Next, for $M = 1$ particles, where the Bethe data consists of a quasi-momenta $(k)$, we find that $m = 0$ and $m = 1$ correspond to the choice vectors $\vec{c} = \emptyset$ for the vacuum $|\emptyset\rangle$ and $\vec{c} = \vec{1} = (1)$ for the original Bethe wavefunction $\left| \Phi_{M=1}^{(k)} \right\rangle$, respectively.

The first non-trivial case is $M = 2$ particles, with Bethe data $((k_1, k_2), \{\theta_{21}\})$. For $m = 0, 1, 2$ we find $1,2,1$ choices, respectively, namely

$$m = 0, \qquad \vec{c} = \emptyset, \qquad |\Phi_{m=0}\rangle = |\emptyset\rangle$$
$$m = 1, \qquad \vec{c} = (1), \qquad \left| \Phi_{m=1}^{(k_1)} \right\rangle$$
$$\vec{c} = (2), \qquad \left| \Phi_{m=1}^{(k_2)} \right\rangle$$
$$m = 2, \ \vec{c} = \quad \vec{1} = (1, 2), \quad \left| \Phi_{m=2}^{(k_1,k_2),\{\theta_{21}\}} \right\rangle \quad (12)$$

Finally, for $M = 3$ particles, with Bethe data $((k_1, k_2, k_3), \{\theta_{21}, \theta_{31}, \theta_{32}\})$, $m = 0, 1, 2, 3$ leads to

$1, 3, 3, 1$ choices, respectively. They are:

$$m = 0, \qquad \vec{c} = \emptyset, \qquad |\Phi_{m=0}\rangle = |\emptyset\rangle$$

$$m = 1, \qquad \vec{c} = (1), \qquad \left|\Phi_{m=1}^{(k_1)}\right\rangle$$

$$\vec{c} = (2), \qquad \left|\Phi_{m=1}^{(k_2)}\right\rangle$$

$$\vec{c} = (3), \qquad \left|\Phi_{m=1}^{(k_3)}\right\rangle$$

$$m = 2, \qquad \vec{c} = (1,2), \qquad \left|\Phi_{m=2}^{(k_1,k_2),\{\theta_{21}\}}\right\rangle$$

$$\vec{c} = (1,3), \qquad \left|\Phi_{m=2}^{(k_1,k_3),\{\theta_{31}\}}\right\rangle$$

$$\vec{c} = (2,3), \qquad \left|\Phi_{m=2}^{(k_2,k_3),\{\theta_{32}\}}\right\rangle$$

$$m = 3, \; \vec{c} = \quad \vec{1} = (1,2,3), \; \left|\Phi_{m=3}^{(k_1,k_2,k_3),\{\theta_{21},\theta_{31},\theta_{32}\}}\right\rangle \quad (13)$$

## III. FRACTAL BIPARTITE DECOMPOSITION

The first result of this paper, Theorem III.1, is a bipartite decomposition of a Bethe wavefunction as a sum of $2^M$ terms, where each term is a product of two local Bethe wavefunctions.

Before we present this decomposition, we review the structure of a Hilbert space of a bipartition made of parts $A$ and $B$ in the presence of particle conservation and the bipartite decomposition of a generic $(M, N)$ wavefunction. We also introduce a decomposition of a permutation $P = R \, S \, Q^{\vec{a},\vec{b}}$ in terms of local permutations $R$ and $S$ for parts $A$ and $B$ and a global permutation $Q^{\vec{a},\vec{b}}$ that depends on local choices $\vec{a}$ and $\vec{b}$. We then explain how the Bethe wavefunction amplitudes factorize into products of amplitudes for parts $A$ and $B$ and a global scattering amplitude $\Theta[Q^{\vec{a},\vec{b}}]$, and introduce local Bethe wavefunctions. We then present the bipartite decomposition in Theorem III.1, followed by a few simple examples.

### A. Bipartition with particle number conservation

Consider a partition of the 1d lattice $\mathcal{L}$, made of $N$ sites, into two regions $A$ and $B$ comprising $N_A$ and $N_B$ sites, respectively, with $N_A + N_B = N$. When the $N_A$ sites of part $A$ are to the left of the $N_B$ sites of part $B$, we call the bipartition a *left-right* bipartition (see Fig. 5).

Given a choice of left-right bipartition, we can express the position vector $\vec{x}$ of $M$ particles as a direct sum of two position vectors $\vec{x}_A$ and $\vec{x}_B$, where $\vec{x}_A$ contains the position of the $M_A$ particles in part $A$ and $\vec{x}_B$ contains the position of the $M_B$ particles in part $B$,

$$\vec{x} = \{\underbrace{x_1, \cdots, x_{M_A}}_{\equiv\, \vec{x}_A}, \underbrace{x_{M_A+1} \cdots, x_M}_{\equiv\, \vec{x}_B}\} = \vec{x}_A \oplus \vec{x}_B. \quad (14)$$

We can then replace the basis vector $|\vec{x}\rangle$ with the tensor product of two basis vectors $|\vec{x}_A\rangle|\vec{x}_B\rangle$ on parts $A$ and $B$

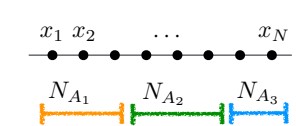

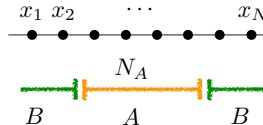 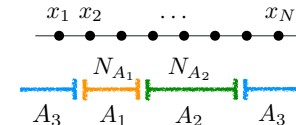

FIG. 5. Types of multi-partitions we consider in this paper.

and the sum $\sum_{\vec{x}}$ with the three sums

$$\sum_{\vec{x} \in \mathcal{D}_M^{\mathcal{L}}} = \sum_{M_A=0}^{M} \; \sum_{\vec{x}_A \in \mathcal{D}_{M_A}^A} \; \sum_{\vec{x}_B \in \mathcal{D}_{M_B}^B}, \qquad (15)$$

where $\mathcal{D}_{M_A}^A$ is the domain of $M_A$ ordered positions in part $A$ and $\mathcal{D}_{M_B}^B$ is the domain of $M_B$ ordered positions in part $B$, with $M_B = M - M_A$. The cardinalities of the reduced domains are $|\mathcal{D}_{M_A}^A| = \binom{N_A}{M_A}$ and $|\mathcal{D}_{M_B}^B| = \binom{N_B}{M_B}$, respectively. In other words, the Hilbert space $\mathcal{H}_M^{\mathcal{L}}$ for $M$ particle states in lattice $\mathcal{L}$ decomposes as a direct sum of tensor products of local Hilbert spaces with $M_A = 0, 1, \cdots, M$ and $M_B \equiv M - M_A$ particles in subregions $A$ and $B$ respectively, namely

$$\mathcal{H}_M^{\mathcal{L}} \cong \bigoplus_{M_A=0}^{M} \mathcal{H}_{M_A}^A \otimes \mathcal{H}_{M_B}^B, \qquad (16)$$

with dimensions $d_{M_A}^A = |\mathcal{D}_{M_A}^A| = \binom{N_A}{M_A}$ and $d_{M_B}^B = |\mathcal{D}_{M_B}^B| = \binom{N_B}{M_B}$, respectively.

A generic $(M, N)$ wavefunction can be decomposed as

$$
\begin{aligned}
|\Psi_M\rangle_{AB} &= \sum_{M_A=0}^{M} \left|\Psi_{(M_A,M_B)}\right\rangle_{AB} \\
&= \sum_{M_A=0}^{M} \sum_{\alpha=1}^{\chi_{M_A}} \lambda_{M_A,\alpha} \left|\Psi_{M_A}^{\alpha}\right\rangle_A \left|\Psi_{M_B}^{\alpha}\right\rangle_B, (17)
\end{aligned}
$$

where the wavefunction $|\Psi_{M_A,M_B}\rangle \in \mathcal{H}_{M_A}^A \otimes \mathcal{H}_{M_B}^B$ has $M_A$ and $M_B = M - M_A$ particles on parts $A$ and $B$, respectively, and in the second line we introduced its Schmidt decomposition $\sum_{\alpha=1}^{\chi_{M_A}} \lambda_{M_A,\alpha} \left|\Phi_{M_A}^{\alpha}\right\rangle_A \left|\Phi_{M_B}^{\alpha}\right\rangle_B$. The (partial) Schmidt rank $\chi_{M_A}$ is generically given by $\min\left(d_{M_A}^A, d_{M_B}^B\right)$ of the dimensions $d_{M_A}^A$ and $d_{M_B}^B$ of $\mathcal{H}_{M_A}^A$ and $\mathcal{H}_{M_B}^B$. For fixed $M$ and large $N \gg M$ (and assuming for simplicity below that $N$ and $M$ are even), these

dimensions are maximized for $N_A = N_B = N/2$, where $d^A_{M_A} \sim (N/2)^{M_A}/(M_A!)$ and $d^B_{M_B} \sim (N/2)^{M_B}/(M_B!)$. We thus see that for large $N$, $d^A_{M_A} \leq d^B_{M-M_A}$ if $M_A \leq M/2$, so that $\chi_{M_A} \sim (N/2)^m/(m!)$ where $m \equiv \min(M_A, M - M_A)$. We conclude that, for constant $M$ and large $N$, the Schmidt rank $\chi = \sum_{M_A=0}^{M} \chi_{M_A}$ for a bipartition with $N_A = N_B = N/2$ is dominated by $\chi_{M_A=M/2} \sim (N/2)^{M_A}/M_A!$, which results in

$$\chi \sim \frac{\left(\frac{N}{2}\right)^{\frac{M}{2}}}{\left(\frac{M}{2}\right)!} \tag{18}$$

to leading order in $N$. This upper bound for the Schmidt rank is seen (e.g. numerically) to be saturated for a generic $(M, N)$ wavefunction. Next, we study the bipartite decomposition of Bethe wavefunctions, which will be seen to contain a number of terms independent of $N$, in sharp contrast with Eq. (18).

## B.  Factorization of Bethe wavefunction amplitudes

To study the bipartite decomposition of Bethe wavefunctions, we need to understand how the scattering amplitude $\Theta[P]$ in Eq. (7) factorizes. We do this in multiple steps; first we discuss the decomposition of permutation $P$.

### Decomposition of permutations

Assuming there are $M_A$ and $M_B$ particles in parts $A$ and $B$, we introduce a canonical decomposition of a permutation $P \in \mathcal{S}_M$ of $M$ symbols with components $P = (P_1, P_2, \cdots, P_{M_A}, P_{M_A+1}, \cdots, P_M)$ as the product

$$P = R\, S\, Q^{\vec{a}, \vec{b}} \tag{19}$$

of three permutations $R$, $S$ and $Q^{\vec{a}, \vec{b}}$. Here $R \in \mathcal{S}_{M_A}$ and $S \in \mathcal{S}_{M_B}$ are *local* permutations of $M_A$ and $M_B$ symbols (with $M = M_A + M_B$), whereas $Q^{\vec{a}, \vec{b}} \in \mathcal{S}_M$ is a *global* permutation of the $M$ symbols chosen to have components

$$Q^{\vec{a}, \vec{b}} \equiv (a_1, a_2, \cdots, a_{M_A}, b_1, b_2, \cdots, b_{M_B}) \equiv \vec{a} \oplus \vec{b}. \tag{20}$$

Specifically, vector $\vec{a} = (a_1, a_2, \cdots, a_{M_A})$ results from increasingly ordering the first $M_A$ components $P_1, P_2, \cdots, P_{M_A}$ of permutation $P$ and vector $\vec{b} = (b_1, b_2, \cdots, b_{M_B})$ is obtained by increasingly ordering the last $M_B$ components $P_{M_A+1}, \cdots, P_M$ of $P$. Notice that $\vec{a}$ is a choice of $M_A$ symbols out of the $M$ symbols (in the language introduced in Sec. II C) and $\vec{b}$ is a another choice with the remaining $M_B$ symbols. To summarize in words, in the canonical decomposition (19) of a permutation $P$ of $M$ symbols in the presence of two parts $A$

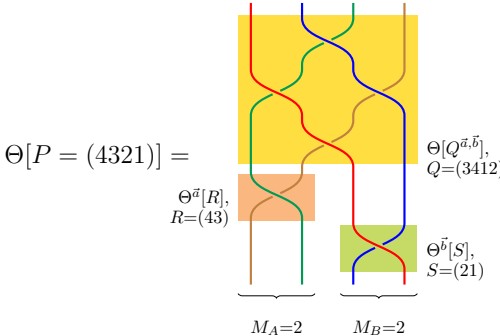

$$\Theta[P = (4321)] =$$

$\Theta^{\vec{a}}[R],$
$R=(43)$

$\Theta[Q^{\vec{a}, \vec{b}}],$
$Q=(3412)$

$\Theta^{\vec{b}}[S],$
$S=(21)$

$M_A=2 \qquad M_B=2$

FIG. 6. Example of factorization of global permutation into local permutations, as well as the corresponding factorization of the scattering amplitudes

and $B$ containing $M_A$ and $M_B$ particles, the global permutation $Q^{\vec{a}, \vec{b}}$ determines which subset of the symbols belongs to each part, and then the local permutations $R$ and $S$ further permute the selected symbols within each part.

A simple example of such factorization is shown in Fig. 6, for the $M = 4$ permutation $P = (4321)$, and the decomposition corresponding to $M_A = 2$ and $M_B = M - M_A = 2$, with $\vec{a} = (3, 4)$ and $\vec{b} = (1, 2)$, and local permutations $R = (4, 3)$ and $S = (2, 1)$.

Following decomposition (19), the sum $\sum_P$ over permutations $P \in \mathcal{S}_M$ in Eq. (7) can be organized as three sums

$$\sum_{P \in \mathcal{S}_M} = \sum_{\vec{a} \in \mathcal{C}^M_{M_A}} \sum_{R \in \mathcal{S}_{M_A}} \sum_{S \in \mathcal{S}_{M_B}}, \tag{21}$$

where we assumed that part $A$ had $M_A$ particles and the sum over choices $\vec{a}$ is a sum over corresponding global permutations $Q^{\vec{a}, \vec{b}}$ (where choice $\vec{b}$ is fully determined by choice $\vec{a}$, see below).

### Choices, choices

Let us introduce some more notation. Given two choice $\vec{a}$ and $\vec{b}$, we define their *union*, denoted $\vec{a} \cup \vec{b}$, as the vector resulting from sorting, in increasing order, the components of the direct sum $\vec{a} \oplus \vec{b}$ of the two choices $\vec{a}$ and $\vec{b}$,

$$\vec{a} \cup \vec{b} \equiv \mathrm{sort}_\uparrow \left( \vec{a} \oplus \vec{b} \right) \tag{22}$$

Notice, for instance, that $\vec{a} \cup \emptyset = \vec{a}$, and that if two choices $\vec{a}$ and $\vec{b}$ have components in common, then $\vec{a} \cup \vec{b}$ is not a choice, since it will have repeated components (recall that the components of a choice $\vec{c}$ are *distinct* positive integers between 1 and $M$, by definition). We then say that two choices $\vec{a}$ and $\vec{b}$ are *complementary*, if their union is $(1, 2, \cdots, M)$, that is,

$$\vec{a} \cup \vec{b} = \vec{1}. \tag{23}$$

As an example, the choices $\vec{a}$ and $\vec{b}$ characterizing the global permutaion $Q^{\vec{a},\vec{b}}$ in Eq. (20) are complementary, and therefore $\vec{b}$ is fully determined by $\vec{a}$ – which is why we only need to sum over $\vec{a}$ in Eq. (21).

To illustrate what terms go into the sum over global permutations $Q^{\vec{a},\vec{b}}$ in Eq. (21), that is, in the sum over choices $\vec{a}$, we consider $M = 0, 1, 2, 3$ particles and show the possible choices $\vec{a} \in \mathcal{C}_{M_A}^M$ and their complementary $\vec{b}$ are, as a function of the number $M_A$ of particles in part $A$. For $M = 0$ particles, or vacuum state, we have that $\vec{1} = \emptyset$ and the only option is $M_A = 0$ with $\vec{a} = \vec{b} = \emptyset$. For a Bethe wavefunction with $M = 1$ particles, that is for a single-particle plane wave, we have

$$
\begin{aligned}
M_A = 0, \qquad &\vec{a} = \emptyset, \qquad &\vec{b} = \vec{1} = (1), \\
M_A = 1, \qquad &\vec{a} = \vec{1} = (1), \quad &\vec{b} = \emptyset.
\end{aligned}
\tag{24}
$$

For a Bethe wavefunction with $M = 2$ particles we find:

$$
\begin{aligned}
M_A = 0, \qquad &\vec{a} = \emptyset, \qquad &\vec{b} = \vec{1} = (1,2), \\
M_A = 1, \qquad &\vec{a} = (2), \qquad &\vec{b} = (1), \\
&\vec{a} = (1), \qquad &\vec{b} = (2), \\
M_A = 2, \qquad &\vec{a} = \vec{1} = (1,2), \quad &\vec{b} = \emptyset.
\end{aligned}
\tag{25}
$$

Finally, for $M = 3$ particles, we have:

$$
\begin{aligned}
M_A = 3, \qquad &\vec{a} = \emptyset, \qquad &\vec{b} = \vec{1} = (1,2,3), \\
M_A = 2, \qquad &\vec{a} = (1), \qquad &\vec{b} = (2,3), \\
&\vec{a} = (2), \qquad &\vec{b} = (1,3), \\
&\vec{a} = (3), \qquad &\vec{b} = (1,2), \\
M_A = 1, \qquad &\vec{a} = (1,2), \qquad &\vec{b} = (3), \\
&\vec{a} = (1,3), \qquad &\vec{b} = (2), \\
&\vec{a} = (2,3), \qquad &\vec{b} = (1), \\
M_A = 0, \qquad &\vec{a} = \vec{1} = (1,2,3), \ &\vec{b} = \emptyset.
\end{aligned}
\tag{26}
$$

### Factorization of amplitudes

For fixed values of $M_A$ and $M_B = M - M_A$, the quasi-momenta vector $\vec{k}$ can also be decomposed as a deirect sum of two quasi-momenta vectors $\vec{k}_A$ and $\vec{k}_B$,

$$
\vec{k} = \{\underbrace{k_1, \cdots, k_{M_A}}_{\equiv \vec{k}_A}, \underbrace{k_{M_A+1} \cdots, k_M}_{\equiv \vec{k}_B}\} = \vec{k}_A \oplus \vec{k}_B,
\tag{27}
$$

and the plane-wave amplitude $e^{i\vec{k}\cdot\vec{x}}$ factorizes as

$$
e^{i\vec{k}\cdot\vec{x}} = e^{i\vec{k}_A\cdot\vec{x}_A} e^{i\vec{k}_B\cdot\vec{x}_B}.
\tag{28}
$$

Similarly, under a permutation $P \in \mathcal{S}_M$, the permuted quasi-momenta vector $\vec{k}^P$ can be decomposed as

$$
\vec{k}^P = \{\underbrace{k_{P_1}, \cdots, k_{P_{M_A}}}_{\equiv (\vec{k}^P)_A}, \underbrace{k_{P_{M_A+1}} \cdots, k_{P_M}}_{\equiv (\vec{k}^P)_B}\} = (\vec{k}^P)_A \oplus (\vec{k}^P)_B.
\tag{29}
$$

Using the decomposition $P = R\,S\,Q^{\vec{a},\vec{b}}$ in Eq. (19), we first notice that $\vec{k}^P = (\vec{k}^{Q^{\vec{a},\vec{b}}})^{RS}$ (first apply the global permutation $Q^{\vec{a},\vec{b}}$ on $\vec{k}$, then apply the local permutations $R$ and $S$) and that we can write $\vec{k}^{Q^{\vec{a},\vec{b}}}$ as $\vec{k}^{\vec{a}} \oplus \vec{k}^{\vec{b}}$, that is, as the direct sum of reduced momenta vectors $\vec{k}^{\vec{a}}$ and $\vec{k}^{\vec{b}}$ by the complementary choices $\vec{a}$ and $\vec{b}$. Next, since $R$ and $S$ only permute the first $M_A$ and last $M_B$ components of the quasi-momenta vector $\vec{k}^{Q^{\vec{a},\vec{b}}}$, respectively, which are contained in $\vec{k}^{\vec{a}}$ and $\vec{k}^{\vec{b}}$, we can write

$$
\vec{k}^P = (\vec{k}^{\vec{a}})^R \oplus (\vec{k}^{\vec{b}})^S.
\tag{30}
$$

Accordingly, plane-wave amplitude $e^{i\vec{k}^P\cdot\vec{x}}$ factorizes as

$$
e^{i\vec{k}^P\cdot\vec{x}} = e^{i(\vec{k}^{\vec{a}})^R\cdot\vec{x}_A} e^{i(\vec{k}^{\vec{b}})^S\cdot\vec{x}_B}.
\tag{31}
$$

Finally, the amplitude $\Theta[P]$ can be seen to factorize as

$$
\Theta[P] = \Theta[Q^{\vec{a},\vec{b}}]\ \Theta^{\vec{a}}[R]\ \Theta^{\vec{b}}[S],
\tag{32}
$$

where we define $\Theta^{\vec{a}}[R]$ and $\Theta^{\vec{b}}[S]$ as the scattering amplitude assigned to local permutations $R$ and $S$ for the reduced data $(\vec{k}^{\vec{a}}, \boldsymbol{\theta}^{\vec{a}})$ in part $A$ and $(\vec{k}^{\vec{b}}, \boldsymbol{\theta}^{\vec{b}})$ in part $B$, respectively. For instance, in the example in Fig. 6 for $P = (4,3,2,1)$, where $Q = (3,4,1,2)$ (that is $\vec{a} = (3,4)$ and $\vec{b} = (1,2)$) and $R = (4,3)$ and $S = (2,1)$, the scattering amplitudes read

$$
\begin{aligned}
\Theta[P] &= (-e^{i\theta_{43}})(-e^{i\theta_{42}})(-e^{i\theta_{41}})(-e^{i\theta_{32}})(-e^{i\theta_{31}})(-e^{i\theta_{21}}), \\
\Theta[Q^{\vec{a},\vec{b}}] &= (-e^{i\theta_{42}})(-e^{i\theta_{41}})(-e^{i\theta_{32}})(-e^{i\theta_{31}}), \\
\Theta^{\vec{a}}[R] &= (-e^{i\theta_{43}}), \quad \Theta^{\vec{b}}[S] = (-e^{i\theta_{21}}),
\end{aligned}
\tag{33}
$$

which indeed agrees with Eq. (32).

### Local Bethe wavefunctions

Using the notation introduced above, we are ready to announce a key property of a Bethe wavefunction for $M$ particles with data $(\vec{k}, \boldsymbol{\theta})$. Given a left-right bipartition of the lattice into parts $A$ and $B$, with fixed particle numbers $M_A$ and $M_B$ and a choice vector $\vec{a}$ for the $M_A$ symbols in part $A$ (which uniquely determines the complementary choice vector $\vec{b}$ of $M_B$ symbols in part $B$), we can partially sum the contribution $\Theta[P]e^{i\vec{k}^P\cdot\vec{x}}|\vec{x}\rangle$ over all position vectors $\vec{x}$ and permutations $P$ compatible with the above constraints (namely the particle numbers in parts $A$ and $B$ must be $M_A$, $M_B$, and permutations must decompose as $P = R\,S\,Q^{\vec{a},\vec{b}}$ for fixed choice $\vec{a}$) to find a product of two *local* Bethe wavefunctions on parts $A$ and

$B$ with data $(\vec{k}^{\vec{a}}, \boldsymbol{\theta}^{\vec{a}})$ and $(\vec{k}^{\vec{b}}, \boldsymbol{\theta}^{\vec{b}})$, respectively:

$$\sum_{\vec{x}_A \in \mathcal{D}^A_{M_A}} \sum_{\vec{x}_B \in \mathcal{D}^B_{M_B}} \sum_{R \in \mathcal{S}_A} \sum_{S \in \mathcal{S}_B} \Theta[P] \, e^{i\vec{k}^P \cdot \vec{x}} \, |\vec{x}\rangle$$

$$= \Theta\left[Q^{\vec{a},\vec{b}}\right] \sum_{\vec{x}_A \in \mathcal{D}^A_{M_A}} \sum_{R \in \mathcal{S}_A} \Theta^{\vec{a}}[R] \, e^{i(\vec{k}^{\vec{a}})^R \cdot \vec{x}_A} \, |\vec{x}_A\rangle \times$$

$$\sum_{\vec{x}_B \in \mathcal{D}^B_{M_B}} \sum_{S \in \mathcal{S}_B} \Theta^{\vec{b}}[S] \, e^{i(\vec{k}^{\vec{b}})^S \cdot \vec{x}_B} \, |\vec{x}_B\rangle$$

$$= \Theta\left[Q^{\vec{a},\vec{b}}\right] \left|\Phi^{\vec{k}^{\vec{a}},\boldsymbol{\theta}^{\vec{a}}}_{M_A}\right\rangle_A \left|\Phi^{\vec{k}^{\vec{b}},\boldsymbol{\theta}^{\vec{b}}}_{M_B}\right\rangle_B, \qquad (34)$$

where the local Bethe wavefunction in part $A$ is

$$\left|\Phi^{\vec{k}^{\vec{a}},\boldsymbol{\theta}^{\vec{a}}}_{M_A}\right\rangle \equiv \sum_{\vec{x}_A \in \mathcal{D}^A_{M_A}} \sum_{R \in \mathcal{S}_A} \Theta^{\vec{a}}[R] \, e^{i(\vec{k}^{\vec{a}})^R \cdot \vec{x}_A} \, |\vec{x}_A\rangle \qquad (35)$$

and the local Bethe wavefunction in part $B$ is

$$\left|\Phi^{\vec{k}^{\vec{b}},\boldsymbol{\theta}^{\vec{b}}}_{M_B}\right\rangle \equiv \sum_{\vec{x}_B \in \mathcal{D}^B_{M_B}} \sum_{S \in \mathcal{S}_B} \Theta^{\vec{b}}[S] \, e^{i(\vec{k}^{\vec{b}})^S \cdot \vec{x}_B} \, |\vec{x}_B\rangle . \qquad (36)$$

Notice that these Bethe wavefunctions are consistent with Eq. (7) if we think of part $A$ (respectively, part $B$) of $\mathcal{L}$ as defining a 1d lattice on its own.

### C. Fractal bipartite decomposition

We can finally announce our main result.

**Theorem III.1 ((Fractal bipartite decomposition of a Bethe wavefunction).** *Given a bipartition of the 1d lattice $\mathcal{L}$ into left and right parts $A$ and $B$, the $M$-particle Bethe wavefunction $|\Phi_M\rangle$ in Eq. (7), with data $(\vec{k},\boldsymbol{\theta})$, can be decomposed into a sum of products of two local Bethe wavefunctions $\left|\Phi^{\vec{k}^{\vec{a}},\boldsymbol{\theta}^{\vec{a}}}_m\right\rangle$ and $\left|\Phi^{\vec{k}^{\vec{b}},\boldsymbol{\theta}^{\vec{b}}}_{m'}\right\rangle$ on parts $A$ and $B$, with data $(\vec{k}^{\vec{a}},\boldsymbol{\theta}^{\vec{a}})$ and $(\vec{k}^{\vec{b}},\boldsymbol{\theta}^{\vec{b}})$ as described in Eqs. (10)-(11). Concretely,*

*1. The Bethe wavefunction can be decomposed as,*

$$\left|\Phi^{\vec{k},\boldsymbol{\theta}}_M\right\rangle_{AB} = \sum_{m=0}^M \sum_{\vec{a} \in \mathcal{C}^M_m} \Theta\left[Q^{\vec{a},\vec{b}}\right] \left|\Phi^{\vec{k}^{\vec{a}},\boldsymbol{\theta}^{\vec{a}}}_m\right\rangle_A \left|\Phi^{\vec{k}^{\vec{b}},\boldsymbol{\theta}^{\vec{b}}}_{m'}\right\rangle_B, \quad (37)$$

*where the first sum is over particle number $m$ in part $A$ (with corresponding $m' = M - m$ particles in part $B$), the second sum is over the choice $\vec{a}$ of $m$ symbols on part $A$ (with complementary choice $\vec{a}$ of $m'$ symbols on part $A$) and the scattering phase $\Theta\left[Q^{\vec{a},\vec{b}}\right]$ is given in, and described after, Eq. (32).*

*2. The number of terms in the bipartite decomposition is at most $2^M$.*

*Proof.* To prove the theorem, in Eq. (7) we simply split the sum $\sum_{\vec{x}}$ into $\sum_m \sum_{\vec{x}_A} \sum_{\vec{x}_B}$ according to Eq. (15) and the sum $\sum_P$ over permutations into $\sum_{\vec{b}} \sum_R \sum_S$ according to Eq. (21), while at the same time using the factorization of amplitudes $e^{i\vec{k}^P \cdot \vec{x}}$ and $\Theta[P]$ in Eqs. (31) and (32), namely $e^{i\vec{k}^P \cdot \vec{x}} = e^{i(\vec{k}^{\vec{a}})^R \cdot \vec{x}_A} e^{i(\vec{k}^{\vec{b}})^S \cdot \vec{x}_B}$ and $\Theta[P] = \Theta[Q^{\vec{a},\vec{b}}] \, \Theta^{\vec{a}}[R] \, \Theta^{\vec{b}}[S]$, which yields

$$\left|\Phi^{\vec{k},\boldsymbol{\theta}}_M\right\rangle_{AB} = \sum_{\vec{x} \in \mathcal{D}^{\mathcal{L}}_M} \sum_{P \in \mathcal{S}_M} \Theta[P] \, e^{i\vec{k}^P \cdot \vec{x}} \, |\vec{x}\rangle$$

$$= \sum_{m=0}^M \sum_{\vec{a} \in \mathcal{C}^M_m} \Theta[Q^{\vec{a},\vec{b}}] \quad \times$$

$$\left( \sum_{\vec{x}_A \in \mathcal{D}^A_{M_A}} \sum_{R \in \mathcal{S}_A} \Theta^{\vec{a}}[R] \, e^{i(\vec{k}^{\vec{a}})^R \cdot \vec{x}_A} \, |\vec{x}_A\rangle \right) \times$$

$$\left( \sum_{\vec{x}_B \in \mathcal{D}^B_{M_B}} \sum_{S \in \mathcal{S}_B} \Theta^{\vec{b}}[S] \, e^{i(\vec{k}^{\vec{b}})^S \cdot \vec{x}_B} \, |\vec{x}_B\rangle \right)$$

$$= \sum_{m=0}^M \sum_{\vec{b} \in \mathcal{C}^M_m} \Theta\left[Q^{\vec{a},\vec{b}}\right] \left|\Phi^{\vec{k}^{\vec{a}},\boldsymbol{\theta}^{\vec{a}}}_m\right\rangle_A \left|\Phi^{\vec{k}^{\vec{a}},\boldsymbol{\theta}^{\vec{b}}}_{m'}\right\rangle_B, \quad (38)$$

where in the last line we used the expressions (35) and (36) defining the local Bethe wavefunctions.

Recall that the cardinality $|\mathcal{C}^M_m| = \binom{M}{m}$ of $\mathcal{C}^M_m$ corresponds to the number of possible choices of $m$ symbols out of $M$. Adding contributions for different values of $m$, we find that decomposition (37) has $\sum_{m=0}^M \binom{M}{m} = 2^M$ product terms.

Above we assumed that each part $A$ and $B$ could host up to $M$ particles, that is, $N_A, N_B \geq M$. We note that if the size of the parts $A$ and $B$ are such that one of them can't host all $M$ particles, then the sum over particle number $m$ on part $B$ runs from $m_{\min} \equiv \max(0, M - N_B)$ to $m_{\max} \equiv \min(M, N_A)$ in Eq. (37), and subsequently the number of terms in the decomposition will be lower than $2^M$. $\qquad \square$

This theorem provides an upper bound on the rank $\chi$ of the Schmidt decomposition according to the left-right bipartition (since the Schmidt decomposition has the minimum possible number of terms over all possible bipartite decompositions as a sum of product states).

**Corollary III.1.** *The Schmidt rank $\chi$ for a left-right bipartite decomposition of an $M$-particle Bethe state on a lattice $\mathcal{L}$ made of $N$ sites is upper-bounded by $2^M$,*

$$\chi \leq 2^M. \qquad (39)$$

*This upper bound is independent of the number $N$ of sites in lattice $\mathcal{L}$ or the number $N_A$ and $N_B$ of sites in parts $A$ and $B$ of the bipartition.*

If the local Bethe wavefunctions in decomposition (37) are all linearly independent, then the Schmidt rank saturates the upper bound, that is $\chi = 2^M$. We emphasize once more that this constant-in-$N$ (upper bound for the) Schmidt rank is a remarkable property of Bethe wavefunctions, in sharp contrast with the Schmidt rank of a generic M-particle wavefunction, which scales with $N$ as $N^M$, see Eq. (18).

*Examples*

Let us discuss some simple examples of the left-right bipartite decomposition (37). We start with the trivial cases of $M = 0, 1$ particles before considering $M = 2, 3$.

For $M = 0$ particles, we have the vacuum state $|\Phi_{M=0}\rangle = |\emptyset\rangle$ representing $N$ empty sites. This is an unentangled state between parts $A$ and $B$, namely the product of two local vacuum states

$$|\Phi_{M=0}\rangle_{AB} = |\emptyset\rangle_A |\emptyset\rangle_B = |\Phi_{m=0}\rangle_A |\Phi_{m'=0}\rangle_B . \quad (40)$$

For $M = 1$, the one-particle plane wave is decomposed into two product terms, corresponding to having $m = 0$ particles or $m = 1$ particles in part $A$, namely

$$\left|\Phi_{M=1}^{(k)}\right\rangle_{AB} = \sum_{x=1}^{N} e^{ikx} |x\rangle$$

$$= |\emptyset\rangle_A \left(\sum_{x\in B} e^{ikx} |x\rangle\right) + \left(\sum_{x\in A} e^{ikx} |x\rangle\right) |\emptyset\rangle_B$$

$$= |\Phi_{m=0}\rangle_A \left|\Phi_{m'=1}^{(k)}\right\rangle_B + \left|\Phi_{m=1}^{(k)}\right\rangle_A |\Phi_{m'=0}\rangle_B . \quad (41)$$

Notice that each term consists of the product of a local vacuum state and a local plane wave state. For $M = 2$, direct re-organization of Eq. (5) produces the following linear combination of terms, with each term being indeed the product of two local Bethe states,

$$\sum_{(x_1,x_2)\in\mathcal{D}_2^{\mathcal{L}}} \left(e^{ik_1 x_1} e^{ik_2 x_2} - e^{i\theta_{21}} e^{ik_2 x_1} e^{ik_1 x_2}\right) |x_1 x_2\rangle$$

$$= |\emptyset\rangle_A \left(\sum_{\substack{(x_1,x_2)\\ \in\mathcal{D}_2^{B}}} \left(e^{ik_1 x_1} e^{ik_2 x_2} - e^{i\theta_{21}} e^{ik_2 x_1} e^{ik_1 x_2}\right) |x_1 x_2\rangle\right)$$

$$+ \left(\sum_{x_1\in A} e^{ik_1 x_1} |x_1\rangle\right)\left(\sum_{x_2\in B} e^{ik_2 x_2} |x_2\rangle\right)$$

$$- e^{i\theta_{21}} \left(\sum_{x_1\in A} e^{ik_2 x_1} |x_1\rangle\right)\left(\sum_{x_2\in B} e^{ik_1 x_2} |x_2\rangle\right)$$

$$+ \left(\sum_{\substack{(x_1,x_2)\\ \in\mathcal{D}_2^{A}}} \left(e^{ik_1 x_1} e^{ik_2 x_2} - e^{i\theta_{21}} e^{ik_2 x_1} e^{ik_1 x_2}\right) |x_1 x_2\rangle\right) |\emptyset\rangle_B$$

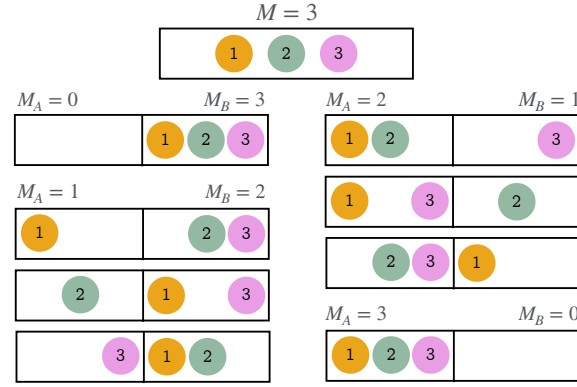

FIG. 7. The terms for the $M = 3$ Bethe state left-right decomposition according to Thm. III.1, which represents the organization of terms in Eq. (43).

or, equivalently,

$$\left|\Phi_{M=2}^{(k_1,k_2),\{\theta_{21}\}}\right\rangle_{AB} = |\Phi_{m=0}\rangle_A \left|\Phi_{m'=2}^{(k_1,k_2),\{\theta_{21}\}}\right\rangle_B$$

$$\left.\begin{array}{l} \left|\Phi_{m=1}^{(k_1)}\right\rangle_A \left|\Phi_{m'=1}^{(k_2)}\right\rangle_B \\ -e^{i\theta_{21}} \left|\Phi_{m=1}^{(k_2)}\right\rangle_A \left|\Phi_{m'=1}^{(k_1)}\right\rangle_B \end{array}\right\} m = 1$$

$$+ \left|\Phi_{m=2}^{(k_1,k_2),\{\theta_{21}\}}\right\rangle_A |\Phi_{m'=0}\rangle_B , \quad (42)$$

where we have organized the 4 product terms in the decomposition according to growing number $m = 0, 1, 2$ of particles in part $A$, which contributed 1,1,1 product terms to the decomposition, respectively.

For $M = 3$, a tedious but otherwise straightforward re-organization of terms in Eq. (9) according to Fig. 7 leads to

$$\left|\Phi_{M=3}^{(k1,k2,k3),\{\theta_{21},\theta_{32},\theta_{31}\}}\right\rangle_{AB}$$

$$= |\Phi_{m=0}\rangle_A \left|\Phi_{m'=3}^{(k1,k2,k3),\{\theta_{21},\theta_{32},\theta_{31}\}}\right\rangle_B$$

$$\left.\begin{array}{l} + \left|\Phi_{m=1}^{(k_1)}\right\rangle_A \left|\Phi_{m'=2}^{(k_2,k_3),\{\theta_{32}\}}\right\rangle_B \\ -e^{i\theta_{21}} \left|\Phi_{m=1}^{(k_2)}\right\rangle_A \left|\Phi_{m'=2}^{(k_1,k_3),\{\theta_{31}\}}\right\rangle_B \\ +e^{i\theta_{32}}e^{i\theta_{31}} \left|\Phi_{m=1}^{(k_3)}\right\rangle_A \left|\Phi_{m'=2}^{(k_1,k_2),\{\theta_{21}\}}\right\rangle_B \end{array}\right\} m = 1$$

$$\left.\begin{array}{l} + \left|\Phi_{m=2}^{(k_1,k_2),\{\theta_{21}\}}\right\rangle_A \left|\Phi_{m'=1}^{(k_3)}\right\rangle_B \\ -e^{i\theta_{32}} \left|\Phi_{m=2}^{(k_1,k_3),\{\theta_{21}\}}\right\rangle_A \left|\Phi_{m'=1}^{(k_2)}\right\rangle_B \\ +e^{i\theta_{21}}e^{i\theta_{31}} \left|\Phi_{m=2}^{(k_2,k_3),\{\theta_{21}\}}\right\rangle_A \left|\Phi_{m'=1}^{(k_1)}\right\rangle_B \end{array}\right\} m = 2$$

$$+ \left|\Phi_{m=3}^{(k1,k2,k3),\{\theta_{21},\theta_{32},\theta_{31}\}}\right\rangle_A |\Phi_{m'=0}\rangle_B , \quad (43)$$

where we have again organized the 8 product terms in the decomposition according to growing number $m = 0, 1, 2, 3$ of particle in part $A$, which contributed 1,3,3,1 product terms to the decomposition, respectively.

## IV. FRACTAL MULTIPARTITE DECOMPOSITION

In this section we generalize the bipartite decomposition (37) of a Bethe wavefunction to the multipartite case, in which the 1d lattice $\mathcal{L}$ is divided into more than just two parts $A$ and $B$. We consider the $L-$partite case with an arbitrary number $L$ of parts $\{A_i\} \equiv \{A_1, A_2, \cdots, A_L\}$, such that each partition is made of contiguous sites and ordered such that $A_i$ is to the left of $A_{i+1}$. We call this partition $\{A_i\}$ a left-right $L-$partition (see Fig. 5).

We will see that it is again possible to decompose a Bethe wavefunction for $M$ particles as a sum of products of local Bethe wavefunctions on the $L$ parts, with a total of $L^M$ product terms, and the result follows from repeated applications of the bipartite decomposition.

### A. Revamped notation for bipartite decomposition

In order to proceed, it is important to both simplify the current notation and introduce new one. We start by rewriting the bipartite decomposition (37) of Theorem III.1 in the previous section.

Recall that the Bethe data $(\vec{k}, \boldsymbol{\theta})$, identified with the unity choice $\vec{1} = (1, 2, \cdots, M)$, characterizes a global Bethe wavefunction $\left|\Phi_M^{\vec{k},\boldsymbol{\theta}}\right\rangle$, whereas all possible reduced data $(\vec{k}^{\vec{c}}, \boldsymbol{\theta}^{\vec{c}})$, identified with choice vectors $\vec{c}$ containing an increasingly ordered subset of the components of $\vec{1}$, characterize all possible local Bethe wavefunctions $\left|\Phi_m^{\vec{k}^{\vec{c}},\boldsymbol{\theta}^{\vec{c}}}\right\rangle$ that appear in the bipartite decomposition (37). It is then clear that the choice $\vec{c}$ contains enough information to identify the corresponding local Bethe wavefunction $\left|\Phi_m^{\vec{k}^{\vec{c}},\boldsymbol{\theta}^{\vec{c}}}\right\rangle$ (with the particle number $m$ being the number of components in choice $\vec{c} = (c_1, c_2, \cdots, c_m)$, that is $m = |\vec{c}|$). We can therefore denote the above local Bethe wavefunction simply as $|\vec{c}\rangle$, and make the replacement

$$\left|\Phi_m^{\vec{k}^{\vec{a}},\boldsymbol{\theta}^{\vec{a}}}\right\rangle_A \left|\Phi_{m'}^{\vec{k}^{\vec{b}},\boldsymbol{\theta}^{\vec{b}}}\right\rangle_B \longrightarrow \left|\vec{a}, \vec{b}\right\rangle. \tag{44}$$

We say that a choice $\vec{c}$ is contained in another choice $\vec{d}$, denoted $\vec{c} \le \vec{d}$, if all the components of $\vec{c}$ are also components of $\vec{d}$. Notice that by construction, $\vec{c} \le \vec{1}$. We then say that the local Bethe wavefunctions are characterized by choice vectors $\vec{c}$ such that $\emptyset \le \vec{c} \le \vec{1}$, and replace the sums over particle number $m$ and choices having that particle number with a single sum over choices,

$$\sum_{m=0}^{M} \sum_{\vec{a} \in \mathcal{C}_m^M} \longrightarrow \sum_{\vec{a}=\emptyset}^{\vec{1}} \longrightarrow \sum_{\vec{a}}, \tag{45}$$

where in the last step we further simplified the above notation by not indicating the summation domain, which is

assumed. We can now also make explicit the dependence between the choices $\vec{a}$ and $\vec{b}$ for the local wavefunctions on parts $A$ and $B$, namely $\vec{a} \cup \vec{b} = \vec{1}$, which we can impose through the delta function, $\delta_{\vec{a} \cup \vec{b}, \vec{1}}$, so that

$$\sum_{\vec{a}} \to \sum_{\vec{a}, \vec{b}} \delta_{\vec{a} \cup \vec{b}, \vec{1}}. \tag{46}$$

We also simplify the notation regarding the scattering amplitude

$$\Theta[Q^{\vec{a},\vec{b}}] \to \Theta\left[\vec{a}, \vec{b}\right] \tag{47}$$

for which we rewrite the definition here,

$$\Theta[\vec{a}, \vec{b}] = \prod_{\alpha=1}^{m} \prod_{\beta=1}^{m'} f_{\vec{a},\vec{b}}(\alpha, \beta)$$

$$f_{\vec{a},\vec{b}}(\alpha, \beta) = \begin{cases} 1, & \text{if } a_\alpha < b_\beta, \\ -e^{i\theta_{a_\alpha b_\beta}} & \text{if } a_\alpha > b_\beta, \end{cases} \tag{48}$$

and remind the reader that it is a product of scattering phases $-e^{i\theta_{a_\alpha b_\beta}}$ for each pair of symbols $(a_\alpha, b_\beta)$ in $\vec{a}$ and $\vec{b}$ such that $a_\alpha > b_\beta$, implying that the order of these two symbols in $\vec{a} \oplus \vec{b}$ is the opposite to their original order (namely $a_\alpha < b_\beta$) in $\vec{1}$. Putting all these changes together, the bipartite decomposition (37) reads

$$|\vec{1}\rangle = \sum_{\vec{a}, \vec{b}} \delta_{\vec{a} \cup \vec{b}, \vec{1}} \, \Theta\left[\vec{a}, \vec{b}\right] \left|\vec{a}, \vec{b}\right\rangle. \tag{49}$$

A crucial observation is that such an analogous bipartite decomposition also applies when we decompose a local Bethe wavefunction $|\vec{\mu}\rangle$ defined by a choice vector $\vec{\mu} \le \vec{1}$, as long as the decomposition involves choices $\vec{a}$ and $\vec{b}$ that are consistent with the delta function constraint $\delta_{\vec{a} \cup \vec{b}, \vec{\mu}}$, in which case we write

$$|\vec{\mu}\rangle = \sum_{\vec{a}, \vec{b}} \delta_{\vec{a} \cup \vec{b}, \vec{\mu}} \, \Theta\left[\vec{a}, \vec{b}\right] \left|\vec{a}, \vec{b}\right\rangle. \tag{50}$$

Moreover, anticipating the multipartite case, where the use of a diagrammatic notation helps significantly in guiding the discussion, we represent the delta function $\delta_{\vec{a} \cup \vec{b}, \vec{\mu}}$ constraining choices $\vec{a}$ and $\vec{b}$ so that their union $\vec{a} \cup \vec{b}$ equates $\vec{\mu}$, as

$$\delta_{\vec{a} \cup \vec{b}, \vec{\mu}} \equiv \quad \vec{\mu} \rightarrow \bullet \quad \vec{a} \quad \vec{b} \tag{51}$$

where outgoing arrows are used for the choices $\vec{a}$ and $\vec{b}$ and an incoming arrow is used for $\vec{\mu}$.

Using this notation we can succinctly write the bipartite decomposition (37) of a Bethe wavefunction as,

$$|\vec{1}\rangle = \sum_{\vec{a},\vec{b}} \quad \includegraphics \quad |\vec{a},\vec{b}\rangle \tag{52}$$

where the scattering amplitude $\Theta[\vec{a},\vec{b}]$ decorates the delta function.

## B. Fractal tripartite decomposition

In order to prepare for the derivation of the multipartite decomposition for an arbitrary number $L$ of parts below, as well as to introduce the key steps in a simplified context, we first describe how to extend the bipartite decomposition of $\left|\vec{1}\right\rangle$ for a left-right bipartition $\mathcal{L} = A \cup A'$ of the 1d lattice $\mathcal{L}$ into parts $A$ and $A'$ to a tripartite decomposition of $\left|\vec{1}\right\rangle$ for a left-right tripartition $\mathcal{L} = A \cup B \cup C$ of the same 1d lattice, where the region $A'$ has further been left-right partitioned into parts $B$ and $C$, that is $A' = B \cup C$. Note that indeed this tripartition is left-right; $A$ is to the left of $B$, which is to the left of $C$.

We start from the bipartite decomposition in (49) for parts $A$ and $A'$, which we rewrite as

$$|\vec{1}\rangle = \sum_{\vec{a},\vec{\mu}} \delta_{\vec{a}\cup\vec{\mu},\vec{1}} \, \Theta[\vec{a},\vec{\mu}] \, |\vec{a},\vec{\mu}\rangle \,, \tag{53}$$

where $|\vec{a}\rangle$ and $|\vec{\mu}\rangle$ are local Bethe wavefunctions on parts $A$ and $A'$, and proceed to further decompose the local Bethe wavefunction $|\vec{\mu}\rangle$ on $A'$ into local Bethe wavefunctions $\left|\vec{b}\right\rangle$ and $|\vec{c}\rangle$ on parts $B$ and $C$.

$$|\vec{\mu}\rangle = \sum_{\vec{b},\vec{c}} \delta_{\vec{b}\cup\vec{c},\vec{\mu}} \, \Theta[\vec{b},\vec{c}] \, \left|\vec{b},\vec{c}\right\rangle \,. \tag{54}$$

Combining these expressions we obtain

$$|\vec{1}\rangle = \sum_{\vec{a},\vec{\mu}} \delta_{\vec{a}\cup\vec{\mu},\vec{1}} \, \Theta[\vec{a},\vec{\mu}] \sum_{\vec{b},\vec{c}} \delta_{\vec{b}\cup\vec{c},\vec{\mu}} \, \Theta[\vec{b},\vec{c}] \left|\vec{a},\vec{b},\vec{c}\right\rangle, \tag{55}$$

which is a linear combination of products $\left|\vec{a},\vec{b},\vec{c}\right\rangle$ of local Bethe wavefunctions. In our final tripartite decomposition, we re-express the global Bethe wavefunction as

$$|\vec{1}\rangle = \sum_{\vec{a},\vec{b},\vec{c}} \delta_{\vec{a}\cup\vec{b}\cup\vec{c},\vec{1}} \, \Theta_3[\vec{a},\vec{b},\vec{c}] \, \left|\vec{a},\vec{b},\vec{c}\right\rangle, \tag{56}$$

namely in terms of (i) a tripartite delta $\delta_{\vec{a}\cup\vec{b}\cup\vec{c},\vec{1}}$ that ensures that the product of the three local Bethe wavefunctions $\left|\vec{a},\vec{b},\vec{c}\right\rangle$ contains all the $M$ symbols in $\vec{1}$ without

repeating any, and (ii) a tripartite scattering amplitude $\Theta_3[\vec{a},\vec{b},\vec{c}]$, which will next be seen to factorize as a product of bipartite scattering amplitudes. Indeed, comparing Eqs. (55) and (56) we have,

$$\delta_{\vec{a}\cup\vec{b}\cup\vec{c},\vec{1}} \, \Theta_3[\vec{a},\vec{b},\vec{c}] = \left(\sum_{\vec{\mu}} \delta_{\vec{a}\cup\vec{\mu},\vec{1}} \, \Theta[\vec{a},\vec{\mu}] \, \delta_{\vec{\mu},\vec{b}\cup\vec{c}}\right) \Theta[\vec{b},\vec{c}]$$
$$= \delta_{\vec{a}\cup\vec{b}\cup\vec{c},\vec{1}} \, \Theta[\vec{a},\vec{b}\cup\vec{c}] \, \Theta[\vec{b},\vec{c}]. \tag{57}$$

Then, using the important bipartite property

$$\Theta[\vec{a},\vec{b}\cup\vec{c}] = \Theta[\vec{a},\vec{b}]\Theta[\vec{a},\vec{c}], \tag{58}$$

which follows from the definition of $\Theta$ in (48) and states that when a choice $\vec{\mu}$ can be expressed as the union of two choices, $\vec{\mu} = \vec{b} \cup \vec{c}$, then the scattering amplitude $\Theta[\vec{a},\vec{\mu}]$ factorizes as the product of two scattering amplitudes $\Theta[\vec{a},\vec{b}] \, \Theta[\vec{a},\vec{c}]$, we arrive at

$$\Theta_3[\vec{a},\vec{b},\vec{c}] = \Theta[\vec{a},\vec{b}] \, \Theta[\vec{a},\vec{c}] \, \Theta[\vec{b},\vec{c}], \tag{59}$$

(for eligible choices $\vec{a},\vec{b},\vec{c}$, that is, for choices fulfilling $\vec{a} \cup \vec{b} \cup \vec{c} = \vec{1}$) which expresses the tripartite scattering amplitude $\Theta_3[\vec{a},\vec{b},\vec{c}]$ as a product of three bipartite scattering amplitudes.

In summary, we have obtained our tripartite decomposition (56), which we can diagramatically represent as,

$$|\vec{1}\rangle = \sum_{\vec{a},\vec{b},\vec{c}} \quad \includegraphics \quad |\vec{a},\vec{b},\vec{c}\rangle \tag{60}$$

by combining the bipartite decompositions (53) and (54), a process that we can diagramatically represent as

$$\includegraphics \tag{61}$$

which is equivalent to Eq. (57). Here we use the convention, standard in the tensor network diagrammatic notation, that there is a sum over the bond index $\vec{\mu}$ connectic two nodes.

Let us analyze the number of independent terms in the tripartite decomposition above. Assume there are totally $M$ particles, and each subregion $A, B, C$ is such that it can host $M$ particles itself (i.e. $N_A, N_B, N_C \geq M$). Since the number of choices for $M$ symbols is $2^M$, naively it would seem that the number of terms can be as large as $\left(2^M\right)^3 = 2^{3M} = 8^M$. However, the tripartite delta function $\delta_{\vec{a}\cup\vec{b}\cup\vec{c},\vec{1}}$ imposes the global constraint $\vec{a} \cup \vec{b} \cup \vec{c} = \vec{1}$, which reduces the number of allowed terms. More

concretely, consider an allocation of $M_A$, $M_B$ and $M_C$ particles on $A, B$ and $C$ respectively. Due to constraint that $M_A + M_B + M_C = M$, the number of independent choice vectors $\vec{a}, \vec{b}, \vec{c}$ is $\frac{M!}{M_A! M_B! M_C!}$. Therefore, if each of the parts can host up to $M$ particles, then there are $\sum_{M_A, M_B, M_C} \frac{M!}{M_A! M_B! M_C!} = 3^M$ (where $M_A + M_B + M_C = M$) terms in the tripartite decomposition, a number that is significantly lower than $8^M$.

### C. Fractal multipartite decomposition

We can extend the above discussion to a left-right partition made of any number $L$ of parts $\{A_l\}$ so as to decompose a Bethe wavefunction into a sum of tensor products $|\vec{a}_1, \vec{a}_2, \cdots \vec{a}_L\rangle$ of local Bethe wavefunctions $\{|\vec{a}_l\rangle\}$ supported on each part. Such decomposition can be diagrammatically represented as,

$$|\vec{1}\rangle = \sum_{\vec{a}_1, \cdots, \vec{a}_L} \vec{1} \xrightarrow{\Theta_L[\vec{a}_1, \cdots, \vec{a}_L]} |\vec{a}_1, \cdots, \vec{a}_L\rangle \quad , \quad (62)$$

in terms of a delta function $\delta_{\vec{a}_1 \cup \vec{a}_2 \cup \cdots \cup \vec{a}_L, \vec{1}}$ establishing that all the symbols in $\vec{1}$ appear once and just once in the product $|\vec{a}_1, \vec{a}_2, \cdots \vec{a}_L\rangle$, and an appropriately defined $L$-partite amplitude $\Theta_L[\vec{a}_1, \vec{a}_2, \cdots, \vec{a}_L]$. We state the result in the form of a theorem,

**Theorem IV.1 (Fractal multipartite decomposition of a Bethe wavefunction).** *Given a left-right partition of the 1d lattice $\mathcal{L}$ into $L$ parts $\{A_l\}$, $l = 1, \cdots, L$, an $M$-particle Bethe wavefunction $\left|\vec{1}\right\rangle$ on $\mathcal{L}$ can be expressed as a sum of tensor products $|\vec{a}_1, \cdots, \vec{a}_L\rangle$ of local Bethe wavefunctions $|\vec{a}_l\rangle$ supported on each part. More concretely,*

*1. The multipartite decomposition is,*

$$|\vec{1}\rangle = \sum_{\vec{a}_1, \cdots, \vec{a}_L} \delta_{\vec{a}_1 \cup \cdots \cup \vec{a}_L, \vec{1}} \, \Theta_L[\vec{a}_1, \cdots, \vec{a}_L] \, |\vec{a}_1, \cdots, \vec{a}_L\rangle . \quad (63)$$

*where the amplitude $\Theta_L[\vec{a}_1, \cdots, \vec{a}_L]$ is given by,*

$$\Theta_L[\vec{a}_1, \vec{a}_2, \cdots, \vec{a}_L] = \prod_{l_1=1}^{L-1} \prod_{l_2=l_1+1}^{L} \Theta[\vec{a}_{l_1}, \vec{a}_{l_2}]. \quad (64)$$

*2. The number of terms in the multipartite decomposition is at most $L^M$, which is independent of the number $N$ of sites in the lattice $\mathcal{L}$ or the number $\{N_l\}$ of sites in each of its parts $\{A_l\}$.*

*Proof.* We prove this by induction, by constructing a $(p+1)$-partite decomposition from a $p$-partite decomposition. [For pedagogical reasons, the $p = 2$ case (constructing the tripartite decomposition (56) from the bipartite

decomposition (49)) was already demonstrated prior to stating the theorem. Notice that the postulated form of the multipartite amplitude $\Theta_L$ in Eq. Eq. (64) is consistent with the tripartite amplitude $\Theta_3$ in Eq. (59).]

Consider a $p$-partite decomposition of a Bethe wavefunction $\left|\vec{1}\right\rangle$ according to a left-right partition of the lattice $\mathcal{L} = A_1 \cup \cdots \cup A_{p-1} \cup M_{p-1}$ into parts $A_1, \cdots, A_{p-1}, M_{p-1}$, namely

$$|\vec{1}\rangle = \sum_{\vec{a}_1, \cdots, \vec{a}_{p-1}, \vec{\mu}_{p-1}} \delta_{\vec{a}_1 \cup \cdots \cup \vec{a}_{p-1} \cup \vec{\mu}_{p-1}, \vec{1}}$$
$$\Theta_p[\vec{a}_1, \cdots, \vec{a}_{p-1}, \vec{\mu}_{p-1}] \, |\vec{a}_1, \cdots, \vec{a}_{p-1}, \vec{\mu}_{p-1}\rangle , \quad (65)$$

where the $p$-partite scattering amplitude $\Theta_p$ is given by Eq. (64). Consider now dividing part $M_{p-1}$ further into two parts $A_p$ and $M_p$, and the bipartite decomposition of the local bethe wavefunction $|\vec{\mu}_{p-1}\rangle$,

$$|\vec{\mu}_{p-1}\rangle = \sum_{\vec{a}_p, \vec{\mu}_p} \delta_{\vec{a}_p \cup \vec{\mu}_p, \vec{\mu}_{p-1}} \, \Theta[\vec{a}_p, \vec{\mu}_p] \, |\vec{a}_p, \vec{\mu}_p\rangle . \quad (66)$$

Replacing Eq. (66) in Eq. (65) we obtain,

$$|\vec{1}\rangle = \sum_{\vec{a}_1, \cdots, \vec{a}_{p-1}, \vec{\mu}_{p-1}} \delta_{\vec{a}_1 \cup \cdots \cup \vec{a}_{p-1} \cup \vec{\mu}_{p-1}, \vec{1}} \, \Theta_p[\vec{a}_1, \cdots, \vec{a}_{p-1}, \vec{\mu}_{p-1}]$$
$$\sum_{\vec{a}_p, \vec{\mu}_p} \delta_{\vec{a}_p \cup \vec{\mu}_p, \vec{\mu}_{p-1}} \, \Theta[\vec{a}_p, \vec{\mu}_p] \, |\vec{a}_1, \cdots, \vec{a}_{p-1}, \vec{a}_p, \vec{\mu}_p\rangle , \quad (67)$$

which is already a decomposition in terms of products $|\vec{a}_1, \cdots, \vec{a}_{p-1}, \vec{a}_p, \vec{\mu}_p\rangle$ of $p + 1$ local Bethe wavefunctions. To prove the theorem, we would like to re-express the above decomposition as

$$|\vec{1}\rangle = \sum_{\vec{a}_1, \cdots, \vec{a}_p, \vec{\mu}_p} \delta_{\vec{a}_1 \cup \cdots \cup \vec{a}_{p-1} \cup \vec{a}_p \cup \vec{\mu}_p, \vec{1}} \quad (68)$$
$$\Theta_{p+1}[\vec{a}_1, \cdots, \vec{a}_{p-1}, \vec{a}_p, \vec{\mu}_p] \, |\vec{a}_1, \cdots, \vec{a}_{p-1}, \vec{a}_p, \vec{\mu}_p\rangle ,$$

in terms of the product of a global delta function and $(p + 1)$-partite scattering amplitudes $\Theta_{p+1}$ that in turn decompose as the product of bipartite scattering amplitudes in Eq. (64). Comparing Eqs. (67) and (68), we find

$$\delta_{\vec{a}_1 \cup \cdots \cup \vec{a}_p \cup \vec{\mu}_p, \vec{1}} \, \Theta_{p+1}[\vec{a}_1, \cdots, \vec{a}_p, \vec{\mu}_p]$$
$$= \left( \sum_{\vec{\mu}_{p-1}} \delta_{\vec{a}_p \cup \vec{\mu}_p, \vec{\mu}_{p-1}} \, \Theta_p[\vec{a}_1, \cdots, \vec{a}_{p-1}, \vec{\mu}_{p-1}] \right.$$
$$\left. \times \, \delta_{\vec{a}_p \cup \vec{\mu}_p, \vec{\mu}_{p-1}} \right) \Theta[\vec{a}_p, \vec{\mu}_p]$$
$$= \delta_{\vec{a}_1 \cup \cdots \cup \vec{a}_p \cup \vec{\mu}_p, \vec{1}} \, \Theta[\vec{a}_1, \cdots, \vec{a}_{p-1}, \vec{a}_p \cup \vec{\mu}_p] \, \Theta[\vec{a}_p, \vec{\mu}_p]. (69)$$

In our last step, we explicitly expand

$$\Theta[\vec{a}_1, \cdots, \vec{a}_{p-1}, \vec{a}_p \cup \vec{\mu}_p]\, \Theta[\vec{a}_p, \vec{\mu}_p]$$

$$= \left(\prod_{i=1}^{p-2}\prod_{j=i+1}^{p-1}\Theta[\vec{a}_i,\vec{a}_j]\right)\left(\prod_{k=1}^{p-1}\Theta[\vec{a}_k,\vec{a}_p\cup\vec{\mu}_p]\right)\Theta[\vec{a}_p,\vec{\mu}_p]$$

$$= \left(\prod_{i=1}^{p-2}\prod_{j=i+1}^{p-1}\Theta[\vec{a}_i,\vec{a}_j]\right)\left(\prod_{k=1}^{p-1}\Theta[\vec{a}_k,\vec{a}_p]\,\Theta[\vec{a}_k,\vec{\mu}_p]\right)\Theta[\vec{a}_p,\vec{\mu}_p]$$

$$= \left(\prod_{i=1}^{p-1}\prod_{j=i+1}^{p}\Theta[\vec{a}_i,\vec{a}_j]\right)\prod_{k=1}^{p}\Theta[\vec{a}_k,\vec{\mu}_p], \qquad (70)$$

where we used Eq. (58) to make the replacement $\Theta[\vec{a}_k, \vec{a}_p \cup \vec{\mu}_p] = \Theta[\vec{a}_k, \vec{a}_p]\,\Theta[\vec{a}_k, \vec{\mu}_p]$. Notice that expression (70) is equivalent to Eq. (64) if we replace $\vec{\mu}_p$ with $\vec{a}_L$, then set $p = L - 1$, and re-organize the resulting products $\left(\prod_{i=1}^{L-2}\prod_{j=i+1}^{L-1}\Theta[\vec{a}_i,\vec{a}_j]\right)\prod_{k=1}^{L-1}\Theta[\vec{a}_k,\vec{a}_L]$ into $\prod_{i=1}^{L-1}\prod_{j=i+1}^{L}\Theta[\vec{a}_i,\vec{a}_j]$.

The crucial observation of the relation between the amplitudes $\Theta_p$ and $\Theta_{p+1}$ is encapsulated in the following diagram,

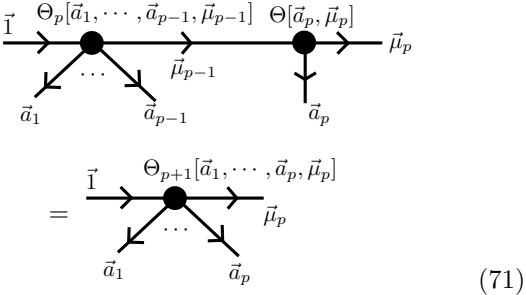

$$(71)$$

which is equivalent to Eq. (69) and is the $L$-partite generalization of the Eq. (61).

Now we analyze the number of terms in the decomposition, Eq. 63. As for the tripartite case, the multipartite decomposition imposes the global constraint $\cup_l \vec{a}_l = \vec{1}$. Given the allocation of $M_l$ symbols on part $A_l$, there are $\frac{M!}{M_1! M_2! \cdots M_k!}$ independent sets of $l$ choices $\vec{a}_1, \vec{a}_2, \cdots, \vec{a}_L$ such that $\cup_l \vec{a}_l = \vec{1}$. Therefore, if each of the parts can host up to $M$ particles, meaning the number of sites $N_l$ is at least $M$, then there are $\sum_{M_1 \cdots M_k} \frac{M!}{M_1! M_2! \cdots M_L!} = L^M$ (where $\sum_i M_i = M$). If some parts are not capable of hosting $M$ symbols, the decomposition contains less than $L^M$ terms, hence the upper-bound on the number of terms. $\qquad\square$

We also have a direct corollary bounding the multipartite Schmidt rank measure of entanglement from the bound on the number of terms in the multipartite decomposition,

**Corollary IV.1.** *For any left-right $L-$partition, the $L-$partite Schmidt rank $\chi_L$ (and related entanglement*

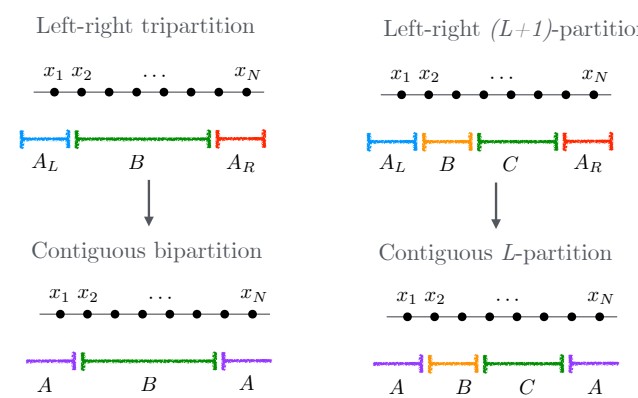

FIG. 8. Converting a left-right $L + 1$ partition to contiguous $L$ partition.

*measure $\log(\chi_L)$ [40]) of an $M-$particle Bethe state is upper-bounded by $L^M$ (and $M\log(L)$), independent of $N$.*

### D. Contiguous multipartite decomposition

So far we have highlighted the multipartite decomposition of Bethe wavefunctions for a left-right partition of a 1d lattice, namely for a partition consisting of $L$ parts $A_1, A_2, \cdots, A_L$ each made of contiguous sites in a 1d chain *with open boundaries*, where the sites in part $A_i$ are to the left of the sites in part $A_j$ when $i < j$. A natural question to ask is whether similar results hold when the $L$ parts are made of contiguous sites *on a ring*, that is, when the last site of the lattice is considered to be nearest neighbor of the first site of the lattice and one of the parts, say part $A_1$, contains sites both at the beginning and at the end of the chain, see Fig. 5. This is relevant since Bethe wavefunctions are often used to represent ground states of integrable 1 dimensional Hamiltonians on a ring (e.g. with periodic boundary conditions). On a ring, there is no natural ordering for the sites, so it is natural to consider also a partition where one of the parts includes both the left-most and right-most sites of the chain, see Fig. 8.

We now show that the a similar decomposition, as a sum of only $L^M$ terms (where each term is the product of $L$ local wavefunctions) is also possible in this case, thus extending to contiguous partitions on a ring the validity of corollary IV.1 that controls the Schmidt measure for multipartite decompositions. In this case, however, each term in the sum is the product of $L-1$ local Bethe wavefunctions and a local wavefunction that is not of that type.

We consider first a contiguous bipartition on the ring with two parts $A$ and $B$, where part $A = A_L \cup A_R$ further breaks into left and right subparts $A_L$ and $A_R$ containing left-most and right-most sites and $B$ contains the rest of sites in between. Our goal is to produce a

bipartite decomposition for an $M$-particle Bethe wavefunction $\left|\vec{1}\right\rangle$ according to the partition $A, B$. Since the three parts $A_L, B, A_R$ define a left-right tripartition, we can use Theorem IV.1 to decompose an $M$-particle Bethe wavefunction $\left|\vec{1}\right\rangle$ as

$$\left|\vec{1}\right\rangle = \sum_{\vec{a}_L, \vec{b}, \vec{a}_R} \delta_{\vec{a}_L \cup \vec{b} \cup \vec{a}_R, \vec{1}} \, \Theta_3[\vec{a}_L, \vec{b}, \vec{a}_R] \left|\vec{a}_L, \vec{b}, \vec{a}_R\right\rangle, \quad (72)$$

where the delta $\delta_{\vec{a}_L \cup \vec{b} \cup \vec{a}_R, \vec{1}}$ ensures that only $3^M$ product terms $\left|\vec{a}_L, \vec{b}, \vec{a}_R\right\rangle$ appear in the decomposition, even though the sum is over $(2^M)^3 = 2^{3M}$ choices $\vec{a}_L, \vec{b}, \vec{a}_R$. Using that

$$\delta_{\vec{a}_L \cup \vec{b} \cup \vec{a}_R, \vec{1}} = \sum_{\vec{a}} \delta_{\vec{a} \cup \vec{b}, \vec{1}} \, \delta_{\vec{a}_L \cup \vec{a}_R, \vec{a}}, \quad (73)$$

we can readily re-organize this decomposition as

$$\left|\vec{1}\right\rangle = \sum_{\vec{a}, \vec{b}} \delta_{\vec{a} \cup \vec{b}, \vec{1}} \left|\Psi_{\vec{a}, \vec{b}}\right\rangle_A \left|\vec{b}\right\rangle_B \quad (74)$$

where

$$\left|\Psi_{\vec{a}, \vec{b}}\right\rangle_A \equiv \sum_{\vec{a}_L \vec{a}_R} \delta_{\vec{a}_L \cup \vec{a}_R, \vec{a}} \, \Theta_3[\vec{a}_L, \vec{b}, \vec{a}_R] \left|\vec{a}_L, \vec{a}_R\right\rangle. \quad (75)$$

Decomposition (74) is a linear combination of at most $2^M$ product states $\left|\Psi_{\vec{a}, \vec{b}}\right\rangle_A \left|\vec{b}\right\rangle_B$, namely one for each value of the choices $\vec{a}, \vec{b}$ such that $\delta_{\vec{a} \cup \vec{b}, \vec{1}} = 1$, as we wanted to show. Notice that $\left|\Psi(\vec{a})\right\rangle$ is in general not a local Bethe wavefunction on part $A$.

Similarly, for a partition into $L$ contiguous parts, we can obtain a decomposition with $L^M$ product terms. We explain it explicitly for $L = 3$, with parts $A, B, C$ where part $A = A_L \cup A_R$ again decomposes into left and right subparts $A_L$ and $A_R$. Our starting point is now the left-right 4-partite decomposition

$$\left|\vec{1}\right\rangle = \sum_{\vec{a}_L, \vec{b}, \vec{c}, \vec{a}_R} \delta_{\vec{a}_L \cup \vec{b} \cup \vec{c} \cup \vec{a}_R, \vec{1}} \, \Theta_4[\vec{a}_L, \vec{b}, \vec{c}, \vec{a}_R] \left|\vec{a}_L, \vec{b}, \vec{c}, \vec{a}_R\right\rangle, \quad (76)$$

of the Bethe wavefunction $\left|\vec{1}\right\rangle$ according to the four parts $A_L, B, C, A_R$, which contains at most $4^M$ terms (instead of $(2^M)^4 = 2^{4M}$) due to the delta $\delta_{\vec{a}_L \cup \vec{b} \cup \vec{c} \cup \vec{a}_R, \vec{1}}$. Using that

$$\delta_{\vec{a}_L \cup \vec{b} \cup \vec{c} \cup \vec{a}_R, \vec{1}} = \sum_{\vec{a}} \delta_{\vec{a} \cup \vec{b} \cup \vec{c}, \vec{1}} \, \delta_{\vec{a}_L \cup \vec{a}_R, \vec{a}}, \quad (77)$$

we rearrange this expression as

$$\left|\vec{1}\right\rangle = \sum_{\vec{a}, \vec{b}, \vec{c}} \delta_{\vec{a} \cup \vec{b} \cup \vec{c}, \vec{1}} \left|\Psi_{\vec{a}, \vec{b}, \vec{c}}\right\rangle_A \left|\vec{b}, \vec{c}\right\rangle_{BC} \quad (78)$$

where

$$\left|\Psi_{\vec{a}, \vec{b}, \vec{c}}\right\rangle_A \equiv \sum_{\vec{a}_L \vec{a}_R} \delta_{\vec{a}_L \cup \vec{a}_R, \vec{a}} \, \Theta_4[\vec{a}_L, \vec{b}, \vec{c}, \vec{a}_R] \left|\vec{a}_L, \vec{a}_R\right\rangle. \quad (79)$$

Decomposition (78) has indeed $3^L$ terms (instead of $(2^M)^3 = 2^{3M}$ terms) because of the delta $\delta_{\vec{a} \cup \vec{b} \cup \vec{c}, \vec{1}}$.

The above argument can be generalized to $L$ parts. We arrive at the result,

**Theorem IV.2 (Contiguous multipartite decomposition of a Bethe wavefunction).** *Given a contiguous multipartition of lattice $\mathcal{L}$ into $L$ parts $\{A_l\}$, $l = 1, \cdots, L$, an $M$-particle Bethe wavefunction $\left|\vec{1}\right\rangle$ on $\mathcal{L}$ can be expressed as a sum of tensor products of local wavefunctions supported on each part. More concretely,*

*1. The multipartite decomposition is,*

$$\left|\vec{1}\right\rangle = \sum_{\vec{a}_1, \cdots, \vec{a}_L} \delta_{\vec{a}_1 \cup \cdots \cup \vec{a}_L, \vec{1}} \left|\Psi_{\{\vec{a}_l\}}\right\rangle_{A_1} |\vec{a}_2, \cdots, \vec{a}_L\rangle_{A_2 \cdots A_L},$$
$$(80)$$

*where $\{|\vec{a}_l\rangle\}$ are local Bethe wavefunctions defined on parts $\{A_l\}$ and the state $\left|\Psi_{\{\vec{a}_l\}}\right\rangle$ is a local wavefunction defined on part $A_1 = A_{1L} \cup A_{1R}$, which is not a Bethe wavefunction but can be built from local Bethe wavefunctions $|\vec{a}_{1L}, \vec{a}_{1R}\rangle$ and scattering phases as*

$$\left|\Psi_{\{\vec{a}_l\}}\right\rangle \equiv \sum_{\vec{a}_{1L}, \vec{a}_{1R}} \delta_{\vec{a}_{1L} \cup \vec{a}_{1R}, \vec{a}_1} \times$$
$$\Theta_{L+1}[\vec{a}_{1L}, \vec{a}_2, \cdots, \vec{a}_L, \vec{a}_{1R}] |\vec{a}_{1L}, \vec{a}_{1R}\rangle. \quad (81)$$

*2. The number of terms in the multipartite decomposition is at most $L^M$, which is independent of the number $N$ of sites in the lattice $\mathcal{L}$ or the numbers $\{N_l\}$ of sites in each of its parts $\{A_l\}$.*

The corollary on the multipartite Schmidt measure follows:

**Corollary IV.2.** *For any contiguous $L-$partition, the $L-$partite Schmidt rank $\chi_L$ (and related entanglement measure $\log(\chi_L)$ [40]) of an $M-$particle Bethe state is upper-bounded by $L^M$ (and $M \log(L)$), independent of $N$.*

## V. TENSOR NETWORK REPRESENTATION

In this section we show that an $M-$particle Bethe wavefunction in a 1d lattice $\mathcal{L}$ made of $N$ sites can be expressed as an exact tensor network state with bond dimension at most $2^M$. We consider explicitly a matrix product state (MPS) and a tree tensor network (TTN) on a regular binary tree as particularly important examples. However, such an exact tensor network representation with upper bounded bond dimension can be obtained also for any network geometry corresponding

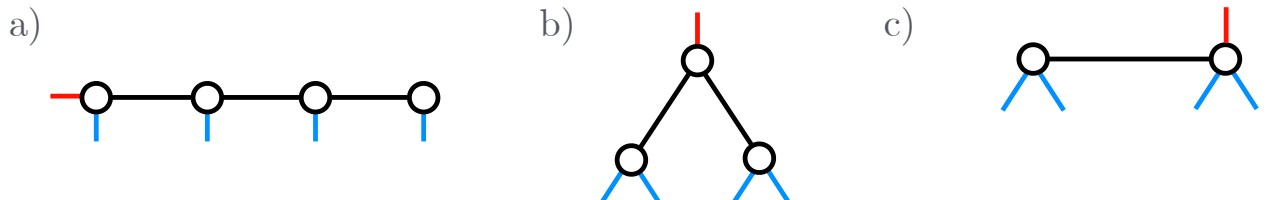

FIG. 9. Three planar trees: (a) chain, leading to an MPS tensor network, see Fig. 10; (b) regular binary tree, leading to a regular binary tree tensor network (regular binary TTN), see Fig. 11; and (c) example of a more general planar tree, leading to a more general planar tree tensor network (planar TTN), see Fig. 13. Given a Bethe wavefunction on a 1d lattice $\mathcal{L}$ and a nested partition of $\mathcal{L}$ according to one such planar tree, in this section we explain how to produce a tensor network representation of the Bethe wavefunction where the network geometry is given by the planar tree.

to a planar tree with one root and $L$ leaves (for $L \leq N$) representing $L$ local parts, see Fig. 9. As mentioned in Sect. I, for an MPS we recover a representation equivalent to that in Ref. [29]. This will be further explained below, see homogeneous MPS representation in Sect. V H.

Our explicit construction of exact tensor network representations for a Bethe wavefunction is based on two types of tensors, denoted $\mathbb{T}$ and $\mathbb{S}$, which we describe next. Their origin will become clear when we describe the MPS and regular binary TTN representations.

### A. Tensor $\mathbb{T}$

Tensor $\mathbb{T}$ has one incoming index and two out-going indices, with each index labelled by a choice vector. It is of the form

$$\mathbb{T}^{\vec{c}}_{\vec{a},\vec{b}} \equiv \delta_{\vec{a}\cup\vec{b},\vec{c}} \, \Theta[\vec{a},\vec{b}] =$$

$$\vec{c} \rightarrow \overset{\mathbb{T}}{\underset{\vec{a} \quad \vec{b}}{\bigcirc}} \tag{82}$$

namely the product of a delta $\delta_{\vec{a}\cup\vec{b},\vec{c}}$, that enforces that choice $\vec{c}$ in the incoming index is equal to the sorted union $\vec{a} \cup \vec{b}$ of the choices $\vec{a}$ and $\vec{b}$ for the two outgoing indices, see Eq. (22), and the scattering amplitude $\Theta[\vec{a},\vec{b}]$ in Eq. (48) involving the choices $\vec{a}$ and $\vec{b}$ of the two outgoing indices. Notice that tensor $\mathbb{T}$ simply encodes the amplitudes of the bipartite decomposition in Eq. (50). Since each choice index can take up to $2^M$ values, this tensor has $(2^M)^3$ components. We emphasize, however, that $\mathbb{T}$ is a decorated delta function and thus a sparse tensor.

Sparsity of tensor $\mathbb{T}$ goes well beyond that of a particle preserving tensor. To illustrate this, let us assume all three indices $\vec{a}, \vec{b}, \vec{c}$ run over the $2^M$ choices compatible with the global Bethe data. Notice that there are $\binom{M}{m}$ incoming choices $\vec{c}$ with $m$ particles, that is with $|\vec{c}| = m$ for $0 \leq m \leq M$. For any of those choices $\vec{c}$, there are $\binom{m}{m_L}$ compatible choices $\vec{a}$ with $m_L$ particles, that is with $|\vec{a}| = m_L$, for $0 \leq m_L \leq m$. For each pair $(\vec{c}, \vec{a})$ of

eligible choices, the delta $\delta_{\vec{a}\cup\vec{b},\vec{c}}$ indicates that there is only one choice $\vec{b}$ leading to a non-zero coefficient $\mathbb{T}^{\vec{c}}_{\vec{a},\vec{b}}$ in $\mathbb{T}$. Therefore the number of non-zero coefficient in tensor $\mathbb{T}$ is

$$\sum_{m=0}^{M} \binom{M}{m} \sum_{m_L=0}^{m} \binom{m}{m_L} = \sum_{m=0}^{M} \binom{M}{m} 2^m \tag{83}$$

In contrast, if only particle conservation was enforced, then non-zero coefficients would correspond to the cases where $|\vec{c}| = m$, $|\vec{a}| = m_L$ and $|\vec{b}| = m_R$ with $m = m_L + m_R$, in which case there would be $\binom{M}{m}\binom{M}{m_L}\binom{M}{m_R}$ non-zero elements. Adding over all possible values of $m$ and $m_L$ (with $0 \leq m_L \leq m \leq M$ and $m_R = m - m_L$) leads to

$$\sum_{m=0}^{M} \binom{M}{m} \sum_{m_L=0}^{m} \binom{M}{m_L} \binom{M}{M - m_L}, \tag{84}$$

which can be a substantially larger number of non-zero coefficients than we found in Eq. (83) for tensor $\mathbb{T}$. It is also easy to check that $\mathbb{T}$ satisfies the normalization

$$\sum_{\vec{a},\vec{b}} \mathbb{T}^{\vec{c}}_{\vec{a},\vec{b}} \mathbb{T}^{\vec{c}'*}_{\vec{a},\vec{b}} = \delta_{\vec{c},\vec{c}'} 2^{|\vec{c}|}. \tag{85}$$

*Examples*

For illustrative purposes, next we describe the explicit form of $\mathbb{T}^{\vec{c}}_{\vec{a},\vec{b}}$ in Eq. (82) for a system with $M = 0, 1$ and 2 particles. Recall from previous sections that in a system with $M$ particle, where $\emptyset$ refers to the null choice of quasi-momenta, and $\vec{1} = (1, 2, 3, \cdots, M)$ refers to the choice of all $M$ Bethe quasi-momenta, a choice index $\vec{\mu}$ typically runs over all $2^M$ possible choices compatible with the global Bethe data. Below, these $2^M$ choices are ordered by increasing number of particles $m = |\vec{\mu}|$, according to

the convention used in Sec. II C, which we restate here

$$
\{\vec{\mu}\} \equiv \Bigg\{ \underbrace{\emptyset}_{m=0}, \\
\underbrace{(1),(2),\cdots,(M)}_{m=1}, \\
\underbrace{(1,2),(1,3),\cdots(M-1,M)}_{m=2}, \\
\cdots, \\
\underbrace{\vec{1}}_{m=M} \Bigg\}. \tag{86}
$$

We present the tensor $\mathbb{T}^{\vec{c}}_{\vec{a},\vec{b}}$ for $M = 0, 1, 2$ particles as a set of $2^M \times 2^M$ matrices after fixing the top index $\vec{c}$. The basis elements are indicated next to the matrices, according to the convention in (86). For $M = 0$ particles, with $\vec{1} = \emptyset$, we simply have

$$
\mathbb{T}^{\emptyset} = \begin{matrix} \emptyset \\ (1) \end{matrix} \begin{pmatrix} \emptyset \end{pmatrix}. \tag{87}
$$

For $M = 1$ particles, with $\vec{1} = (1)$, tensor $\mathbb{T}$ has components

$$
\mathbb{T}^{\emptyset} = \begin{pmatrix} \emptyset & (1) \\ 1 & 0 \\ 0 & 0 \end{pmatrix} \begin{matrix} \emptyset \\ (1) \end{matrix},
$$

$$
\mathbb{T}^{(1)} = \begin{pmatrix} \emptyset & (1) \\ 0 & 1 \\ 1 & 0 \end{pmatrix} \begin{matrix} \emptyset \\ (1) \end{matrix}. \tag{88}
$$

Finally, for $M = 2$ particles, with $\vec{1} = (1,2)$ and scattering angle $\theta_{21}$, tensor $\mathbb{T}$ has components

$$
\mathbb{T}^{\emptyset} = \begin{matrix} \emptyset\ (1)\ (2)\ (1,2) \\ \begin{pmatrix} 1 & 0 & 0 & 0 \\ 0 & 0 & 0 & 0 \\ 0 & 0 & 0 & 0 \\ 0 & 0 & 0 & 0 \end{pmatrix} \end{matrix} \begin{matrix} \emptyset \\ (1) \\ (2) \\ (1,2) \end{matrix}
$$

$$
\mathbb{T}^{(1)} = \begin{matrix} \emptyset\ (1)\ (2)\ (1,2) \\ \begin{pmatrix} 0 & 1 & 0 & 0 \\ 1 & 0 & 0 & 0 \\ 0 & 0 & 0 & 0 \\ 0 & 0 & 0 & 0 \end{pmatrix} \end{matrix} \begin{matrix} \emptyset \\ (1) \\ (2) \\ (1,2) \end{matrix}
$$

$$
\mathbb{T}^{(2)} = \begin{matrix} \emptyset\ (1)\ (2)\ (1,2) \\ \begin{pmatrix} 0 & 0 & 1 & 0 \\ 0 & 0 & 0 & 0 \\ 1 & 0 & 0 & 0 \\ 0 & 0 & 0 & 0 \end{pmatrix} \end{matrix} \begin{matrix} \emptyset \\ (1) \\ (2) \\ (1,2) \end{matrix}
$$

$$
\mathbb{T}^{(1,2)} = \begin{matrix} \emptyset\quad\ (1)\ (2)\ (1,2) \\ \begin{pmatrix} 0 & 0 & 0 & 1 \\ 0 & 0 & 1 & 0 \\ 0 & -e^{i\theta_{21}} & 0 & 0 \\ 1 & 0 & 0 & 0 \end{pmatrix} \end{matrix} \begin{matrix} \emptyset \\ (1) \\ (2) \\ (1,2) \end{matrix} \tag{89}
$$

As these examples make manifest, tensor $\mathbb{T}$ only depends on the two-particle scattering angles $\boldsymbol{\theta}$, and not on the quasi-momenta $\vec{k}$. Notice also that the tensor for $M - 1$ particles can be obtained from that for $M$ particles by simply restricting our attention to a subset of its entries (namely those entries labelled by choices $\vec{a}, \vec{b}, \vec{c}$ containing only the first $M - 1$ symbols).

## B.   Tensor $\mathbb{S}$

Tensor $\mathbb{S}$ implements a (non-unitary) change of basis that expresses a local Bethe wavefunction $|\vec{a}\rangle$, specified by choice $\vec{a}$, in terms of the occupation basis $|\vec{\sigma}\rangle$ (see discussion before Eq. (1)). Let $A$ denote the part of the 1d lattice $\mathcal{L}$ on which $|\vec{\sigma}\rangle$ is supported, and let $N_A$ be the number of site in $A$. Then $\vec{\sigma}$ is a bitstring of length $N_A$. Recall that the local Bethe wavefunction $|\vec{a}\rangle$ is given by

$$
|\vec{a}\rangle \equiv \sum_{\vec{x} \in \mathcal{D}_m^A} \sum_{R \in S_m} \Theta^{\vec{a}}[R] e^{i(\vec{k}^a)^R \cdot \vec{x}} |\vec{x}\rangle, \tag{90}
$$

Then tensor $\mathbb{S}$ has components

$$
\mathbb{S}^{\vec{a}}_{\vec{\sigma}} = \langle \vec{\sigma}|\vec{a}\rangle = \sum_{R \in S_m} \Theta^{\vec{a}}[R] \sum_{\vec{x} \in \mathcal{D}_m^A} e^{i(\vec{k}^a)^R \cdot \vec{x}} \langle \vec{\sigma}|\vec{x}\rangle. \tag{91}
$$

Notice that index $\vec{a}$ runs over $2^M$ choices (when all possible choices are included, as is the case when both

part $A$ and its complement $A'$ in $\mathcal{L}$ can host $M$ particles), whereas index $\vec{\sigma}$ runs over $2^{N_A}$ bitstrings. Accordingly, tensor $\mathbb{S}$ has $2^M \times 2^{N_A}$ components. Tensor $\mathbb{S}$ is also sparse, due to particle number preservation: $\mathbb{S}^{\vec{a}}_{\vec{\sigma}}$ is non-zero only if the particle number $|\vec{a}|$ associated with choice $\vec{a}$ matches the number of ones in $\vec{\sigma}$. Two local Bethe wavefunctions $|\vec{a}\rangle$, $|\vec{a}'\rangle$ are orthogonal if they have a different number of particles $|\vec{a}|$ and $|\vec{a}'|$, that is $\langle \vec{a}'|\vec{a}\rangle = \delta_{|\vec{a}|,|\vec{a}'|} \langle \vec{a}'|\vec{a}\rangle$, and generically not orthogonal, i.e. $\langle \vec{a}'|\vec{a}\rangle \neq \delta_{\vec{a}',\vec{a}}$, if their number of particles is the same. We thus have,

$$\sum_{\vec{\sigma}} \mathbb{S}^{\vec{a}}_{\vec{\sigma}} \mathbb{S}^{\vec{a}'*}_{\vec{\sigma}} = \sum_{\vec{\sigma}} \langle \vec{a}'|\vec{\sigma}\rangle \langle \vec{\sigma}|\vec{a}\rangle = \delta_{|\vec{a}|,|\vec{a}'|} \langle \vec{a}'|\vec{a}\rangle . \quad (92)$$

*Examples*

Let us use Eq. (91) to write down the explicit form of $\mathbb{S}^{\vec{a}}_{\vec{\sigma}_A}$ when the choice $\vec{a}$ corresponds to $0, 1$, or $2$ particles in a system with $M$ particles. We use the notation $\mathbb{S}[i]$ to indicate that this tensor explicitly depends on the part $A_i$ of the 1d lattice $\mathcal{L}$ in which it is defined. Below $\vec{\sigma}_{A_i}$ denotes a bitstring of length $N_{A_i}$, which is the number of sites in part $A_i$.

For 0 particles, that is $\vec{a} = \emptyset$, we obtain

$$\mathbb{S}[i]^{\emptyset}_{\vec{\sigma}_{A_i}} = \delta_{\vec{\sigma}_{A_i},(0,0,\cdots,0)}, \quad (93)$$

whereas in the one-particle sector, namely $\vec{a} = (a)$ for $a = 1, \cdots, M$, we find

$$\mathbb{S}[i]^{(a)}_{\vec{\sigma}_{A_i}} = \sum_{x \in A_i} e^{ik_a x} \, \delta_{\vec{\sigma}_{A_i},(0,0,\cdots,1_x,\cdots,0)}. \quad (94)$$

Finally, in the two-particle sector $\vec{a} = (a_1, a_2)$ (for $1 \leq a_1 < a_2 \leq M$) we have

$$\mathbb{S}[i]^{(a_1,a_2)}_{\vec{\sigma}_{A_i}} = \sum_{\vec{x} \in \mathcal{D}_2^{A_i}} \left( e^{i\left(k_{a_1} x_1 + k_{a_2} x_2\right)} - e^{i\theta_{a_2 a_1}} e^{i\left(k_{a_2} x_1 + k_{a_1} x_2\right)} \right)$$
$$\times \, \delta_{\vec{\sigma}_{A_i},(0,0,\cdots,1_{x_1},\cdots,1_{x_2},\cdots,0)}, \quad (95)$$

and so on.

Notice that in contrast with tensor $\mathbb{T}$, which only depends on the two-particle scattering angles $\boldsymbol{\theta}$, tensor $\mathbb{S}[i]$ depends on both the quasi-momenta $\vec{k}$ and the two-particle scattering angles $\boldsymbol{\theta}$.

## C. Combining tensors $\mathbb{T}$ and $\mathbb{S}$ into a tree tensor network (TTN) representation

Our next step is to explain how these two types of tensors $\mathbb{T}$ and $\mathbb{S}$ can be customized and put together into a tensor network representation of an $M-$particle Bethe wavefunction $\left|\vec{1}\right\rangle$, for a planar tree tensor network.

Given a left-right partition of the 1d lattice $\mathcal{L}$ into $L$ parts $A_1, A_2, \cdots, A_L$ (where part $A_i$ is to the left of part

$A_{i+1}$), we choose a tensor network geometry consisting of a planar tree with one root and $L$ leaves, labelled by $i = 1, 2, \cdots, L$, see Fig. 9. Here leaf $i$ is associated to part $A_i$ of lattice $\mathcal{L}$. We then build a tensor network representation for the Bethe wavefunction $\left|\vec{1}\right\rangle$ in two steps: first (i) we use tensors $\mathbb{T}$ to build a tensor network representation of the amplitudes $\delta_{\vec{a}_1 \cup \cdots \cup \vec{a}_L, \vec{1}} \, \Theta_L[\vec{a}_1, \cdots, \vec{a}_L]$ of the multipartite decomposition (63); then (ii) we dress each leaf with a tensor $\mathbb{S}[i]$ that expresses the local Bethe wavefunction $|\vec{a}_i\rangle$ in Eq. (63) in terms of the local occupation basis $|\vec{\sigma}_{A_i}\rangle$.

For clarity, below we explain these two steps in more detail for two simple, yet particularly important cases of planar trees, namely for a linear chain (leading to an MPS) and a regular binary tree (producing a regular binary TTN), before briefly returning to the general case.

## D. Matrix product state (MPS)

We consider first the case where the planar tree is a chain of $L$ nodes, labelled by $i = 1, 2, \cdots, L$, see Figs. 9(a) and 10. For $1 < i < L$, node $i$ has three edges: two *bond* edges connecting it to the nodes $i-1$ and $i+1$ and a *physical* edge whose other vertex is leaf $i$ of the tree, which is assigned to part $A_i$ of the lattice $\mathcal{L}$. The leftmost node, node $i = 1$, has a left edge connecting it to the root of the tree, in addition to a right bond edge and a physical edge connecting it to leaf $i = 1$. The right-most node, node $i = L$, has a right edge that we added for notational consistency in the final tensor network (but which can be otherwise omitted) as well as a left bond edge and a physical edge connecting it to leaft $i = L$.

Our first step is to build an MPS representation of the amplitudes $\delta_{\vec{a}_1 \cup \cdots \cup \vec{a}_L, \vec{1}} \, \Theta_L[\vec{a}_1, \cdots, \vec{a}_L]$ in Eq. (63) using tensors $\mathbb{T}$. We proceed sequentially from left to right as follows. First we consider a bipartition of the lattice into parts $A_1$ and $M_1 \equiv A_2 \cup \cdots \cup A_L$, and the corresponding bipartite decomposition (49) in terms of local Bethe wavefunctions $|\vec{a}_1\rangle$ and $|\vec{\mu}_1\rangle$ for $A_1$ and $M_1$, which we express as

$$|\vec{1}\rangle = \sum_{\vec{a}_1,\vec{\mu}_1} \overset{\vec{1}}{\longrightarrow} \overset{\mathbb{T}}{\underset{\vec{a}_1}{\bigcirc}} \overset{\vec{\mu}_1}{\longrightarrow} |\vec{a}_1,\vec{\mu}_1\rangle \quad (96)$$

Next, we bipartite decompose the state $|\vec{\mu}_1\rangle$ into local Bethe wavefunctions $|\vec{a}_2\rangle$ and $|\vec{\mu}_2\rangle$ for parts $A_2$ and $M_2 \equiv A_3 \cup \cdots \cup A_L$ respectively, which we also express using a tensor $\mathbb{T}$, and obtain

$$|\vec{1}\rangle = \sum_{\vec{a}_1,\vec{\mu}_1} \sum_{\vec{a}_2,\vec{\mu}_2} \overset{\vec{1}}{\longrightarrow} \overset{\mathbb{T}}{\underset{\vec{a}_1}{\bigcirc}} \overset{\vec{\mu}_1}{\longrightarrow} \overset{\mathbb{T}}{\underset{\vec{a}_2}{\bigcirc}} \overset{\vec{\mu}_2}{\longrightarrow} |\vec{a}_1,\vec{a}_2,\vec{\mu}_2\rangle \quad (97)$$

After iterating bipartite decompositions $L - 1$ times, we end up with the following decomposition,

$$|\vec{1}\rangle = \sum_{\vec{a}_1,\vec{\mu}_1} \sum_{\vec{a}_2,\vec{\mu}_2} \cdots \sum_{\vec{a}_L} \quad \xrightarrow{\vec{1}} \underset{\vec{a}_1}{\overset{\mathbb{T}}{\circ}} \xrightarrow{\vec{\mu}_1} \underset{\vec{a}_2}{\overset{\mathbb{T}}{\circ}} \xrightarrow{\vec{\mu}_2} \cdots \xrightarrow{\vec{\mu}_{L-1}} \underset{\vec{a}_L}{\overset{\mathbb{T}}{\circ}} \xrightarrow{\emptyset} \quad |\vec{a}_1,\vec{a}_2,\cdots,\vec{a}_L\rangle \tag{98}$$

Here, the label $\vec{a}_L$ at the right-most end of the chain is equivalent to $\vec{\mu}_{L-1}$, as enforced by the right-most tensor $\mathbb{T}$, with components $\mathbb{T}_{\vec{a}_L,\emptyset}^{\vec{\mu}_{L-1}} = \delta_{\vec{\mu}_{L-1},\vec{a}_L \cup \emptyset} \; \Theta[\vec{a}_L,\emptyset]$, with $\Theta[\vec{a}_L,\emptyset] = 1$, which is introduced for notational consistency. This completes our construction of an explicit MPS for the Bethe wavefunction $\left|\vec{1}\right\rangle$ when expressed in terms of products of $L$ local Bethe wavefunctions $|\vec{a}_1,\vec{a}_2,\cdots,\vec{a}_L\rangle$, which we can also express as

$$\delta_{\vec{a}_1 \cup \cdots \cup \vec{a}_L, \vec{1}} \; \Theta_L[\vec{a}_1,\cdots,\vec{a}_L] =$$
$$\sum_{\{\vec{\mu}_i\}} \mathbb{T}_{\vec{a}_1,\vec{\mu}_1}^{\vec{1}} \left( \prod_{i=2}^{L-1} \mathbb{T}_{\vec{a}_i,\vec{\mu}_i}^{\vec{\mu}_{i-1}} \right) \mathbb{T}_{\vec{a}_L,\emptyset}^{\vec{\mu}_{L-1}}. \tag{99}$$

Our second step is to transform the above MPS representation of $\left|\vec{1}\right\rangle$ in the local Bethe wavefunction basis $\{|\vec{a}_1,\vec{a}_2,\cdots,\vec{a}_L\rangle\}$ into an MPS representation in the original, local occupation basis $|\vec{\sigma}\rangle = |\vec{\sigma}_{A_1},\vec{\sigma}_{A_2},\cdots,\vec{\sigma}_{A_L}\rangle$. To do so, we simply append a change-of-basis tensor $\mathbb{S}[i]$ to the outgoing downward index $\vec{a}_i$ of each tensor $\mathbb{T}$ in Eq. (98). In doing so, we produce $L$ new tensors $\mathbb{R}[i]$ of the form

$$\mathbb{R}[i]_{\vec{\sigma}_{A_i}}^{\vec{\mu}_L,\vec{\mu}_R} \equiv \sum_{\vec{a}_i} \mathbb{T}_{\vec{a}_i,\vec{\mu}_R}^{\vec{\mu}_L} \mathbb{S}[i]_{\vec{\sigma}_{A_i}}^{\vec{a}_i}. \tag{100}$$

Then the MPS representation for the Bethe wavefunction

$$|\vec{1}\rangle = \sum_{\vec{\sigma}_{A_1},\cdots,\vec{\sigma}_{A_L}} \Psi_L[\vec{\sigma}_{A_1},\cdots,\vec{\sigma}_{A_L}] \, |\vec{\sigma}_{A_1},\cdots,\vec{\sigma}_{A_L}\rangle \tag{101}$$

reads

$$\Psi_L[\vec{\sigma}_{A_1},\cdots,\vec{\sigma}_{A_L}] = \tag{102}$$
$$\sum_{\{\vec{\mu}_i\}} \mathbb{R}[1]_{\vec{\sigma}_{A_1}}^{\vec{1},\vec{\mu}_1} \left( \prod_{i=2}^{L-1} \mathbb{R}[i]_{\vec{\sigma}_{A_i}}^{\vec{\mu}_{i-1},\vec{\mu}_i} \right) \mathbb{R}[L]_{\vec{\sigma}_{A_L}}^{\vec{\mu}_{L-1},\emptyset}.$$

Note, the bond indices in this MPS are labeled by a choice vector $\vec{\mu}_i$. Given a global Bethe wavefunction of $M$ particle on $N$ sites, the number of independent choice vectors is at most $2^M$; thus the bond dimension is upper-bounded by $2^M$, independent of $N$. Let us summarise the result in the form of a theorem,

**Theorem V.1 (MPS representation of Bethe wavefunctions).** *Given an $M$−particle Bethe wavefunction with Bethe data $(\vec{k},\boldsymbol{\theta})$ on a 1d lattice $\mathcal{L}$ made of $N$ sites, and a left-right partition of lattice $\mathcal{L}$ into $L$*

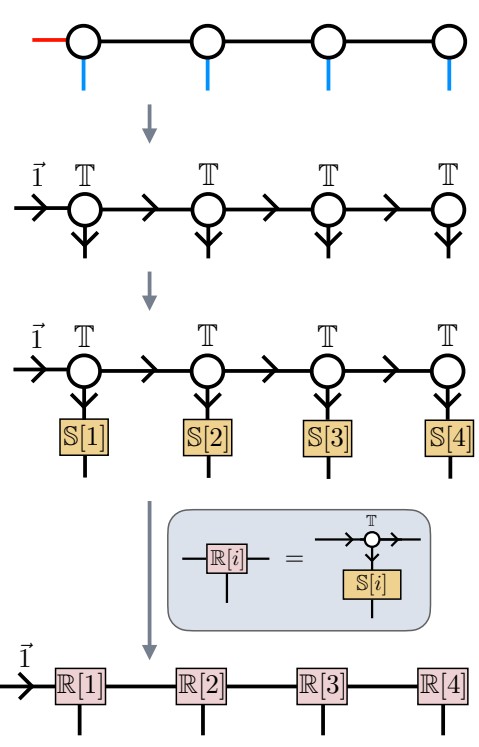

FIG. 10. Starting from the chain in Fig. 9a, we build a MPS representation of a Bethe wavefunction in three steps. First we replace each node of the chain with a tensor $\mathbb{T}$ in Eq. (82), which already produces an MPS representation of the Bethe wavefunction when expressed in a local basis made of local Bethe wavefunctions, see Eq. (99). Our second step is to introduce change-of-basis tensors $\mathbb{S}[i]$ in Eq. (91), which leads to a tensor network description of the Bethe wavefunction in the local occupation basis. Finally, the MPS is naturally expressed in terms of the tensors $\mathbb{R}[i]$ in Eq. (100), which result from multiplying tensors $\mathbb{T}$ and $\mathbb{S}[i]$ together.

*parts $\{A_i\}$, one can explicitly construct the exact matrix product state representation in Eq. (102), made of $L$ tensors $\mathbb{R}$, which has bond dimension upper-bounded by $2^M$, independent of $N$.*

For a proof, see how we constructed the exact MPS representation (102), written in terms of MPS tensors $\mathbb{R}[i]$, by reviewing the definitions of the tensors $\mathbb{R}, \mathbb{S}, \mathbb{T}$ and the bipartite scattering amplitudes $\Theta$, Eqs. (48), (82), (91), and (100) (see also Fig. 10).

Two remarks are in order. First, here we chose to present the explicit MPS representation in terms of a

single tensor $\mathbb{R}[i]$ for each site, as this is how a MPS is usually expressed. However, in practical applications one can also directly use the pair of tensors $(\mathbb{R}, \mathbb{S}[i])$ instead of $\mathbb{R}[i]$, which may have advantages in terms of storage and computational costs due to their sparsity. Second, in an MPS representation one often chooses $L = N$, so that each part $A_i$ corresponds to a single site of the 1d lattice $\mathcal{L}$. Our MPS representation is more general. For instance, for an integer $p$ such that $N/p$ is also an integer, we could set $L = N/p$ and have each part $A_i$ correspond to $p$ contiguous sites of the lattice, instead of just one site.

*Examples*

Let us build explicit MPS tensors $\mathbb{R}[i]$ for Bethe wavefunctions with $M = 0, 1, 2, 3$ particles, for the special case where each part $A_i$ is a single site of the 1d lattice $\mathcal{L}$, corresponding to local occupation basis $\{|0\rangle, |1\rangle\}$, for an arbitrary number $N$ of sites or, equivalently, for an arbitrary number $L$ of parts (with $L = N$), where we can use the site label $x = 1, 2, \cdots, N$ instead of the part label $i = 1, 2, \cdots, L$. We interpret tensor $\mathbb{R}[x]_\sigma^{\vec{\mu}_L, \vec{\mu}_R}$ as a collection of two $2^M \times 2^M$ matrices $\mathbb{R}[x]_\sigma$, one matrix for each value $\sigma = 0, 1$ of the on-site physical index. Each entry of these matrices is labeled by $\vec{\mu}_L, \vec{\mu}_R$, where the basis is organized according to convention (86). We then have

$$\mathbb{R}[x]_0^{\vec{\mu}\vec{\mu}'} = \delta_{\vec{\mu}, \vec{\mu}'}, \tag{103}$$

$$\mathbb{R}[x]_1^{\vec{\mu}\vec{\mu}'} = \sum_{j=1}^{M} \delta_{\vec{\mu}, \vec{\mu}' \cup (j)} \, \Theta[(j), \vec{\mu}'] \, e^{ik_j x}, \tag{104}$$

and the number of non-zero coefficients is obtained from Eq. (83) by properly restricting the range of the second sum:

$$\sum_{m=0}^{M} \binom{M}{m} \sum_{m_L=0}^{1} \binom{m}{m_L} = \sum_{m=0}^{M} \binom{M}{m}(m+1). \tag{105}$$

More specifically, for $M = 0$ particles, the matrices are,

$$\mathbb{R}[x]_0 = \begin{pmatrix} 1 \end{pmatrix}, \quad \mathbb{R}[x]_1 = \begin{pmatrix} 0 \end{pmatrix}. \tag{106}$$

This corresponds to an MPS with bond dimension $\chi = 1$, representing the product state $|\emptyset\rangle = |00 \cdots 0\rangle$ in Eq. (4).

For $M = 1$ particles, the matrices are

$$\mathbb{R}[x]_0 = \begin{pmatrix} 1 & 0 \\ 0 & 1 \end{pmatrix}, \quad \mathbb{R}[x]_1 = \begin{pmatrix} 0 & 0 \\ e^{ikx} & 0 \end{pmatrix}. \tag{107}$$

This corresponds to an MPS with bond dimension $\chi = 2$, which can be seen to indeed represent a plane wave with quasi-momenta $k$ in Eq. (4).

For an $M = 2$ particles with quasi-momenta $k_1, k_2$ and scattering angle $\theta_{21}$, the matrices $\mathbb{R}[x]^0$ and $\mathbb{R}[x]^1$ are of the form $\mathbb{R}[x]_0 = \mathbb{I}_4$,

$$\mathbb{R}[x]_1 = \begin{pmatrix} 0 & 0 & 0 & 0 \\ e^{ik_1 x} & 0 & 0 & 0 \\ e^{ik_2 x} & 0 & 0 & 0 \\ 0 & -e^{i\theta_{21}}e^{ik_2 x} & e^{ik_1 x} & 0 \end{pmatrix}. \tag{108}$$

This corresponds to an MPS with bond dimension $\chi = 4$, which can again be checked to represent the two-particle Bethe wavefunction in Eq. (5).

For $M = 3$ particles with quasi-momenta $k_1, k_2, k_2$ and scattering angle $\theta_{21}, \theta_{31}, \theta_{32}$, we have $8 \times 8$ matrices $\mathbb{R}[x]_0 = \mathbb{I}_8$ and

$$\mathbb{R}[x]_1 = \begin{pmatrix} 0 & 0 & 0 & 0 & 0 & 0 & 0 & 0 \\ e^{ik_1 x} & 0 & 0 & 0 & 0 & 0 & 0 & 0 \\ e^{ik_2 x} & 0 & 0 & 0 & 0 & 0 & 0 & 0 \\ e^{ik_3 x} & 0 & 0 & 0 & 0 & 0 & 0 & 0 \\ 0 & -e^{i\theta_{21}}e^{ik_2 x} & e^{ik_1 x} & 0 & 0 & 0 & 0 & 0 \\ 0 & -e^{i\theta_{31}}e^{ik_3 x} & 0 & e^{ik_1 x} & 0 & 0 & 0 & 0 \\ 0 & 0 & -e^{i\theta_{32}}e^{ik_3 x} & e^{ik_2 x} & 0 & 0 & 0 & 0 \\ 0 & 0 & 0 & 0 & e^{i\theta_{31}}e^{i\theta_{32}}e^{ik_3 x} & -e^{i\theta_{21}}e^{ik_2 x} & e^{ik_1 x} & 0 \end{pmatrix}. \tag{109}$$

This corresponds to an MPS with bond dimension $\chi = 8$ that can be seen to represent the three-particle Bethe wavefunction in Eq. (9).

Notice that the matrix elements $\mathbb{R}[x]_0^{\vec{\mu}_L, \vec{\mu}_R}$ and $\mathbb{R}[x]_1^{\vec{\mu}_L, \vec{\mu}_R}$ for $M-1$ particles can be obtained from those for $M$ particles by restricting our attention to a proper subset of choices $\vec{\mu}_L, \vec{\mu}_R$.

### E. Tree tensor network (TTN): regular binary tree

We now consider a tensor network where the network geometry is that of a regular binary tree, see Figs. 9b and 11. As in the MPS case discussed above, the tree has $L$ leaves labelled by integer $i$, with $i = 1, 2, \cdots, L$, where now we assume that $L = 2^Z$ for some integer $Z \geq 1$, and the graph is organized in $Z$ layers of nodes, each labelled

by a pair of integers $(z,i)$, where $z = 0,1,\cdots,Z-1$ denotes the layer or *scale* of the node and $i = 1,2,\cdots,2^z$ denotes the position of the node within that layer. Each node has three edges. For $z = 1,2,\cdots,Z-2$, the three edges of node $(z,i)$ consist of a *bond* edge connecting it to node $(z-1,(i+1)$ div $2)$ of the layer $z-1$ above (here $i$ div $2$ denotes integer division of $i$ by $2$) and two *bond* edges connecting it to nodes $(z+1,2i-1)$ and $(z+1,2i)$ of the layer $z+1$ below. At the top layer $z = 0$, node $(z = 0,i = 1)$ has an edge representing the *root* of the tree and two bond edges connecting it to nodes $(z = 1,i = 1)$ and $(z = 1,i = 2)$, see Fig. 9b. At the bottom layer $z = Z-1$, we have $2^{Z-1}$ nodes. Each such node $(z = Z-1,i)$ has one bond edge connecting it to node $(z = Z-2,(i+1)$ div $2)$ of layer $z = Z-2$ and two *physical* edges connecting them to two leaves $2i-1$ and $2i$.

As we did for the MPS case earlier on, our first step is to build a TTN representation of the amplitudes $\delta_{\vec{a}_1\cup\cdots\cup\vec{a}_L\vec{1}}\,\Theta_L[\vec{a}_1,\cdots,\vec{a}_L]$ in Eq. (63) using tensors $\mathbb{T}$, but now for the above regular binary tree. We proceed by first bipartite decomposing the Bethe wavefunction $|\vec{1}\rangle$ into two local Bethe wavefunctions $|\vec{\mu}_{[1,1]}\rangle$ and $|\vec{\mu}_{[1,2]}\rangle$ supported on parts $M_{[1,1]} \equiv \bigcup_{i=1}^{L/2} A_i$ and $M_{[1,2]} \equiv \bigcup_{i=L/2+1}^{L} A_i$ respectively,

$$|\vec{1}\rangle = \sum_{\vec{\mu}_{[1,1]},\vec{\mu}_{[1,2]}} \quad \raisebox{-0.5em}{\includegraphics{}} \quad |\vec{\mu}_{[1,1]},\vec{\mu}_{[1,2]}\rangle \tag{110}$$

Here we have introduced the choice vectors $\vec{\mu}_{[z,i]}$ with $z = 1$ and $i = 1,2$ that label Bethe wavefunctions $|\vec{\mu}_{[z,i]}\rangle$ supported on the two parts. Next, we bipartite decompose each $|\vec{\mu}_{[1,i]}\rangle$ further as,

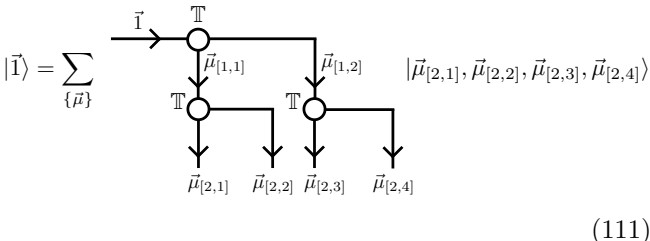

$$|\vec{1}\rangle = \sum_{\{\vec{\mu}\}} \quad\quad |\vec{\mu}_{[2,1]},\vec{\mu}_{[2,2]},\vec{\mu}_{[2,3]},\vec{\mu}_{[2,4]}\rangle \tag{111}$$

in terms of the next set of local Bethe wavefunctions $|\vec{\mu}_{[2,2i-1]}\rangle$ and $|\vec{\mu}_{[2,2i]}\rangle$. Here we have divided lattice $\mathcal{L}$ into four subregions $M_{[2,i]}$ for $i = 1,2,3,4$, with $M_{[2,1]} \equiv \bigcup_{i=1}^{L/4} A_i$, $M_{[2,2]} \equiv \bigcup_{i=L/4+1}^{L/2} A_i$, etc, and $\sum_{\{\vec{\mu}\}}$ refers to sums over all choice vectors $\vec{\mu}_{z,i}$ in the diagram.

This process of hierarchical bipartite decompositions can be iterated $Z$ times, until the Bethe wavefunction $|\vec{1}\rangle$ is expressed in terms of $L = 2^Z$ local Bethe wavefunctions suppported on the original $L$ parties $\{A_i\}$. For notational consistency, we label the lowest layer choice

vectors as $\vec{a}_i$, and the diagrammatic representation of the regular binary TTN decomposition of coefficients $\delta_{\vec{a}_1\cup\cdots\cup\vec{a}_L\vec{1}}\,\Theta_L[\vec{a}_1,\cdots,\vec{a}_L]$ in Eq. (63) using tensors $\mathbb{T}$ is

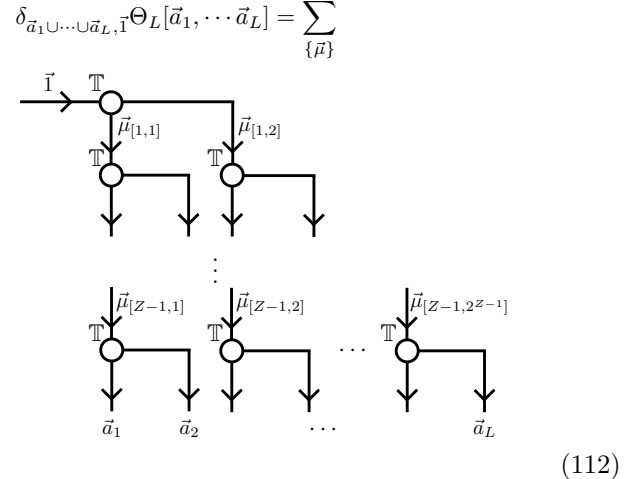

$$\delta_{\vec{a}_1\cup\cdots\cup\vec{a}_L,\vec{1}}\Theta_L[\vec{a}_1,\cdots\vec{a}_L] = \sum_{\{\vec{\mu}\}} \tag{112}$$

The above TTN representation has depth $Z = \log_2 L$, that is, we used $Z$ layers of tensors, with a total of $1 + 2 + \cdots + 2^{L-1} = L-1$ tensors $\mathbb{T}$. Its bond dimension for an $M-$particle Bethe wavefunction is upper-bounded by $2^M$. As with the MPS, this tensor network can be converted into a TTN representation for $\left|\vec{1}\right\rangle$ in the local occupation basis by the local basis transformation using the matrices $\mathbb{S}$ as defined in Eq. (91), see Fig. 11, where tensor $\mathbb{S}[i]$ acts on leaf $i$ and depends on the position vector $\vec{x}_{A_i} \in A_i$, see Fig. 11.

We summarise the above result in a theorem,

**Theorem V.2 (Regular binary TTN representation of Bethe wavefunctions).** *Given an $M-$particle Bethe wavefunction with Bethe data $(\vec{k},\boldsymbol{\theta})$ on a 1d lattice $\mathcal{L}$ made of $N$ sites, and a left-right partition of lattice $\mathcal{L}$ into $L$ parts $\{A_i\}$, one can explicitly construct the exact regular binary TTN representation in Fig. (11), made of $L-1$ tensors $\mathbb{T}$ and $L$ tensors $\mathbb{S}[i]$, which have bond dimension upper-bounded by $2^M$, independent of $N$.*

For a proof, see preceding discussions.

*Examples*

Consider a Bethe wavefunction of $M = 2$ particles on a 1d lattice $\mathcal{L}$ made of $N$ sites (for $N = 4p$, for an integer $p > 0$), left-right partitioned into $L = 4$ parts $A_1, A_2, A_3, A_4$, each made of $p = N/4$ contiguous sites. In this case, we obtain a regular binary TTN representation made of 2 layers of $\mathcal{T}$ tensors (with a total of $1+2 = 3$ tensors $\mathbb{T}$) and one layer of tensors $\mathbb{S}[i]$ (with a total of 4 tensors $\mathbb{S}[i]$). Let the Bethe data be the two quasi-momenta $k_1,k_2$ and the scattering angle $\theta_{21}$. Then the TTN is given by Fig. 11, where each tensor $\mathbb{T}$, with

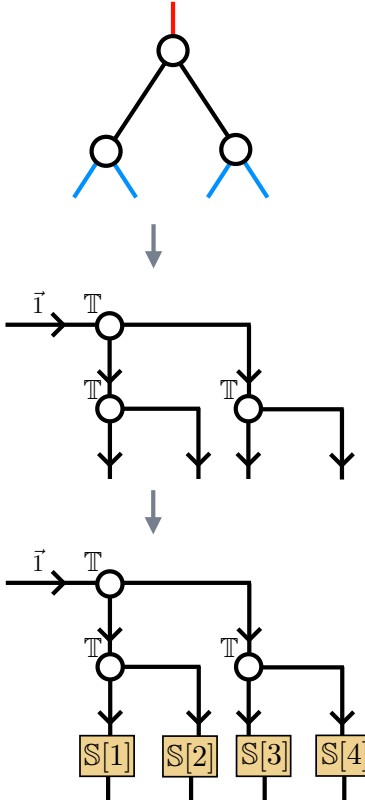

FIG. 11. TTN representation of Bethe wavefunction in local occupation basis. Starting from the regular binary tree in Fig. 9b, we build a TTN representation of a Bethe wavefunction in two steps. First we replace each node of the tree with a tensor $\mathbb{T}$ in Eq. (82), which already produces a regular binary TTN representation of the Bethe wavefunction when expressed in a local basis made of local Bethe wavefunctions, see Eq. (112). Our second step is to introduce change-of-basis tensors $\mathbb{S}[i]$ in Eq. (91), which leads to our final regular binary TTN representation of the Bethe wavefunction in the local occupation basis.

$M = 2$, is given in Eq. (89). The relevant tensors $\mathbb{S}$ are given in Eqs. (93), (94) and (95).

### F. Tree tensor network (TTN): generic planar tree

Finally, let us consider a generic planar tree made of some number $K$ of nodes interconnected by bond edges, $L$ physical edges and a root edge. For instance, Figs. 9c and 12 show an example with $K = 2$ nodes and $L = 4$ physical edges). We unambiguously assign a direction to each edge of the planar tree graph by defining a flow from the root to any of the leafs. As a result, each node has one in-coming index and a number of out-going indices. We assign to each of the $K$ nodes a (generalized) tensor $\mathbb{T}$ with one incoming index $\vec{\mu}$ and the corresponding number, say $q$, of out-going indices $\vec{\nu}_1, \vec{\nu}_2, \cdots, \vec{\nu}_q$, defined

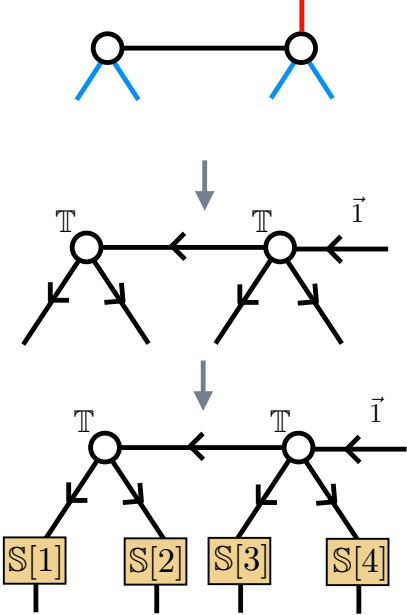

FIG. 12. Given a planar tree with $K$ nodes, a root (red leg) and $L$ leaves (blue legs), we build a corresponding planar TTN representation for a Bethe wavefunction on an $N$-site 1d lattice $\mathcal{L}$ partitioned into $L$ parts $\{A_i\}$ in two steps. First we define a flow from the root to the leaves, which gives each edge a direction, and assign a tensor $\mathbb{T}$ in Eq. (113) to each node. This already results in a TTN for the Bethe wavefunction when expressed in terms of products of $L$ local Bethe wavefunctions. Then we assign a change-of-basis tensor $\mathbb{S}[i]$ in Eq. (91) to each leaf of the tree, resulting in the final TTN representation of the Bethe wavefunction in the local occupation basis.

by

$$\mathbb{T}^{\vec{\mu}}_{\vec{\nu}_1, \vec{\nu}_2, \cdots, \vec{\nu}_q} = \delta_{\vec{\nu}_1 \cup \vec{\nu}_2 \cup \cdots \cup \vec{\nu}_q, \vec{\mu}} \; \Theta_q[\vec{\nu}_1, \vec{\nu}_2, \cdots, \vec{\nu}_q]. \quad (113)$$

Then a TTN representation of the amplitudes $\delta_{\vec{a}_1 \cup \cdots \cup \vec{a}_L \vec{1}} \; \Theta_L[\vec{a}_1, \cdots, \vec{a}_L]$ in Eq. (63) for a Bethe wavefunction $\left|\vec{1}\right\rangle$ is obtained by setting to choice $\vec{1}$ the index corresponding to the root of the tree. Furthermore, by dressing each of the $L$ leaves with a corresponding tensor $\mathbb{S}[i]$, we obtain a TTN decomposition of the same Bethe wavefunction $\left|\vec{1}\right\rangle$ expressed in the occupation basis.

**Theorem V.3 (General planar TTN representation of Bethe wavefunctions).** *Given (1) an $M$-particle Bethe wavefunction $\left|\vec{1}\right\rangle$ with the Bethe data $(\vec{k}, \boldsymbol{\theta})$ on a 1d lattice $\mathcal{L}$ made of $N$ sites, (2) a left-right partition of lattice $\mathcal{L}$ into $L$ parts $\{A_i\}$, and (3) a planar tree with $K$ nodes, one root edge and $L$ leaf edges (see text above for further description), one can explicitly construct an exact TTN representation of $\left|\vec{1}\right\rangle$ made*

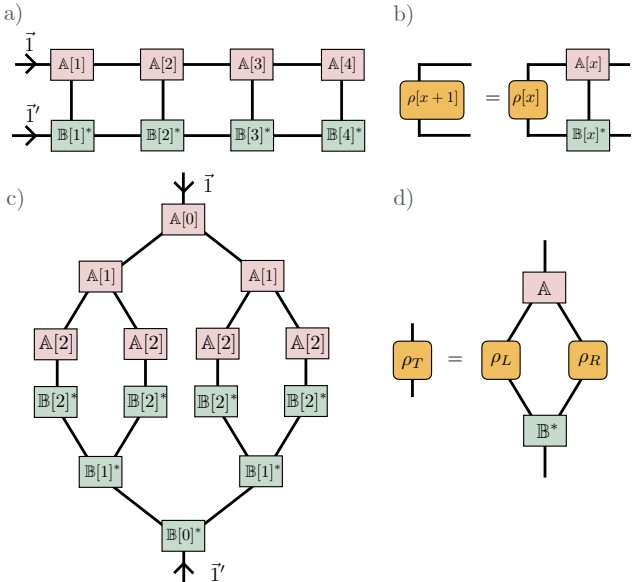

FIG. 13. (a) Tensor network representing the overlap between two wavefunctions expressed as a MPS, see Eq. (114). (b) Typical tensor contraction required in order to contract the MPS overlap tensor network. (c) Tensor network representing the overlap between two wavefunctions expressed as a regular binary TTN. (d) Typical tensor contraction required in order to contract the regular binary TTN overlap tensor network, see Eq. (124).

of $K$ tensors $\mathbb{T}$ (of appropriate degree, see Eq. (113)) and $L$ tensors $\mathbb{S}[i]$ in Eq. (91), with bond dimension upper-bounded by $2^M$, independent of $N$.

For a proof, see preceding discussion.

Note that this theorem includes both Theorem V.1 for MPS and Theorem V.2 for a regular binary TTN as special cases.

### G. Computation of norms and overlaps

Given a tensor network representation of Bethe wavefunctions, as the ones provided earlier on in this section, an important question is how to extract quantities of interest. Two examples of quantities we would like to compute from the tensor network representation are:

(i) The overlap $\left\langle \Phi_M^{\vec{k}',\boldsymbol{\theta}'} \middle| \Phi_M^{\vec{k},\boldsymbol{\theta}} \right\rangle$ between two Bethe wavefunctions $\left| \Phi_M^{\vec{k},\boldsymbol{\theta}} \right\rangle$ and $\left| \Phi_M^{\vec{k}',\boldsymbol{\theta}'} \right\rangle$; this includes the squared norm $\left\langle \Phi_M^{\vec{k},\boldsymbol{\theta}} \middle| \Phi_M^{\vec{k},\boldsymbol{\theta}} \right\rangle$ of a single Bethe wavefunction $\left| \Phi_M^{\vec{k},\boldsymbol{\theta}} \right\rangle$. Notice that so far all decompositions and tensor network representations presented in this work were for a Bethe wavefunction that had not been normalized.

(ii) The expectation value

$$\left\langle \Phi_M^{\vec{k},\boldsymbol{\theta}} \middle| o_1(x_1) o_2(x_2) \cdots o_k(x_k) \middle| \Phi_M^{\vec{k},\boldsymbol{\theta}} \right\rangle$$

of a product of local operators $o_1(x_1), o_2(x_2), \cdots, o_k(x_q)$.

The second computation can be seen to be closely related to the first one, which is the only one that we will analyze here. To compute the overlap $\left\langle \Phi_M^{\vec{k}',\boldsymbol{\theta}'} \middle| \Phi_M^{\vec{k},\boldsymbol{\theta}} \right\rangle$, we first build a single tensor network by merging together the tensor networks for $\left\langle \Phi_M^{\vec{k}',\boldsymbol{\theta}'} \right|$ and $\left| \Phi_M^{\vec{k},\boldsymbol{\theta}} \right\rangle$ through their physical indices, see Figs. 13a and 13c, then contract all the indices using a standard contraction order specific to that tensor network. Bellow we outline the computational cost involved in such calculations for the specific cases of an MPS and a regular binary TTN. We assume that both Bethe wavefunctions have the same particle number $M$ (otherwise the overlap vanishes) but possibly different Bethe data.

#### MPS

Consider an MPS representation of $\left| \Phi_M^{\vec{k},\boldsymbol{\theta}} \right\rangle$ and $\left| \Phi_M^{\vec{k}',\boldsymbol{\theta}'} \right\rangle$ where, for simplicity, we assume that each physical index corresponds to a single site of the 1d lattice $\mathcal{L}$ (that is, the number $L$ of parts in the multipartition coincides with the number $N$ of sites in lattice $\mathcal{L}$, $L = N$, as discussed in a previous example). Let the tensors of the two MPS representations (which are of the form of the tensor $\mathbb{R}[x]$ defined in Eq. (100)) be denoted by $\mathbb{A}[x]$ and $\mathbb{B}[x]$, where $x = 1, 2, \cdots, N$ labels lattice sites. To compute the overlap $\left\langle \Phi_M^{\vec{k}',\boldsymbol{\theta}'} \middle| \Phi_M^{\vec{k},\boldsymbol{\theta}} \right\rangle$, we build a one-dimensional tensor network by joining the MPS representation of $\left| \Phi_M^{\vec{k},\boldsymbol{\theta}} \right\rangle$ and of $\left\langle \Phi_M^{\vec{k}',\boldsymbol{\theta}'} \right|$ through their physical indices as in the example of Fig. 13a. A key step in the computation of this overlap by following the standard (left-to-right) contraction order is the product of three tensors, see Fig.13:

$$\rho[x+1]_{\vec{\nu}'}^{\vec{\mu}'} = \sum_{\vec{\mu},\vec{\nu},i} \rho[x]_{\vec{\nu}}^{\vec{\mu}} \, \mathbb{A}[x]_i^{\vec{\mu},\vec{\mu}'} \, \mathbb{B}[x]_i^{*\vec{\nu},\vec{\nu}'} \qquad (114)$$

where $\rho[x]_{\vec{\nu}}^{\vec{\mu}}$ represents the result of having previously contracted all tensors to the left of $\mathbb{A}[x]$ and $\mathbb{B}[x]$ and has the form

$$\rho[x]_{\vec{\nu}}^{\vec{\mu}} = \delta_{|\vec{\mu}|,|\vec{\nu}|} \, \rho[x]_{\vec{\nu}}^{\vec{\mu}} \qquad (115)$$

namely it vanishes unless the particle number is conserved, that is unless $|\vec{\mu}| = |\vec{\nu}|$, and where from Eqs. (103)-(104) we have

$$\mathbb{A}[x]_0^{\vec{\mu}\vec{\mu}'} = \delta_{\vec{\mu},\vec{\mu}'}, \qquad (116)$$

$$\mathbb{A}[x]_1^{\vec{\mu}\vec{\mu}'} = \sum_{j=1}^{M} \delta_{\vec{\mu},\vec{\mu}'\cup(j)} \, \Theta[(j),\vec{\mu}'] \, e^{ik_j x}, \qquad (117)$$

$$\mathbb{B}[x]_0^{*\vec{\nu}\vec{\nu}'} = \delta_{\vec{\nu},\vec{\nu}'}, \qquad (118)$$

$$\mathbb{B}[x]_1^{*\vec{\nu}\vec{\nu}'} = \sum_{j=1}^{M} \delta_{\vec{\nu},\vec{\nu}'\cup(j)} \, \Theta'[(j),\vec{\nu}'] \, e^{-i\tilde{k}_j x}. \qquad (119)$$

Therefore $\rho$ is block-diagonal in particle number $m = |\vec{\mu}|$, with blocks of size $\binom{M}{m} \times \binom{M}{m}$ and thus contains

$$\sum_{m=0}^{M} \binom{M}{m}^2 \qquad (120)$$

non-zero coefficients, whereas each of $\mathbb{A}$ and $\mathbb{B}$ contains

$$\sum_{m=0}^{M} \binom{M}{m} \times (m+1) \qquad (121)$$

non-zero coefficients, see Eq. (105). The product $\rho$ times $\mathbb{A}$ (namely the sum over index $\vec{\mu}$ in Eq. (114)) involves a sum over particle number $m = |\vec{\mu}|$ of the product of a $\binom{M}{m} \times \binom{M}{m}$ block of $\rho$ with a $\binom{M}{m} \times (m+1)$ block of $\mathbb{A}$ and thus requires a number of operations that scales as

$$\sum_{m=0}^{M} \binom{M}{m}^2 \times (m+1). \qquad (122)$$

(And so does the product of the resulting tensor with $\mathbb{B}^*$, namely the sum over index $\vec{\nu}$ in Eq. (114)). This is less than $\sum_{m=0}^{M} \binom{M}{m}^3$ – which is what would follow from particle number conservation alone. Computing the overlap between two Bethe wavefunctions on $N$ sites requires $O(N)$ contractions of the form (114) and thus scales as

$$O\left( N \sum_{m=0}^{M} \binom{M}{m}^2 \times (m+1) \right), \qquad (123)$$

that is, the computational cost is proportional to the lattice size $N$. In contrast, the overlap between two generic $M-$particle wavefunctions scales as $\binom{N}{M} \approx N^M/M!$, and thus as a higher power of the system size for $M > 1$ particles. Similar expressions can be obtained for the expectation value of local observables.

### *TTN*

Fig. 13c shows an example of tensor network describing the overlap $\left\langle \Phi_M^{\vec{k}',\boldsymbol{\theta}'} \middle| \Phi_M^{\vec{k},\boldsymbol{\theta}} \right\rangle$ between two Bethe wavefunctions $\left| \Phi_M^{\vec{k},\boldsymbol{\theta}} \right\rangle$ and $\left| \Phi_M^{\vec{k}',\boldsymbol{\theta}'} \right\rangle$ expressed as regular binary TTNs, where tensors $\mathbb{A}$ are used to represent either tensors $\mathbb{T}$ or change-of-basis tensors $\mathbb{S}$ for the first wavefunction and tensors $\mathbb{B}$ do the same for the second wavefunction. A typical computation encountered in contracting this overlap tensor network is of the form (see Fig. 13d)

$$\rho_{\vec{\nu}}^{\vec{\mu}} = \sum_{\vec{\mu}_L, \vec{\mu}_R, \vec{\nu}_L, \vec{\nu}_R} \mathbb{A}_{\vec{\mu}_L \vec{\mu}_R}^{\vec{\mu}} \ (\rho_L)_{\vec{\nu}_L}^{\vec{\mu}_L} (\rho_R)_{\vec{\nu}_R}^{\vec{\mu}_R} \ \mathbb{B}_{\vec{\nu}_L \vec{\nu}_R}^{*\vec{\nu}} \qquad (124)$$

where now $\mathbb{A}$ and $\mathbb{B}$ are tensors $\mathbb{T}$ in Eq. (82), which have a number of non-zero coefficients given by Eq. (83). However, $\rho_L$ and $\rho_R$ are only particle-number conserving,

meaning that they mix different choices $\vec{\mu}$ and $\vec{\nu}$ with the same particle number, that is such that $|\vec{\mu}| = |\vec{\nu}|$. To obtain an upper bound for the cost of performing the above sum, we will only assume that the tensors are particle-number conserving, which leads to a computational cost $\sum_{m=0}^{M} 2^{3m} \approx 8^M$. Computing the overlap involves $L/2 + L/4 + \cdots + 2 + 1 = L - 1$ contractions of the form (124) (namely $L/2$ for the lowest layer of tensors, $L/4$ for the next layer, etc) and thus the total computational cost is upper bounded by

$$O(L 8^M). \qquad (125)$$

Again, similar expressions can be obtained for the expectation value of local observables.

### H. Homogenous tensor networks

In the above MPS and TTN representations, tensor $\mathbb{T}$ is the same in all locations (up to possible restrictions in the range of its indices), whereas tensors $\mathbb{S}[i]$ depend on the part $A_i$ of lattice $\mathcal{L}$ in which they act. In some cases, however, we can re-organize the content of the wavefunction into tensors in such a way as to be able to use copies of the same tensor throughout an entire MPS or throughout an entire layer of a regular TTN. These *homogeneous* tensor networks can be stored using less memory and their manipulation has lower computation cost. We note that being able to represent a wavefunction using a homogeneous tensor network does not imply that the wavefunction itself is translation invariant, as the shape of the tensor network (e.g. the existence of a left-most tensor and a right-most tensor in the MPS representation) breaks translation invariance explicitly. [One can still obtain a translation invariant Bethe wavefunction by fine-tuning the Bethe data with respect to the size $N$ of the 1d lattice $\mathcal{L}$, but this is independent of whether its tensor network representation is chosen to be homogeneous]. As mentioned below in more detail, our homogeneous MPS representation can be seen to coincide (up to normalization factors) with the MPS representation previously presented in Ref. [29].

We will next see that we are able to produce homogeneous MPS and TTN representations of a Bethe wavefunction when the following two conditions are fulfilled:

(1) The underlying 1d lattice $\mathcal{L}$ is regular, i.e. the position $x$ of $n^{th}$ site of $\mathcal{L}$ (where $n = 1, 2, \cdots, N$ labels the $N$ sites of $\mathcal{L}$) is equal to $\kappa n$, where the constant $\kappa$ is the lattice spacing, which for simplicity we assume to be $\kappa = 1$ from now on.

(2) The partition $\{A_i\}$ of the lattice is also regular, meaning that the number of sites $N_{A_i}$ is the same for all parts $A_i$ for $i = 1, \cdots, L$, with $N_{A_i} = N/L$. (For a regular TTN, we also need that all the effective sites in a layer of the tree be equivalent in terms of the number of original sites of the lattice that they represent). In particular, condition (2) is satisfied when the number of

partitions $L$ is the same as the number of sites $N$, which we used above in our example of MPS representation.

To get started, we will only assume that condition (1) is satisfied and use it to build an alternative bipartite decomposition of a Bethe wavefunction for a left-right bipartition $\mathcal{L} = A \cup B$ of the 1d lattice $\mathcal{L}$ made of $N$ sites, $\mathcal{L} = \{1, 2, \cdots, N\}$, where the $N_A$ and $N_B$ sites in parts $A$ and $B$ are $A = \{1, 2, \cdots, N_A\}$ and $B = \{N_A + 1, N_A + 2, \cdots, N_A + N_B = N\}$. In the previously discussed bipartite decomposition (49), the local Bethe wavefunction $|\vec{b}\rangle$ on part $B$ implicitly depends on the embedding of the $N_B$ sites of part $B$ within the lattice through the position of its particles. Concretely, $|\vec{b}\rangle$ for a choice vector $\vec{b}$ with corresponding to $M_B = |\vec{b}|$ particles is given by (see Eq. (36)),

$$|\vec{b}\rangle = \sum_{\vec{x} \in \mathcal{D}^B_{M_B}} \sum_{P \in S_{M_B}} \Theta[P] e^{i\vec{k}^{\vec{b}} \cdot \vec{x}} |\vec{x}\rangle, \qquad (126)$$

where each coordinate $x_j$ in $\vec{x} = (x_1, x_2, \cdots, x_{M_B})$ knows about the absolute position of the site in $B$ as embedded in $\mathcal{L}$, in that each position $x_j$ fulfills $N_A + 1 \leq x_j \leq N$. However, we can express the local Bethe wavefunction $|\vec{b}\rangle$ above as a product of a global phase $\Omega_A(\vec{b})$ and a *shifted* local Bethe wavefunction $|\tilde{\vec{b}}\rangle$ that only depends on the *relative* position $y \equiv x - N_A$ of the sites within $B$. Indeed, let us define the *shifted* region $\tilde{B}$ as $\tilde{B} = \{1, 2, \cdots, N_B\}$ and introduce the local Bethe wavefunction $|\tilde{\vec{b}}\rangle$ defined on $\tilde{B}$ as

$$|\tilde{\vec{b}}\rangle = \sum_{\vec{y} \in \mathcal{D}^{\tilde{B}}_{M_B}} \sum_{P \in S_{M_B}} \Theta[P] e^{i\vec{k}^{\vec{b}} \cdot \vec{y}} |\vec{y}\rangle. \qquad (127)$$

Since by definition $\vec{y} \in \mathcal{D}^{\tilde{B}}_{M_B}$ and $\vec{x} \in \mathcal{D}^B_{M_B}$ satisfy $\vec{y} = \vec{x} - N_A(1, 1, \cdots, 1)$, it follows that $|\tilde{\vec{b}}\rangle$ and $|\vec{b}\rangle$ are related by an overall phase factor,

$$|\vec{b}\rangle = \Omega_A(\vec{b})|\tilde{\vec{b}}\rangle, \qquad \Omega_A(\vec{b}) \equiv \left(\prod_{j=1}^{M_B} e^{ik_{b_j}}\right)^{N_A}. \qquad (128)$$

We thus arrive to an alternative bipartite decomposition of a Bethe wavefunction,

$$|\vec{1}\rangle = \sum_{\vec{a}, \vec{b}} \Theta[\vec{a}, \vec{b}] \, \Omega_A(\vec{b}) \, \delta_{\vec{a} \cup \vec{b}, \vec{1}} \left| \vec{a}, \tilde{\vec{b}} \right\rangle, \qquad (129)$$

which is also a sum of products of local Bethe wavefunctions as in decomposition (49) and where $|\vec{a}\rangle$ is the same local Bethe wavefunction on region $A$ as in decomposition (49), but where now $|\tilde{\vec{b}}\rangle$ is a local Bethe wavefunctions on the 'shifted' region $\tilde{B}$ that replaces $|\vec{b}\rangle$, and the amplitude $\Theta[\vec{a}, \vec{b}] \, \delta_{\vec{a} \cup \vec{b}, \vec{1}}$ in (49) has been accordingly replaced with $\Theta[\vec{a}, \vec{b}] \, \Omega_A(\vec{b}) \, \delta_{\vec{a} \cup \vec{b}, \vec{1}}$.

We can use this alternative bipartite decomposition as a basis for an alternative multipartite decomposition

$$|\vec{1}\rangle = \sum_{\vec{a}_1, \cdots, \vec{a}_L} \Theta_L[\vec{a}_1, \cdots, \vec{a}_L] \, \Omega(\vec{a}_1, \cdots, \vec{a}_L)$$
$$\times \, \delta_{\vec{a}_1 \cup \cdots \cup \vec{a}_L, \vec{1}} \left| \tilde{\vec{a}}_1, \cdots, \tilde{\vec{a}}_L \right\rangle. \quad (130)$$

where the new complex phase $\Omega(\vec{a}_1, \cdots, \vec{a}_L) = \Omega_1(\vec{a}_1)\Omega_2(\vec{a}_2)\cdots\Omega_L(\vec{a}_L)$ is the product of $L$ complex phases $\Omega_i(\vec{a}_i) \equiv \left(\prod_{j=1}^{M_{A_i}} e^{ik(\vec{a}_i)_j}\right)^{N_{A_1} + \cdots + N_{A_{i-1}}}$. More importantly for our current purposes, we also obtain alternative explicit tensor network representations. In particular, the new decomposition gives rise to new tensors $\tilde{\mathbb{T}}$ and $\tilde{\mathbb{S}}$, with components

$$\tilde{\mathbb{T}}^{\vec{c}}_{\vec{a}, \vec{b}} = \Theta[\vec{a}, \vec{b}] \, \Omega_A(\vec{b}) \, \delta_{\vec{a} \cup \vec{b}, \vec{c}}, \qquad (131)$$

$$\tilde{\mathbb{S}}^{\vec{a}_i}_{\vec{\sigma}_{A_i}} = \left\langle \vec{\sigma}_{A_i} \middle| \tilde{\vec{a}}_i \right\rangle = \sum_{R \in S_{M_{A_i}}} \Theta^{\vec{a}_i}[R] \sum_{\vec{y} \in \mathcal{D}^{\tilde{A}_i}_{M_{A_i}}} e^{i(\vec{k}^a)^R \cdot \vec{y}} \langle \vec{\sigma}_{A_i} | \vec{y} \rangle, \qquad (132)$$

where now both $\tilde{\mathbb{T}}$ and $\tilde{\mathbb{S}}$ depend on the quasi-momenta $\vec{k}$ *and* the two-particle scattering angles $\boldsymbol{\theta}$. The motivation to introduce this alternative formulation becomes clear if condition (2) is also fulfilled, namely if the parts $A_i$ in a multipartition of lattice $\mathcal{L}$ are all of the same size (meaning that $N_{A_i}$ is independent of $i$). Then for an MPS and a regular TTN, both tensor $\tilde{\mathbb{T}}$ and tensor $\tilde{\mathbb{S}}$ above are independent of the region(s) on which they act. Indeed, on the one hand $\Omega_A(\vec{b})$ in Eq. (128) only depends on region $A$ through the number of sites $N_A$, so by condition (2) $\Omega_{A_i}(\vec{b})$ is the same for all choices of part $A_i$, since $N_{A_i}$ is constant. Then tensor $\tilde{\mathbb{T}}$ in Eq. (131), where $\vec{a}$ and $\vec{b}$ refer to two consecutive parts $A = A_i$ and $B = A_{i+1}$, is also independent of $A_i$. On the other hand, the shifted local Bethe wavefunction $\left| \tilde{\vec{a}}_i \right\rangle$ for a region $A_i$ again only depends on the choice $\vec{a}_i$ and the number of sites $N_{A_i}$ in part $A_i$. Since by condition (2) $N_{A_i}$ does not depend on $i$, the overlap $\left\langle \vec{\sigma}_{A_i} \middle| \tilde{\vec{a}}_i \right\rangle$ depends both on the bitstring $\vec{\sigma}_{A_i}$ and choice $\vec{a}_i$, but not on the part $A_i$, so tensor $\tilde{\mathbb{S}}$ is the same for all parts $A_i$.

*MPS*

Let us first apply the above considerations to an MPS. Repeating the construction of an MPS in Sect. V D but using the alternative bipartite decomposition, one can see that if conditions (1) and (2) are met, then we can fully characterize the wavefunction in terms of a single pair of tensors $(\tilde{\mathbb{T}}, \tilde{\mathbb{S}})$ or, equivalently, in terms of a single MPS tensor $\tilde{\mathbb{R}}$ defined as

$$\tilde{\mathbb{R}}^{\vec{\mu}_L, \vec{\mu}_R}_{\vec{\sigma}_{A_i}} \equiv \sum_{\vec{a}_i} \tilde{\mathbb{T}}^{\vec{\mu}_L}_{\vec{a}_i, \vec{\mu}_R} \tilde{\mathbb{S}}^{\vec{a}_i}_{\vec{\sigma}_{A_i}}, \qquad (133)$$

also independent of part $A_i$. For example, for an $M-$particle Bethe wavefunction and a partition made of $L = N$ parts with each part $A_i$ consisting of a single site of the 1d lattice $\mathcal{L}$, instead of the site-dependent tensor $\mathbb{R}[x]$ of Eq. (103) we obtain the site-independent tensor $\tilde{\mathbb{R}}$ with components

$$\tilde{\mathbb{R}}_0^{\vec{\mu}\vec{\mu}'} = \delta_{\vec{\mu},\vec{\mu}'} \prod_{j \in \vec{\mu}} e^{ik_j}, \quad (134)$$

$$\tilde{\mathbb{R}}_1^{\vec{\mu}\vec{\mu}'} = \sum_{j=1}^{M} \delta_{\vec{\mu},\vec{\mu}' \cup (j)} \, \Omega_{A_i}(\vec{\mu}')\Theta[(j),\vec{\mu}'] \, e^{ik_j}. \quad (135)$$

Specifically, for $M = 0$ particles it reads

$$\tilde{\mathbb{R}}_0 = \begin{pmatrix} 1 \end{pmatrix}, \quad \tilde{\mathbb{R}}_1 = \begin{pmatrix} 0 \end{pmatrix}; \quad (136)$$

for $M = 1$ particles we have

$$\tilde{\mathbb{R}}_0 = \begin{pmatrix} 1 & 0 \\ 0 & e^{ik} \end{pmatrix}, \quad \tilde{\mathbb{R}}_1 = \begin{pmatrix} 0 & 0 \\ e^{ik} & 0 \end{pmatrix}; \quad (137)$$

for $M = 2$ particles we have

$$\tilde{\mathbb{R}}_0 = \begin{pmatrix} 1 & 0 & 0 & 0 \\ 0 & e^{ik_1} & 0 & 0 \\ 0 & 0 & e^{ik_2} & 0 \\ 0 & 0 & 0 & e^{ik_2}e^{ik_1} \end{pmatrix}, \quad (138)$$

$$\tilde{\mathbb{R}}_1 = \begin{pmatrix} 0 & 0 & 0 & 0 \\ e^{ik_1} & 0 & 0 & 0 \\ e^{ik_2} & 0 & 0 & 0 \\ 0 & -e^{i\theta_{21}}e^{ik_2}e^{ik_1} & e^{ik_1}e^{ik_2} & 0 \end{pmatrix}, \quad (139)$$

and so on. We reiterate that in this MPS representation, the $N$ tensors are all copies of the same tensor $\tilde{\mathbb{R}}$, but the first (left-most) copy of $\tilde{\mathbb{R}}$ has its left bond index $\vec{\mu}$ set to $\vec{1}$ and the last (right-most) copy of $\tilde{\mathbb{R}}$ has its right bond index $\vec{\mu}'$ set to $\emptyset$. Importantly, the overlap $\left\langle \Phi_M^{\vec{k},\boldsymbol{\theta}} \middle| \Phi_M^{\vec{k}',\boldsymbol{\theta}'} \right\rangle$ of two Bethe wavefunctions each represented by a homogeneous MPS can now be computed by studying the eigenvalue decomposition of the mixed *transfer matrix* $\sum_i \tilde{\mathbb{A}}_i^{\vec{\mu}\vec{\mu}'} \cdot \tilde{\mathbb{B}}_i^{*\vec{\nu}\vec{\nu}'}$ instead of having to iterate $N$ times the product in Eq. (114). Particle number conservation implies that the above transfer matrix can be numerically brought into a block diagonal form (where each block is an upper-triangular Schur matrix). This leads to the evaluation of the scalar product with a cost that is independent of the number $L$ of parts or, equivalently, the number $N$ of sites of lattice $\mathcal{L}$. This is a significant reduction compared to the linear cost in Eq. (123) for non-homogeneous MPS. Similar manipulations are possible for the computation of expectation value of local operators.

We note that matrices $\tilde{\mathbb{R}}_0$ and $\tilde{\mathbb{R}}_1$ in Eqs.(138)-(139) above correspond, up to normalization and notational variations, to matrices $\Lambda^0$ and $\Lambda^1$ right after Eq. 116 and in Eq. 119 of Ref. [29]. Thus our formalism for a generic planar TTN representation recovered, when applied to a homogeneous MPS, the MPS representation previously proposed in Ref. [29].

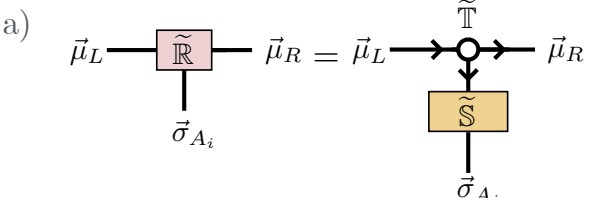

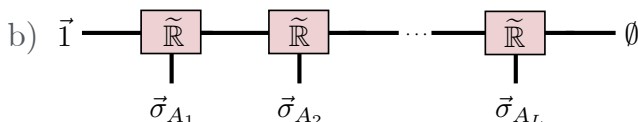

FIG. 14. Homogenous MPS for Bethe wavefunctions

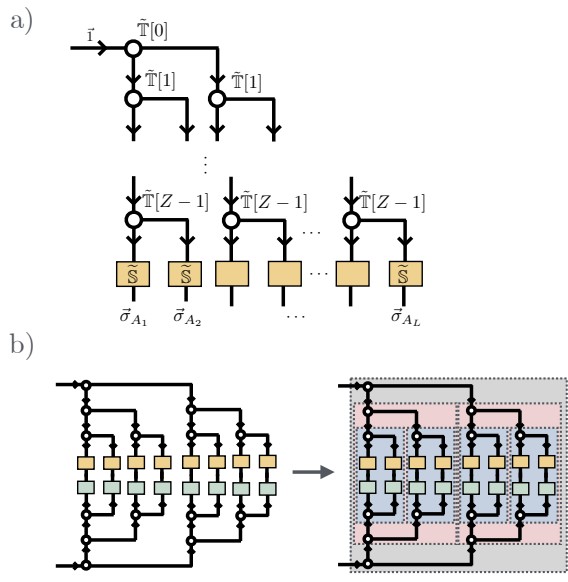

FIG. 15. (a) Scale-dependent, homogeneous TTN for Bethe wavefunctions, made of $Z \sim \log(N)$ layers of tensors. Layer $z$ of this regular binary TTN is characterized by copies of the same tensor $\tilde{\mathbb{T}}[z]$ (for $z = 0, 1, \cdots, Z - 1$) or change-of-basis tensor $\tilde{\mathbb{S}}$ (for $z = Z$ at the bottom). (b) The overlap of two homogeneous TTN representations can be computed hierarchically, starting at scale $z = Z$ and moving towards scale $z = Z - 1, \cdots, 1, 0$. The contraction of the sub-trees (colored rectangles) can be performed only once per layer due to the homogeneity of the TTN. This reduces the complexity of computing the overlaps to $O(\log N)$.

*TTN*

Next we repeat the derivation of a regular binary TTN representation in Sect. V E for a 1d lattice $\mathcal{L}$ divided into $L = 2^Z$ parts, but now using the alternative bipartite decomposition. One can see that, if conditions (1) and (2) are met, then we end up with a TTN representation

made of a set of scale dependent tensors $\tilde{\mathbb{T}}[z]$

$$\tilde{\mathbb{T}}[z]^{\vec{c}}_{\vec{a},\vec{b}} = \Theta(\vec{a},\vec{b}) \prod_{j=1}^{|\vec{b}|} e^{ik_{b_j} N/2^z} \delta(\vec{c},\vec{a}\cup\vec{b}), \qquad (140)$$

for $z = 0, 1, \cdots, L-1$ together with the change-of-basis tensor $\tilde{\mathbb{S}}$ in Eq. (132) at scale $z = Z$, all of which are independent of position, meaning that layer $z$ of the TTN consists of $2^z$ copies of the same tensor (either $\tilde{\mathbb{T}}[z]$ for $z = 0, 1, \cdots, Z-1$ or $\tilde{\mathbb{S}}$ for $z = Z$). That is, we obtain a TTN that is homogeneous within each layer, see Fig. 15a. Note that tensor $\tilde{\mathbb{T}}[z]$ depends on the scale $z$ because the size (in terms of the number of sites in $\mathbb{L}$) of each part at scale $z$ is $N/2^z$, and thus not constant. As with the homogeneous MPS representation, we emphasize that this homogeneous TTN representation does not imply that the Bethe wavefunction it encodes is itself translation invariant.

The homogeneous TTN representation only requires storing $Z = \log_2 L$ tensors in memory (instead of $O(L)$ tensors in the non-homogeneous case). In addition, the cost of computing the overlap $\left\langle \Phi_M^{\vec{k},\boldsymbol{\theta}} \middle| \Phi_M^{\vec{k}',\boldsymbol{\theta}'} \right\rangle$ of two Bethe wavefunctions each represented by a homogeneous TTN is significantly reduced to

$$O\left( \log_2(L) \sum_{m=0}^{M} \binom{M}{m}^3 \right), \qquad (141)$$

see Fig. 15b, in contrast to the space dependent TTN introduced earlier, with cost proportional to the number $L$ of parts, Eq. (125). This reduction is possible because the overlap of the wavefunctions can be done hierarchically, starting from the lowest layer. Since tensors are homogenous at each scale, we need to perform only one computation to contract the sub-tree per layer (see Fig. 15b). One can obtain similar reductions in the computation of expectation value of local observables.

## VI. BETHE QUANTUM CIRCUITS

In the previous section we have discussed how to exactly represent an $M-$particle Bethe wavefunction on a 1d lattice $\mathcal{L}$ made of $N$ sites using a planar tree tensor network, including an MPS and a regular binary TTN as particular cases. For certain tensor networks, one can transform the tensor network representation into a quantum circuit that maps an initial unentangled state into the Bethe wavefunction, thus paving the way for the experimentally preparation of the Bethe wavefunction on a quantum computer [29, 30, 35–38]. Prior works have highlighted the usefulness of the MPS representation of such wavefunctions [38], which naturally leads to a sequential unitary quantum circuit of depth $O(N)$, that is, depth proportional to the lattice size $N$ [21]. Other deterministic approaches to prepare a generic $M-$particle wavefunction on $N$ sites require depth $O(\binom{N}{M})$ [30], which

scales as $O(N^M)$, that is, polynomially in the system size $N$.

In this section, we show that our TTN representation leads to a different quantum circuit with depth $O(\log(N))$, just logarithmic with the system size $N$, which may therefore be advantageous over previous circuits based on an MPS representation. For concreteness, we consider the scale-dependent, homogeneous TTN representation of a Bethe wavefunction described in Sec. V H, for a partition of lattice $\mathcal{L}$ into $L = N/M$ parts each made of $M$ contiguous sites. As before, we assume the number of parts to be $L = 2^Z$ (equivalently, the number of sites $N = M2^Z$). These conditions can be relaxed easily.

From now on we refer ti a set of $M$ sites (or $M$ qubits) as a qudit of dimension $2^M$. Our first goal is to transform the TTN representation in Fig. 15a, which is written in terms of tensors $\tilde{\mathbb{T}}[z]$ and $\tilde{\mathbb{S}}$, into its canonical form [18], which is written in terms of isometric tensors $\mathbb{W}[z]$, see Fig. 16a. Then we will transform each isometric tensor $\mathbb{W}[z]$ into unitary gates $\mathbb{U}[z]$ and $\tilde{\mathbb{U}}[z]$), see Fig. 16b. Fig. 16c shows the complete transformation from the initial TTN to the final quantum circuit.

## A. Canonical form of the TTN representation

Recall that the QR decomposition of an $m \times n$ matrix $F$ (with $m \geq n$) is the product $F = WS$ of an isometric $m \times n$ matrix $W$ (meaning that $W$ satisfies $W^\dagger W = \mathbb{I}$) and an $n \times n$ upper-triangular matrix $S$. The computational cost is $O(mn^2)$.

Consider now a tensor $\mathbb{F} : \mathbf{C}_{2^M} \to \mathbf{C}_{2^M} \otimes \mathbf{C}_{2^M}$ with components $\mathbb{F}_{\beta\gamma}^\alpha$, which maps states of one qudit into states of two qudits. We can apply the QR decomposition to tensor $\mathbb{F}$ by regarding it as a matrix $F$ with components $F_{(\beta\times\gamma),\alpha} = \mathbb{F}_{\beta\gamma}^\alpha$ and QR-decomposing it as $F_{(\beta\times\gamma),\alpha} = \sum_{\alpha'} W_{(\beta\times\gamma),\alpha'} S_{\alpha'\alpha}$. The final result is the decomposition

$$\mathbb{F}_{\beta\gamma}^\alpha = \sum_{\alpha'} \mathbb{W}_{\beta\gamma}^{\alpha'} \, \mathbb{S}_{\alpha'}^\alpha, \qquad (142)$$

where $\mathbb{W} : \mathbf{C}_{2^M} \to \mathbf{C}_{2^M} \otimes \mathbf{C}_{2^M}$ is an isometric tensor, meaning that $\sum_{\beta\gamma} \mathbb{W}_{\beta\gamma}^\alpha (\mathbb{W}^*)_{\beta\gamma}^{\alpha'} = \delta_{\alpha,\alpha'}$, and $\mathbb{S} : \mathbf{C}_{2^M} \to \mathbf{C}_{2^M}$ is a one-qudit linear map. The cost of this QR decomposition scales as $O(2^{4M})$.

To bring the TTN in Fig. 15a into its canonical form, we start at the bottom layer of the tree and apply the transformation outlined in Fig. 16a, namely we first multiply tensor $\tilde{\mathbb{T}}[Z-1]$, with components $\tilde{\mathbb{T}}[Z-1]_{\vec{a}_L \vec{a}_R}^{\vec{\mu}}$, with two copies of the change-of-basis tensor $\tilde{\mathbb{S}}$ (denoted $\mathbb{S}_L$ and $\mathbb{S}_R$ in the figure), with components $\tilde{\mathbb{S}}_{\vec{\sigma}_L}^{\vec{a}_L}$ and $\tilde{\mathbb{S}}_{\vec{\sigma}_R}^{\vec{a}_R}$, by contracting the indices $\vec{a}_L$ and $\vec{a}_R$, which results in an intermediate tensor $\mathbb{F}[Z-1]$. We then QR decompose

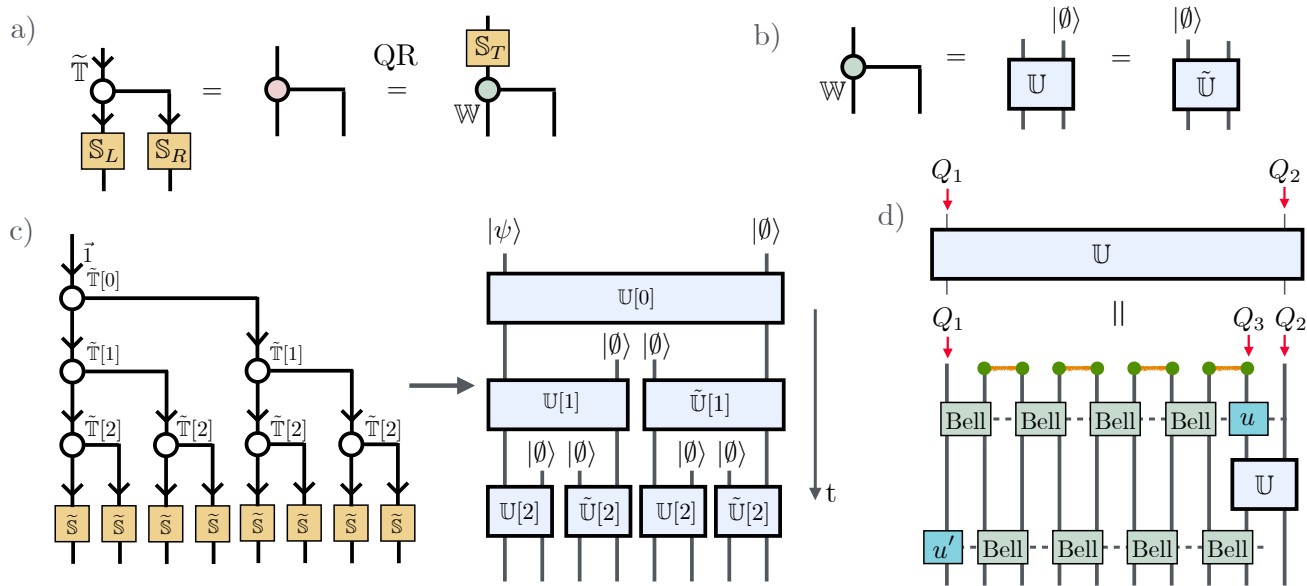

FIG. 16. (a) Transformation used to bring a TTN into its canonical form using a QR decomposition. Using this transformation, we can bring a layer of the TTN into an isometric form. Notice that the resulting tensor $\mathbb{W}$ is isometric and matrix $\mathbb{S}_T$ is fed as either $\mathbb{S}_L$ or $\mathbb{S}_R$ for the next iteration of this transformation on the layer above. (b) The isometric tensor $\mathbb{W}$ can be regarded as a unitary tensor (either $\mathbb{U}$ or $\tilde{\mathbb{U}}$) with one incoming index set to the empty/vacuum state $|\emptyset\rangle$ of a qudit. (c) After sequentially applying trasformations (a) and (b) above to each layer of the TTN starting from the bottom one and moving upwards, the original TTN is mapped into a quantum circuits with long-range gates. (d) A long-range, two-qudit gate between qudits $Q1$ and $Q2$ can be implemented with a number of nearest-neighbor two-qudit gates that scales proportional to the distance between $Q1$ and $Q2$. This involves preparing pairs of nearest-neighbor qudit Bell pairs, applying qudit Bell measurements, and single-qudit unitaries that are conditioned on the measurement outcomes. The two circles connected by a wide line indicate a Bell pair, the 'Bell' boxes refer to measurements in the 2 qudit Bell basis, and the dashed horizontal line indicates that the single-qudit unitary $u$ or $u'$ must be conditioned on the measurement outcomes, which requires classical communication between distant parts of the circuit.

$\mathbb{F}[Z-1]$ into tensors $\mathbb{W}[Z-1]$ and $\mathbb{S}[Z-1]$, namely

$$\sum_{\vec{a}_L, \vec{a}_R} \tilde{\mathbb{S}}^{\vec{a}_L}_{\vec{\sigma}_L} \; \tilde{\mathbb{S}}^{\vec{a}_R}_{\vec{\sigma}_R} \; \tilde{\mathbb{T}}[Z-1]^{\vec{\mu}}_{\vec{a}_L \vec{a}_R} = \mathbb{F}[Z-1]^{\vec{\mu}}_{\vec{\sigma}_L \vec{\sigma}_R} \quad (143)$$

$$= \sum_{\alpha'} \mathbb{W}[Z-1]^{\alpha'}_{\beta\gamma} \; \mathbb{S}[Z-1]^{\alpha}_{\alpha'}, \quad (144)$$

or, using more compact notation,

$$\tilde{\mathbb{S}} \, \tilde{\mathbb{S}} \, \tilde{\mathbb{T}}[Z-1] = \mathbb{F}[Z-1] = \mathbb{W}[Z-1] \, \mathbb{S}[Z-1]. \quad (145)$$

Notice that since the TTN is homogeneous, we only need to apply this transformation once for the entire lowest layer of the TTN. For the next layer of the TTN, we again apply the transformation outlined in Fig. 16a, namely we multiply two copies of $\mathbb{S}[Z-1]$ with a copy of $\tilde{\mathbb{T}}[Z-2]$, creating an intermediate tensor $\mathbb{F}[Z-2]$, which we then QR decompose into tensors $\mathbb{W}[Z-2]$ and $\mathbb{S}[Z-2]$, which we write in compact form as

$$\mathbb{S}[Z-1] \, \mathbb{S}[Z-1] \, \tilde{\mathbb{T}}[Z-2] \;=\; \mathbb{F}[Z-2]$$
$$= \; \mathbb{W}[Z-2] \, \mathbb{S}[Z-2]. \quad (146)$$

We represent this transformation in Fig. 16a. After applying such transformation to each layer of the TTN

from its bottom to its top, we are left with a new scale-dependent, homogeneous TTN representation in terms of isometric tensors $\mathbb{W}[z]$, and a top vector $|\psi\rangle$ that results from setting the incoming index of the top matrix $\mathbb{S}[z = 0]$ to $\vec{1}$.

## B. Quantum circuit from the TTN representation

Recall that an isometry $\mathbb{W} : \mathbf{C}_{2^M} \to \mathbf{C}_{2^M} \otimes \mathbf{C}_{2^M}$ from one qudit to two qudits can be augmented into a two-qudit unitary tensor $\mathbb{U} : \mathbf{C}_{2^M} \otimes \mathbf{C}_{2^M} \to \mathbf{C}_{2^M} \otimes \mathbf{C}_{2^M}$, with components $\mathbb{U}^{\alpha\delta}_{\beta\gamma}$ such that $\mathbb{U}^{\alpha\delta=0}_{\beta\gamma} = \mathbb{W}^{\alpha}_{\beta\gamma}$. We can think of $\mathbb{W}$ as the result of multiplying the vacuum state $|\emptyset\rangle \equiv |0\rangle^{\otimes M} \in \mathbf{C}_{2^M}$ of a qudit (that is, of $M$ qubits) with the new incoming index of $\mathbb{U}$, namely $\mathbb{W} = \mathbb{U}(I_{2^M} \otimes |\emptyset\rangle)$. Alternatively, we can build a unitary $\tilde{\mathbb{U}} : \mathbf{C}_{2^M} \otimes \mathbf{C}_{2^M} \to \mathbf{C}_{2^M} \otimes \mathbf{C}_{2^M}$ with components $\tilde{\mathbb{U}}^{\delta\alpha}_{\beta\gamma}$ such that $\tilde{\mathbb{U}}^{\delta=0,\alpha}_{\beta\gamma} = \mathbb{W}^{\alpha}_{\beta\gamma}$, with $\mathbb{W} = \tilde{\mathbb{U}}(|\emptyset\rangle \otimes I_{2^M})$.

Accordingly, we replace each isometric tensor $\mathbb{W}[z]$ by a unitary tensor $\mathbb{U}[z]$ with state $|\emptyset\rangle$ acting on its incoming second index, or by a unitary tensor $\tilde{\mathbb{U}}[z]$ with state $|\emptyset\rangle$

acting on its first incoming index, see Fig. 16b. The net result is the quantum circuit depicted in Fig. 16c.

This quantum circuit, made of unitary gates acting on pairs of qudits, has depth $Z = \log_2 L = \log_2(N/M)$, which is exponentially shorter than the depth of circuits based on an MPS. However, it involves long-range gates, which may be perceived as off-setting any gains due to the circuit's logarithmic depth. Fortunately, it is well-known how to use mid-circuit measurements and short-range entangled ancillary systems to perform these long-range gates in constant depth [41]. This strategy is sketched in Fig. 16d and reviewed next.

Consider a two-qudit gate $\mathbb{U}[z]$ [a similar derivation applies to $\tilde{\mathbb{U}}[z]$] acting on qudits $Q_1$ (with input state $|\psi\rangle$) and $Q_2$ (with input state $|\emptyset\rangle$), which are $L/2^{z-1} - 1$ qudits apart. To perform this gate in constant depth, we introduce a chain of $L/2^{z-1}$ qudits initialized in alternating nearest neighboring qudit Bell pairs (indicated as a bond in Fig. 16c). The quantum state on qudit $Q_1$ can be teleported to the rightmost qudit ($Q_3$) of the chain by alternating two-qudit measurements in the Bell basis, followed by a single qubit unitary $u$ conditioned on the measurement outcomes (indicated by dashed horizontal line in Fig. 16c). Next one performs the nearest neighbor two-qudit gate $\mathbb{U}[z]$ on qudits $Q3$ and $Q2$. The state of qudit $Q3$ then has to be teleported back into the original qudit $Q_1$ by using the same chain of qudits, which is now in a shifted alternating Bell pair state after the previous round of measurements. To teleport the state we again perform alternating two-qudit Bell measurements and a conditional single qudit unitary $u'$.

Above we have provided a local circuit for $2^M$ dimensional qudits. For a qubit-based local circuit, we need to decompose each gate above into a subcircuit of two-qubit gates; however the depth of each gate decomposition does not scale with $N$. In particular, each two-qudit Bell measurement can be compiled as $M$ two-qubit Bell measurements along with the requisite swap gates. This protocol performs the long-range gate with a constant overhead, at the cost of non-local classical communication and adaptive quantum gates. [For simplicity, we applied the long-range gate protocol to $\mathbb{U}[z]$ in its standard form, which works for any input states. Notice that we could have saved the initial teleportation round by exploiting that one of the inputs of the non-local gate $\mathbb{U}[z]$ is the state $|\emptyset\rangle$, although this does not alter the scaling of the circuit's depth with the lattice size $N$.]

The TTN circuit is particularly useful for fixed $M$, when $N$ is large. Each scale $z$ requires $O(N)$ two-qubit gates, and therefore it takes $O(N \log(N))$ two-qubit gates to prepare such states, arranged in a circuit of depth $O(\log(N))$. This provides an exponential advantage in depth for constant $M$, compared to the previous strategies for exact preparation of such states using a circuit of depth $O(N)$.

Note, the advantage of our circuit over MPS-based circuits is comparable to that in recent work [25] showing that generic matrix product states can be approxi-

mately prepared using adaptive (measurement-controlled unitaries) protocols with very short depth circuits. That proposal essentially amounts to transforming the MPS representation into an approximate TTN representation.

## VII. GENERALIZED BETHE WAVEFUNCTIONS

In this section we introduce a generalization of the Bethe wavefunction, dubbed *generalized Bethe wavefunction*, which we build such that it also accepts the multipartite decompositions, and tensor network and quantum circuit representations for Bethe wavefunctions that we have derived in this work. This generalization is possible by identifying the specific properties of a Bethe wavefunction that were essential in our preceding derivations. Our generalized Bethe wavefunction, which depends on a number of parameters that scales in $N$ as $O(MN)$, includes as a particular case the inhomogeneous Bethe wavefunctions (which depend on $M + N$ parameters) recently and independently proposed in Ref. [31], and can be seen to actually equate that proposal if we omit the integrability constraints imposed in Ref. [31].

Let us start by briefly recalling how an $M$-particle Bethe wavefunction is constructed, see Sec. II for more details. It is built with two ingredients:

$(i)$ a set of $M$ single-particle wavefunctions, namely $M$ single-particle plane waves with amplitudes $\{e^{ik_j x}\}$ for $1 \le j \le M$, as specified by $M$ quasi-momenta $\vec{k} = (k_1, k_2, \cdots, k_M)$, and

$(ii)$ a scattering amplitude $\Theta[P]$ (where $P = (P_1, P_2, \cdots, P_M)$ is a permutation of the $M$ symbols representing the above $M$ single-particle plane waves) that factorizes as the product of two-symbol scattering amplitudes $-e^{i\theta_{j_2 j_1}}$. The latter are specified in terms of $M(M-1)/2$ scattering angles $\boldsymbol{\theta} = \{\theta_{j_2 j_1}\}$, with $\theta_{j_2 j_1} \in [0, 2\pi)$ and $1 \le j_1 < j_2 \le M$.

A possible recipe to build a Bethe wavefunction with these ingredients starts with the introduction, given the vector $\vec{k} = (k_1, k_2, \cdots, k_M)$ of quasi-momenta, of a *spatially-ordered* $M-$particle plane-wave state, namely

$$|I\rangle \equiv \sum_{\vec{x} \in \mathcal{D}_M^{\mathcal{L}}} e^{ik_1 x_1} e^{ik_2 x_2} \cdots e^{ik_M x_M} |x_1 x_2 \cdots x_M\rangle, \quad (147)$$

which represents a system of $M$ distinct particles, labelled by $j = 1, 2, \cdots, M$. , Particle $j$, in position $x_j$, is constrained to be spatially located between particle $j-1$ (in position $x_{j-1}$) to its left and particle $j+1$ (in position $x_{j+1}$) to its right, and is in the $j$th plane-wave state (with amplitudes $e^{ik_j x_j}$). One can thus think of wavefunction $|I\rangle$ as corresponding to $M$ particles, properly ordered from left to right in lattice $\mathcal{L}$, that propagate as free particles except for the constraint that they cannot hope over each other and alter their relative order in $\mathcal{L}$ (e.g. particle $j = 1$ is always to the left of the rest of particles). More generally, we introduce $M!$ different such

spatially-ordered $M-$particle wavefunctions, each characterized by a permutation $P \in \mathcal{S}_M$,

$$|P\rangle \equiv \sum_{\vec{x} \in \mathcal{D}_M^{\mathcal{L}}} e^{ik_{P_1}x_1} \, e^{ik_{P_2}x_2} \cdots e^{ik_{P_M}x_M} \, |x_1 x_2 \cdots x_M\rangle , \tag{148}$$

which corresponds to now having changed the ordering of the $M$ distinct particles from $(1, 2, \cdots, M)$ to $(P_1, P_2, \cdots, P_M)$ (e.g. particle $j = P_1$ is always to the left of the rest of particles).

Then the $M-$particle Bethe wavefunction is finally obtained as a linear combination of such spatially-ordered $M-$particle wavefunctions,

$$\left| \Phi_M^{\vec{k}, \boldsymbol{\theta}} \right\rangle = \sum_{P \in \mathcal{S}_M} \Theta[P] \, |P\rangle . \tag{149}$$

In this linear combination, the $M$ particles can now be thought of as appearing in any order in the lattice (e.g. the left-most particle can correspond to any value of $j = 1, 2, \cdots, M$). In addition, there is a scattering amplitude $\Theta[P]$ that only depends on the order of the $M$ particles and not on their specific positions. We could think that this phase occurs as the result of having two particles in nearest neighbor position exchange their locations.

Our generalized Bethe wavefunctions consider also two ingredients, which generalize the ones above:

*(i)* a set of $M$ *completely general* single-particle wavefunctions $\{|\varphi_j\rangle\}$ (as opposed to plane-wave states), where

$$|\varphi_j\rangle = \sum_{x \in \mathcal{L}} \varphi_{jx} \, |x\rangle , \qquad j = 1, 2, \cdots, M, \tag{150}$$

as specified by an $M \times N$ matrix $\boldsymbol{\varphi}$ with complex coefficients $\varphi_{jx}$, and

*(ii)* a scattering amplitude $\Theta[P]$ as in Eq. (8), which factorizes as the product of two-symbol scattering amplitudes $-e^{i\theta_{j_2 j_1}}$ depending on $M(M-1)/2$ generalized scattering angles $\boldsymbol{\theta} = \theta_{j_2 j_1}$, with $1 \leq j_1 < j_2 \leq M$, where now the angle $\theta_{j_2 j_1}$ can be complex, in such a way that the amplitude $-e^{i\theta_{j_2 j_1}} \in \mathbb{C}$ is an arbitrary complex number (as opposed to being restricted to being a complex phase).

We then build a generalized Bethe wavefunction from these ingredients in close analogy as how we built a regular Bethe wavefunction. We again introduce a *spatially-ordered* $M-$particle wavefunction

$$|I\rangle \equiv \sum_{\vec{x} \in \mathcal{D}_M^{\mathcal{L}}} \varphi_{1x_1} \, \varphi_{2x_2} \cdots \varphi_{Mx_M} \, |x_1 x_2 \cdots x_M\rangle \tag{151}$$

which represent $M$ particles $j = 1, 2, \cdots, M$ spatially ordered from left to right in lattice $\mathcal{L}$, where particle $j$ is in the single-particle wavefunction $|\varphi_j\rangle$, in the sense that the amplitude for the position basis vector $|\vec{x}\rangle$ has a multiplicative factor $\varphi_{jx_j}$ when that particle is in position $x_j$, but particle $j$ still needs to be to the right of particle $j-1$ and to the left of particle $j+1$. We can again defined

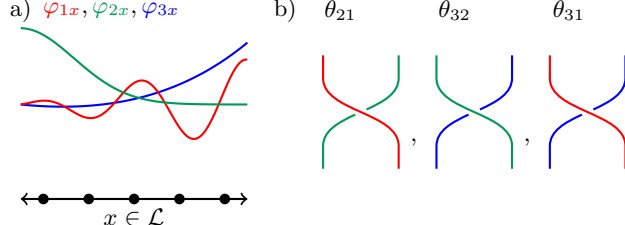

a) $\varphi_{1x}, \varphi_{2x}, \varphi_{3x}$     b)   $\theta_{21}$    $\theta_{32}$    $\theta_{31}$

$x \in \mathcal{L}$

FIG. 17. An $M-$particle generalized Bethe wavefunction on a 1d lattice $\mathcal{L}$ is described by $M$ single-particle wavefunctions $\{|\varphi_j\rangle\}$ (not necessarily plane waves) and $M(M-1)/2$ scattering angles $\theta_{j_2 j_1}$ (possibly complex). In this example, $M = 3$. This is to be compared with Fig. 4 for a standard Bethe wavefunction.

$M!$ spatially-ordered $M-$particle wavefunctions

$$|P\rangle \equiv \sum_{\vec{x} \in \mathcal{D}_M^{\mathcal{L}}} \varphi_{P_1 x_1} \, \varphi_{P_2 x_2} \cdots \varphi_{P_M x_M} \, |x_1 x_2 \cdots x_M\rangle \tag{152}$$

where the order of the $M$ particles is determined by the permutation $P$, and finally define our generalized Bethe wavefunction as the linear combination

$$\left| \Phi_M^{\boldsymbol{\varphi}, \boldsymbol{\theta}} \right\rangle = \sum_{P \in \mathcal{S}_M} \Theta[P] \, |P\rangle , \tag{153}$$

where particles are no longer restricted to appear in spatial order and $\Theta[P]$ can be interpreted as contributing a generalized scattering amplitude when the order of two neighboring particles is exchanged.

Our definition of this generalized class of wavefunctions is simply a rewrite of Eq. (153).

**Definition VII.1.** *Generalized Bethe wavefunctions: Given an $M \times N$ matrix $\boldsymbol{\varphi} \equiv \varphi_{jx}$ representing $M$ single-particle wavefunctions on lattice $\mathcal{L}$ and $M(M-1)/2$ (complex) scattering angles $\boldsymbol{\theta} \equiv \{\theta_{j_2 j_1}\}$ for $1 \leq j_1 < j_2 \leq M$, an $M-$particle generalized Bethe wavefunction on lattice $\mathcal{L}$ is defined as,*

$$\left| \Phi_M^{\boldsymbol{\varphi}, \boldsymbol{\theta}} \right\rangle \equiv \sum_{\vec{x} \in \mathcal{D}_M^{\mathcal{L}}} \sum_{P \in S_M} \Theta[P] \left( \prod_{j=1}^{M} \varphi_{P_j x_j} \right) |\vec{x}\rangle , \tag{154}$$

*where the complex coefficient $\Theta[P]$ are defined as in Eq. (8) for the Bethe wavefunctions (see Fig. 17).*

Note, the Bethe wavefunctions are a subclass of the above generalized Bethe wavefunctions, where the single particle states are plane-wave states and the scattering angles are real. These generalized Bethe wavefunctions can also be obtained from the inhomogeneous Bethe wavefunctions recently proposed Eq. 3.25 of Ref. [31], by omitting the parameterization in Eqs. 3.13 and 3.26 in that reference (which was used to ensure that the Yang-Baxter equation of integrability holds).

As it can be readily verified by inspecting the corresponding proofs or adjacent derivations, most of the results for multipartite decompositions and tensor network

and quantum circuit representations presented in this paper can be extended to a generalize Bethe wavefunction, including Theorems III.1 and IV.1 for the left-right bipartite and multipartite decomposition as a sum of $2^M$ and $L^M$ products of local Bethe wavefunctions, Theorem IV.2 for the contiguous multipartite decomposition, and Theorems V.1, V.2, and V.3 for exact MPS, regular binary TTN and a generic (planar) TTN representations with bond dimension $2^M$. An important exception is that we can no longer produce a homogeneous MPS or TTN representation as we did in Sect. V H, since general single-particle wavefunctions lead to change-of-basis tensors $\mathbb{S}[i]$ that necessarily depend on their position in the network].

We briefly exemplify these generalizations. Let us first consider the left-right bipartite decomposition in Theorem III.1. Eq. (38) becomes

$$
\left|\Phi_M^{\boldsymbol{\varphi},\boldsymbol{\theta}}\right\rangle \equiv \sum_{\vec{x}\in\mathcal{D}_M^{\mathcal{L}}} \sum_{P\in S_M} \Theta[P] \left(\prod_{j=1}^{M}\varphi_{P_j x_j}\right)|\vec{x}\rangle
$$

$$
= \sum_{m=0}^{M} \sum_{\vec{a}\in\mathcal{C}_m^M} \Theta[Q^{\vec{a},\vec{b}}] \quad \times
$$

$$
\left(\sum_{\vec{x}_A\in\mathcal{D}_{M_A}^{A}} \sum_{R\in\mathcal{S}_A} \Theta^{\vec{a}}[R]\left(\prod_{j\in\vec{a}}\varphi_{P_j x_j}\right)|\vec{x}_A\rangle\right) \times
$$

$$
\left(\sum_{\vec{x}_B\in\mathcal{D}_{M_B}^{B}} \sum_{S\in\mathcal{S}_B} \Theta^{\vec{b}}[S]\left(\prod_{j\in\vec{b}}\varphi_{P_j x_j}\right)|\vec{x}_B\rangle\right)
$$

$$
= \sum_{m=0}^{M} \sum_{\vec{a}\in\mathcal{C}_m^M} \Theta\left[Q^{\vec{a},\vec{b}}\right]\left|\Phi_m^{\boldsymbol{\varphi}_A^{\vec{a}},\boldsymbol{\theta}^{\vec{a}}}\right\rangle_A\left|\Phi_{m'}^{\boldsymbol{\varphi}_B^{\vec{b}},\boldsymbol{\theta}^{\vec{b}}}\right\rangle_B ,\text{(155)}
$$

where the only changes are that we replaced the plane-wave amplitudes $e^{i(\vec{k}^{\vec{a}})^R\cdot\vec{x}_A}$ and $e^{i(\vec{k}^{\vec{b}})^S\cdot\vec{x}_B}$ with the more general amplitudes $\prod_{j\in\vec{a}}\varphi_{R_j x_j}$ and $\prod_{j\in\vec{b}}\varphi_{S_j x_j}$ and, accordingly, the generalized local Bethe wavefunctions in parts $A$ and $B$ read

$$
\left|\Phi_m^{\boldsymbol{\varphi}_A^{\vec{a}},\boldsymbol{\theta}^{\vec{a}}}\right\rangle \equiv \sum_{\vec{x}_A\in\mathcal{D}_m^{A}} \sum_{R\in\mathcal{S}_A} \Theta^{\vec{a}}[R]\left(\prod_{j\in\vec{a}}\varphi_{R_j x_j}\right)|\vec{x}_A\rangle ,
$$

$$
\left|\Phi_{m'}^{\boldsymbol{\varphi}_B^{\vec{b}},\boldsymbol{\theta}^{\vec{b}}}\right\rangle \equiv \sum_{\vec{x}_B\in\mathcal{D}_{m'}^{B}} \sum_{S\in\mathcal{S}_B} \Theta^{\vec{b}}[S]\left(\prod_{j\in\vec{b}}\varphi_{S_j x_j}\right)|\vec{x}_B\rangle \text{ (156)}
$$

Here $\boldsymbol{\varphi}_A^{\vec{a}}$ is the $m\times N_A$ matrix (where $m$ is the number of symbols in choice $\vec{a}$ and $N_A$ is the number of sites in part $A$) that is obtained from the $M\times N$ matrix $\boldsymbol{\varphi}$ by keeping the $N_A$ first elements in the $m$ rows indicated by the components of $\vec{a}$, whereas $\boldsymbol{\varphi}_B^{\vec{b}}$ the $m'\times N_B$ matrix by keeping the $N_B$ last elements in the $m'$ rows indicated by the components of $\vec{b}$.

For the construction of exact MPS and TTN representations, tensor $\mathbb{T}$ in Eq. (82) remains unchanged except

that it now depends on possibly complex scattering angles. On the other hand, Eqs. (90) and (91) for a local Bethe wavefunction $|\vec{a}_i\rangle$ in part $A_i$ and the corresponding components of tensor $\mathbb{S}[i]$ are now simply replaced with

$$
|\vec{a}_i\rangle \equiv \sum_{\vec{x}_{A_i}\in\mathcal{D}_m^{A_i}} \sum_{R\in S_m} \Theta^{\vec{a}_i}[R]\left(\prod_{j\in\vec{a}_i}\varphi_{R_j x_j}\right)|\vec{x}_{A_i}\rangle , \quad \text{(157)}
$$

and

$$
\mathbb{S}[i]_{\vec{\sigma}_{A_i}}^{\vec{a}_i} = \sum_{R\in S_m} \Theta^{\vec{a}_i}[R] \sum_{\vec{x}\in\mathcal{D}_m^{A_i}} \left(\prod_{j\in\vec{a}_i}\varphi_{R_j x_j}\right)\langle\vec{\sigma}_{A_i}|\vec{x}\rangle . \text{ (158)}
$$

In summary, we have generalized the Bethe wavefunction by introducing generalizations of the two ingredients ($M$ distinct single-particle wavefunctions and $M(M-1)/2$ scattering angles) used to build them, and by then following essentially the same recipe to turn those ingredients into an $M$-particle wavefunction, namely gluing together parts of the single-particle wavefunctions using a scattering amplitude $\Theta[P]$ that factorizes into products of 2-particle scattering amplitudes. Since a number of derivations in this paper were based on that recipe, they are also valid for the resulting generalized Bethe wavefunction introduced in this section.

Identifying in which systems the generalized Bethe wavefunction may result useful as an exact or approximate ansatz is beyond the scope of this work. However, it is natural to speculate that it may find applications in the study of particle-preserving systems that are not translation invariant, where plane-wave states do not represent single-particle eigenstates.

## VIII. CONCLUDING REMARKS

The Bethe wavefunction was originally introduced almost a century ago by Hans Bethe in order to solve the spin half Heisenberg chain, and has since become key to solving a number of integrable one dimensional systems [1–9]. Our work is based on applying the lens of quantum information and quantum computation [10–25] to such wavefunctions and builds upon, and overlaps with, a number of other recent, related contributions [26–38].

Here we have uncovered a remarkable property of a Bethe wavefunction, namely that it accepts a fractal decomposition that expresses it as a linear combination of products of local Bethe wavefunctions. For an $M$-particle Bethe wavefunction on a 1d lattice $\mathcal{L}$ made of $N$ sites, the fractal bipartite decomposition includes at most $2^M$ terms, leading to an $\chi = 2^M$ upper bound on the bipartite Schmidt rank that is independent of the size $N$ of the lattice as well as the size of the two parts. More generally, when lattice $\mathcal{L}$ is divided into $L$ parts, the fractal multipartite decomposition is also always possible and contains at most $L^M$ terms, leading to a $\chi = L^M$ upper bound on

the multipartite Schmidt rank. This signals at a particularly restrictive entanglement structure, when compared to generic $M$-particle wavefunctions or even $M$-particle ground states of particle-preserving local Hamiltonians.

The existence of such fractal decompositions has nontrivial consequences for the tensor network representability of Bethe wavefunctions. We provided exact tensor network representations, with bond dimension $\chi = 2^M$, for any planar TTN, which includes an MPS and a regular binary TTN as particular examples. From the regular binary TTN, we obtained a $\log(N)$ depth adaptive quantum circuit to prepare Bethe wavefunctions on a quantum circuit. This logarithmic depth is an exponential improvement over previous linear depth unitary preparations based on the MPS representations.

It is natural to ask how our results may contribute to solving many-body problems - the original purpose of Bethe wavefunctions. The following three lines of investigation deserve further attention. On the one hand, the tensor network representations provided here, including the homogeneous MPS and TTN representations, open the path to highly efficient ways of computing overlaps and norms, as well as long-range and high-order correlation functions - and we provided some partial analysis of such possibilities. Such exact tensor network representations can also be useful in order to benchmark the performance of novel algorithms to optimize a tensor network to represent a ground state or an excited state, since it provides exact solutions that the optimization algorithm should be able to find.

Secondly, the $O(\log(N))$-depth proposed quantum circuits may boost the interest in realizing Bethe wavefunctions on a quantum computer, given that they require significantly less time/gates than previous proposals and, as a result, may be significantly easier to implement in current and short-term NISQ devices, where they could be used to explore dynamical properties that are otherwise challenging to simulate classically. Notice that, additionally, the quantum circuit based on our TTN representation can be very significantly simplified (by only implementing the gates in the past causal cone of preselected lattice sites) in order to compute long-range or high-order correlation functions.

Finally, the generalized Bethe wavefunction that we put forward, which also accepts exact tensor network and quantum circuit representations, may become useful as an ansatz for models that are not translation invariant, including those obtained by local deformations of translation invariant, integrable models.

**Note added:** Upon completion of our work we learned [1] of previous results reported in Eq 5.11 of Page 153 of Ref. [6] by Korepin, Bogoliubov and Izergin, who built what amounts to a fractal bipartite decomposition for *constrained* Bethe wavefunctions (namely for Bethe wavefunctions satisfying integrability constraints) using the Algebraic Bethe Ansatz formalism, see also Refs. [42–44]. Our fractal bipartite decomposition for *unconstrained* Bethe wavefunctions (not required to satisfy integrability constraints) in Sec. III can therefore be understood as a generalization of that bipartite decomposition to a larger class of wavefunction - which we then also extend further in Sec. VII to the much larger class *generalized* Bethe wavefunctions, see Fig. 2.

**Acknowledgements:**

We thank Roberto Ruiz, Esparanza López, and Germán Sierra for a careful reading of the manuscript and thoughtful suggestions. G.V. is a CIFAR associate fellow in the Quantum Information Science Program, a Distinguished Invited Professor at the Institute of Photonic Sciences (ICFO), and a Distinguished Visiting Research Chair at Perimeter Institute. Research at Perimeter Institute is supported in part by the Government of Canada through the Department of Innovation, Science and Economic Development Canada and by the Province of Ontario through the Ministry of Colleges and Universities.

---

[1] H. Bethe. Zur Theorie der Metalle. i. Eigenwerte und Eigenfunktionen der linearen Atomkette. Zeit. für Physik, 71:205, 1931. ; English translation in D. C. Mattis., ed. The Many-Body Problem. World Scientific, Singapore.

[2] C. N. Yang and C. P. Yang, One-dimensional chain of anisotropic spin-spin interactions. ii. properties of the ground-state energy per lattice site for an infinite system, Phys. Rev. **150**, 327 (1966).

[3] E. H. Lieb and W. Liniger, Exact analysis of an interacting bose gas. i. the general solution and the ground state, Phys. Rev. **130**, 1605 (1963).

[4] E. H. Lieb and F. Y. Wu, Absence of mott transition in an exact solution of the short-range, one-band model in one dimension, Phys. Rev. Lett. **21**, 192 (1968).

[5] L. Faddeev, How the algebraic bethe ansatz works for integrable models, in *Fifty Years of Mathematical Physics*, pp. 370–439, arXiv:hep-th/9605187 [hep-th].

[6] V. E. Korepin, N. M. Bogoliubov, and A. G. Izergin, *Quantum Inverse Scattering Method and Correlation Functions*, Cambridge Monographs on Mathematical Physics (Cambridge University Press, 1993).

[7] J.-S. Caux and J. Mossel, Remarks on the notion of quantum integrability, Journal of Statistical Mechanics: Theory and Experiment **2011**, P02023 (2011).

[8] B. M. Wouters *et al.*, *The quench action approach to out-of-equilibrium quantum integrable models* (Universiteit van Amsterdam [Host], 2015).

[9] F. Franchini, *An Introduction to Integrable Techniques for One-Dimensional Quantum Systems* (Springer International Publishing, 2017).

---

[1] Roberto Ruiz, private communication.

[10] G. Vidal, J. I. Latorre, E. Rico, and A. Kitaev, Entanglement in quantum critical phenomena, Physical Review Letters **90**, 10.1103/physrevlett.90.227902 (2003).

[11] A. Kitaev and J. Preskill, Topological entanglement entropy, Physical Review Letters **96**, 10.1103/physrevlett.96.110404 (2006).

[12] M. Levin and X.-G. Wen, Detecting topological order in a ground state wave function, Physical Review Letters **96**, 10.1103/physrevlett.96.110405 (2006).

[13] M. B. Hastings, An area law for one-dimensional quantum systems, Journal of Statistical Mechanics: Theory and Experiment **2007**, P08024–P08024 (2007).

[14] M. Fannes, B. Nachtergaele, and R. F. Werner, Finitely correlated states on quantum spin chains, Communications in Mathematical Physics **144**, 443 (1992).

[15] J. I. Latorre, E. Rico, and G. Vidal, Ground state entanglement in quantum spin chains (2004), arXiv:quant-ph/0304098 [quant-ph].

[16] F. Verstraete and J. I. Cirac, Matrix product states represent ground states faithfully, Physical Review B **73**, 10.1103/physrevb.73.094423 (2006).

[17] F. Verstraete and J. I. Cirac, Renormalization algorithms for quantum-many body systems in two and higher dimensions (2004), arXiv:cond-mat/0407066 [cond-mat.str-el].

[18] Y.-Y. Shi, L.-M. Duan, and G. Vidal, Classical simulation of quantum many-body systems with a tree tensor network, Physical Review A **74**, 10.1103/physreva.74.022320 (2006).

[19] G. Vidal, Entanglement renormalization, Phys. Rev. Lett. **99**, 220405 (2007).

[20] G. Vidal, Class of quantum many-body states that can be efficiently simulated, Phys. Rev. Lett. **101**, 110501 (2008).

[21] C. Schön, E. Solano, F. Verstraete, J. I. Cirac, and M. M. Wolf, Sequential generation of entangled multiqubit states, Physical Review Letters **95**, 10.1103/physrevlett.95.110503 (2005).

[22] I. H. Kim and B. Swingle, Robust entanglement renormalization on a noisy quantum computer (2017), arXiv:1711.07500 [quant-ph].

[23] Z.-Y. Wei, D. Malz, and J. I. Cirac, Sequential generation of projected entangled-pair states, Phys. Rev. Lett. **128**, 010607 (2022).

[24] M. Foss-Feig, D. Hayes, J. M. Dreiling, C. Figgatt, J. P. Gaebler, S. A. Moses, J. M. Pino, and A. C. Potter, Holographic quantum algorithms for simulating correlated spin systems, Phys. Rev. Res. **3**, 033002 (2021).

[25] D. Malz, G. Styliaris, Z.-Y. Wei, and J. I. Cirac, Preparation of matrix product states with log-depth quantum circuits, Physical Review Letters **132**, 10.1103/physrevlett.132.040404 (2024).

[26] T. J. Sewell and S. P. Jordan, Preparing renormalization group fixed points on nisq hardware (2021), arXiv:2109.09787 [quant-ph].

[27] S. Anand, J. Hauschild, Y. Zhang, A. C. Potter, and M. P. Zaletel, Holographic quantum simulation of entanglement renormalization circuits, PRX Quantum **4**, 030334 (2023).

[28] R. Haghshenas, E. Chertkov, M. DeCross, T. M. Gatterman, J. A. Gerber, K. Gilmore, D. Gresh, N. Hewitt, C. V. Horst, M. Matheny, T. Mengle, B. Neyenhuis, D. Hayes, and M. Foss-Feig, Probing critical states of matter on a digital quantum computer (2023), arXiv:2305.01650 [quant-ph].

[29] R. Ruiz, A. Sopena, M. H. Gordon, G. Sierra, and E. López, The bethe ansatz as a quantum circuit, Quantum **8**, 1356 (2024).

[30] D. Raveh and R. I. Nepomechie, Deterministic bethe state preparation (2024), arXiv:2403.03283 [quant-ph].

[31] R. Ruiz, A. Sopena, E. López, G. Sierra, and B. Pozsgay, Bethe ansatz, quantum circuits, and the f-basis (2024), arXiv:2411.02519 [quant-ph].

[32] F. C. Alcaraz and M. J. Lazo, Generalization of the matrix product ansatz for integrable chains, Journal of Physics A: Mathematical and General **39**, 11335 (2006).

[33] H. Katsura and I. Maruyama, Derivation of the matrix product ansatz for the heisenberg chain from the algebraic bethe ansatz, Journal of Physics A: Mathematical and Theoretical **43**, 175003 (2010).

[34] V. Murg, V. E. Korepin, and F. Verstraete, Algebraic bethe ansatz and tensor networks, Physical Review B **86**, 10.1103/physrevb.86.045125 (2012).

[35] R. I. Nepomechie, Bethe ansatz on a quantum computer? (2021), arXiv:2010.01609 [quant-ph].

[36] J. S. Van Dyke, G. S. Barron, N. J. Mayhall, E. Barnes, and S. E. Economou, Preparing bethe ansatz eigenstates on a quantum computer, PRX Quantum **2**, 10.1103/prxquantum.2.040329 (2021).

[37] J. S. Van Dyke, E. Barnes, S. E. Economou, and R. I. Nepomechie, Preparing exact eigenstates of the open xxz chain on a quantum computer, Journal of Physics A: Mathematical and Theoretical **55**, 055301 (2022).

[38] A. Sopena, M. H. Gordon, D. García-Martín, G. Sierra, and E. López, Algebraic bethe circuits, Quantum **6**, 796 (2022).

[39] J. J. Mossel *et al.*, *Quantum integrable models out of equilibrium*, Ph.D. thesis, Universiteit van Amsterdam [Host] (2012).

[40] J. Eisert and H. J. Briegel, Schmidt measure as a tool for quantifying multiparticle entanglement, Physical Review A **64**, 10.1103/physreva.64.022306 (2001).

[41] T.-C. Lu, L. A. Lessa, I. H. Kim, and T. H. Hsieh, Measurement as a shortcut to long-range entangled quantum matter, PRX Quantum **3**, 10.1103/prxquantum.3.040337 (2022).

[42] A. G. Izergin and V. E. Korepin, The quantum inverse scattering method approach to correlation functions, Communications in mathematical physics **94**, 67 (1984).

[43] A. Izergin and V. Korepin, Correlation functions for the heisenberg xxz-antiferromagnet, Communications in mathematical physics **99**, 271 (1985).

[44] A. Izergin, V. Korepin, and N. Y. Reshetikhin, Correlation functions in a one-dimensional bose gas, Journal of Physics A: Mathematical and General **20**, 4799 (1987).