# Peer review of "Fractal decompositions and tensor network representations of Bethe wavefunctions"

_SciPost_

## Round 2 · Referee Report · Anonymous (Referee 2) · 2025-7-22

Strengths
1- The paper has a generalization of Bethe states, which is very useful for theoretical studies. I think that the generalization in Section VII did not appear previously in any work, definitely not in this form. This is a valuable addition to the literature.
Weaknesses
1- The authors are not familiar with many earlier results.
Report
The later parts are fine. I think the TTN part is completely new. And while the authors do not solve the state preparation with polynomially complex unitary circuits (which might or might not be possible), they give useful new results.
Requested changes
-
Refs [7-9] are not really reviews about Algebraic Bethe Ansatz. The authors state that "for recent reviews" we should consult these papers. But [7] is about the notion of integrability (very general!), [8] is about quench action (no nested Bethe Ansatz), and [9] is a good introductory material, but again, it does not really review new results about Algebraic Bethe Ansatz. Actually I don't know a good "review", but for example https://arxiv.org/abs/1804.07350 does talk about Algebraic Bethe Ansatz in depth, and in fact it explains some old results that are relevant. For nested Bethe Ansatz I would suggest perhaps https://arxiv.org/abs/1911.12811 The authors can choose to include these references or similar ones, and/or just replace the wording "for recent reviews" because those are not recent reviews about Algebrai Bethe Ansatz.
-
The authors write in the Introduction: "In spite of the solvable character of integrable models, the exact com- putation of their long-range and high order correlation functions and of dynamical properties remains a chal- lenging task." There should be some references. I understand that the authors are not experts in this area, nevertheless they should make an effort. If I had to collect just one document, then I would cite the habilitation thesis of Karol Kozlowski: https://arxiv.org/abs/1508.06085 This was 10 years ago, and it includes lots of exact results about long range correlation functions in the ground state. Maybe there are more recent reviews too. And if I had the chance to include one more document, I would add https://journals.aps.org/prl/abstract/10.1103/PhysRevLett.106.257201 This has lots of information also about generic higher point correlation functions, including finite temperature situations. But this group of authors has many many more recent works, unfortunately no reviews, as far as I know. And for excited state correlation functions see https://arxiv.org/abs/1605.09347 So actually a lot is known, but these computations can indeed be challenging. Something should be added, it is up to the authors whether they add these references or other ones. But the statement without references is definitely not good.
-
In the intro the authors write: "we uncover a surprising property of Bethe wavefunctions" This result that the authors have is actually well known. What they do later, namely the tree tensor network, is their own. But the fact that the Bethe wave function can be decomposed in this particular way, is very well known since a long time. It is a new result for the generalization that they have, but for previously studied Bethe wave functions this is not new. The authors acknowledge this is a "Note added". However, this is not sufficient for publication in a journal. I understand that perhaps the authors wanted to push the paper to the arxiv, for a variety of reasons. But now when they have the time to prepare the submission, they should make an effort, and add comments wherever appropriate. The paper should not appear this way, that there is just a single Note at the end, explaining that so many results were known since the 80's.
Recommendation
Ask for minor revision
Report
Recommendation
Publish (meets expectations and criteria for this Journal)

---

## Round 3 · Referee Report · Anonymous (Referee 2) · 2025-9-5

Strengths

-

Weaknesses

-

Report

The authors addressed my concerns. The paper can be published soon, but I would like to ask for one more minor modification, see below.

Requested changes

The authors really made an effort to make the paper readable, and to refer to previous literature. However, one issue remains.

In writing down Theorem 3.1, there is no reference to previous work. This might be a matter of style, but in my point of view, when the authors claim that this is their main new result, and if in some special cases it appeared in the literature before, then the authors should give a quick citation also at that point.

The precise statement is that Theorem 3.1 is NEW, I agree with the authors. It is new, because it concerns the most general Bethe Ansatz wave function. However, specifically for Bethe states of the XXZ model, given by ABA, this theorem was known.

Therefore, I would like to ask the authors, to just give a quick footnote or something, that in the special case of XXZ spin chain, computed via Algebraic Bethe Ansatz, this theorem was known. Reference could be the QISM book [6], or some concrete paper. (any equivalent acknowledgement of the known result is also good)

Recommendation

Ask for minor revision

---

## Round 3 · Author Response

We thank referee 2 for their detailed comments and suggestions. We have addressed their concerns in the current version, with the changes listed below. Overall, we have highlighted the new findings of this work in the context of the fractal (or nested) decomposition that was already known before, while also highlighting the existing literature better in the introduction.

---

## Round 3 · List of Changes

1. Modified the abstract and the inroduction to highlight the fact that the first part of the paper reviews known facts about decomposition of Bethe wavefunctions, and contextualize our contribution for the tensor network constructions in that regard.
  2. Added more references for Algebraic Bethe ansatz.
  3. Corrected a few typos.

---

## Round 4 · List of Changes

We have added a clarification near Theorem 3.1 as per the referee's suggestion.

---

## Editorial Decision

unknown